# Probabilistic Inference in Reinforcement Learning Done Right

**Jean Tarbouriech**
Google DeepMind
jtarbouriech@google.com

**Tor Lattimore**
Google DeepMind
lattimore@google.com

**Brendan O'Donoghue**
Google DeepMind
bodonoghue@google.com

## Abstract

A popular perspective in Reinforcement learning (RL) casts the problem as probabilistic inference on a graphical model of the Markov decision process (MDP). The core object of study is the probability of each state-action pair being visited under the optimal policy. Previous approaches to approximate this quantity can be arbitrarily poor, leading to algorithms that do not implement genuine statistical inference and consequently do not perform well in challenging problems. In this work, we undertake a rigorous Bayesian treatment of the posterior probability of state-action optimality and clarify how it flows through the MDP. We first reveal that this quantity can indeed be used to generate a policy that explores efficiently, as measured by regret. Unfortunately, computing it is intractable, so we derive a new variational Bayesian approximation yielding a tractable convex optimization problem and establish that the resulting policy also explores efficiently. We call our approach `VAPOR` and show that it has strong connections to Thompson sampling, K-learning, and maximum entropy exploration. We conclude with some experiments demonstrating the performance advantage of a deep RL version of `VAPOR`.

## 1 Introduction

Reinforcement learning (RL) is the problem of learning to control an unknown system by taking actions to maximize its cumulative reward through time [76]. As the agent navigates the environment, it receives noisy observations which it can use to update its (posterior) beliefs about the environment [21]. Unlike supervised learning, where the performance of an algorithm does not influence the data it will later observe, in RL the policy of the agent affects the data it will collect, which in turn affects the policy, and so on. As a result, an agent must sometimes take actions that lead to states where it has *epistemic uncertainty* about the value of those states, and sometimes take actions that lead to more certain payoff. The tension between these two modes is the *exploration-exploitation* trade-off [2, 35]. In this light, RL is a *statistical inference problem* wrapped in a *control problem*, and the two problems must be tackled simultaneously for good data efficiency.

It is natural to apply Bayesian probabilistic inference to the uncertain parameters in RL [21] and, since the goal of the agent is to find the optimal policy, a relevant object of study is the posterior probability of *optimality* for each state-action pair. In fact, the popular 'RL as inference' approach, most clearly summarized in the tutorial and review of [37], embeds the control problem into a graphical model by introducing optimality binary random variables that indicate whether a state-action is optimal. In the 'RL as inference' setup these optimality variables are assumed to be observed and take the value one with probability depending on the local reward. This surrogate potential seems like a peculiar and arbitrary choice, yet it has been widely accepted due to its mathematical convenience: for deterministic dynamics, the resulting probabilistic inference is equivalent to maximum entropy RL [80, 87]. However, this approximation leads to a critical shortcoming: The surrogate potential functions do not take into account the Bayesian (epistemic) uncertainty, and consequently do not perform genuine statistical inference on the unknowns in the MDP. This leads to 'posteriors' that are

37th Conference on Neural Information Processing Systems (NeurIPS 2023).

in no way related to the *true* posteriors in the environment, and acting with those false posteriors leads to poor decision making and agents that require exponential time to solve even small problems [53].

Our contributions can be centered around **a new quantity that we uncover as key for inference and control**, denoted by $\mathbb{P}_{\Gamma^\star}$, which represents the *posterior probability of each state-action pair being visited under the optimal policy*.

- We reveal that $\mathbb{P}_{\Gamma^\star}$ formalizes from a Bayesian perspective what it means for a 'state-action pair to be optimal', an event at the core of the 'RL as inference' framework which had never been properly analyzed.
- We establish that knowledge of $\mathbb{P}_{\Gamma^\star}$ is sufficient to derive a policy that explores efficiently, as measured by regret (Section 3).
- Since computing $\mathbb{P}_{\Gamma^\star}$ is intractable, we propose a variational optimization problem that tractably approximates it (Section 4).
- We first solve this optimization problem exactly, resulting in a new tabular model-based algorithm with a guaranteed regret bound (Section 5).
- We then solve a variant of this optimization problem using policy-gradient techniques, resulting in a new scalable model-free algorithm (Section 7).
- We show that both Thompson sampling [78, 75, 59, 71] and K-learning [47] can be directly linked to $\mathbb{P}_{\Gamma^\star}$, thus shedding a new light on these algorithms and tightly connecting them to our variational approach (Section 6).
- Our approach has the unique algorithmic feature of adaptively tuning optimism and entropy regularization for *each* state-action pair, which is empirically beneficial as our experiments on 'DeepSea' and Atari show (Section 8).

## 2 Preliminaries

We model the RL environment as a finite state-action, time-inhomogeneous MDP given by the tuple $\mathcal{M} := \{\mathcal{S}, \mathcal{A}, L, P, R, \rho\}$, where $L$ is the horizon length, $\mathcal{S} = \mathcal{S}_1 \cup \ldots \cup \mathcal{S}_L$ is the state space with cardinality $S = \sum_{l=1}^{L} S_l$ where $S_l = |\mathcal{S}_l|$, $\mathcal{A}$ is the action space with $A$ possible actions, $P_l : \mathcal{S}_l \times \mathcal{A} \to \Delta(\mathcal{S}_{l+1})$ denotes the transition dynamics at step $l$, $R_l : \mathcal{S}_l \times \mathcal{A} \to \Delta(\mathbb{R})$ is the reward function at step $l$ and $\rho \in \Delta(\mathcal{S}_1)$ is the initial state distribution. Concretely, the initial state $s_1 \in \mathcal{S}_1$ is sampled from $\rho$, then for steps $l = 1, \ldots, L$ the agent is in state $s_l \in \mathcal{S}_l$, selects action $a_l \in \mathcal{A}$, receives a reward sampled from $R_l(s_l, a_l)$ with mean $r_l(s_l, a_l) \in \mathbb{R}$ and transitions to the next state $s_{l+1} \in \mathcal{S}_{l+1}$ with probability $P_l(s_{l+1} \mid s_l, a_l)$. An agent following a policy $\pi_l \in \Delta(\mathcal{A})^{S_l}$ at state $s$ at step $l$ selects action $a$ with probability $\pi_l(s, a)$. The value functions are defined as

$$Q_l^\pi(s, a) = r_l(s, a) + \sum_{s' \in \mathcal{S}_{l+1}} P_l(s' \mid s, a) V_{l+1}^\pi(s'), \quad V_l^\pi(s) = \sum_{a \in \mathcal{A}} \pi_l(s, a) Q_l^\pi(s, a), \qquad V_{L+1}^\pi = 0,$$

$$Q_l^\star(s, a) = r_l(s, a) + \sum_{s' \in \mathcal{S}_{l+1}} P_l(s' \mid s, a) V_{l+1}^\star(s'), \quad V_l^\star(s) = \max_{a \in \mathcal{A}} Q_l^\star(s, a), \qquad V_{L+1}^\star = 0.$$

An optimal policy $\pi^\star$ satisfies $V_l^{\pi^\star}(s) = V_l^\star(s)$ for each state $s$ and step $l$. Let $\{\pi_l^\star(s) = a\}$ be the event that action $a$ is optimal for state $s$ at step $l$, with ties broken arbitrarily so that only one action is optimal at each state. Let $\mathbb{R}_+^{L,S,A} := \{\mathbb{R}_+^{S_l \times A}\}_{l=1}^{L}$ be the set of functions $\mathcal{S}_l \times \mathcal{A} \to \mathbb{R}_+$ for $l \in [L]$.

**Occupancy measures.** We denote the set of occupancy measures (a.k.a. stationary state-action distributions) with respect to the transition dynamics $P$ as

$$\Lambda(P) := \left\{ \lambda \in \mathbb{R}_+^{L,S,A} \ : \ \sum_a \lambda_1(s, a) = \rho(s), \ \ \sum_{a'} \lambda_{l+1}(s', a') = \sum_{s,a} P_l(s' \mid s, a) \lambda_l(s, a) \right\}.$$

Note the correspondence between $\Lambda(P)$ and the set of policies $\Pi = \{\Delta(A)^{S_l}\}_{l=1}^{L}$ [66]. Any policy $\pi \in \Pi$ induces $\lambda^\pi \in \Lambda(P)$ such that $\lambda_l^\pi(s, a)$ denotes the probability of reaching $(s, a)$ at step $l$ under $\pi$. Conversely, any $\lambda \in \Lambda(P)$ induces $\pi^\lambda \in \Pi$ given by $\pi_l^\lambda(s, a) := \lambda_l(s, a)/(\sum_{a'} \lambda_l(s, a'))$ so long as $\sum_{a'} \lambda_l(s, a') > 0$; otherwise $\pi_l^\lambda(s, \cdot)$ can be any distribution, *e.g.*, uniform.

**Bayesian RL.** We consider the Bayesian view of the RL problem, where the agent has some beliefs about the MDP represented by a distribution $\phi$ on the space of all MDPs. We use the shorthand notation $\mathbb{E}_\phi$ and $\mathbb{P}_\phi$ to denote the expectation and probability under $\phi$. In the Bayesian view, the mean rewards $r$ and transition dynamics $P$ are random variables, and thus so are $Q^\star, V^\star, \pi^\star$.

## 2.1 Previous Approach to 'RL as Inference'

The popular line of research casting 'RL as inference' is most clearly summarized in the tutorial and review of Levine, 2018 [37]. It embeds the control problem into a graphical model, mirroring the dual relationship between optimal control and inference [81, 34, 68]. As shown in Figure 5a, this approach defines an additional *optimality* binary random variable denoted by $\Gamma_l^\star$, which indicates which state-action at timestep $l$ is 'optimal', *i.e.*, which state-action is visited under the (unknown) optimal policy. In these works the probability of $\Gamma_l^\star$ is then modeled as proportional to the exponentiated reward.

**Approximation 1** (Inference over exponentiated reward [37]). *Denoting by $\Gamma_l^\star(s, a)$ the event that state-action pair $(s, a)$ is 'optimal' at timestep $l$, set $\mathbb{P}(\Gamma_l^\star(s, a)) \propto \exp(r_l(s, a))$.*

Under Approximation 1, the optimality variables become observed variables in the graphical model, over which we can apply standard tools for inference. This approach has the advantage of being mathematically and computationally convenient. In particular, under deterministic dynamics it directly leads to RL algorithms with 'soft' Bellman updates and added entropy regularization, thus recovering the maximum-entropy RL framework [80, 87] and giving a natural, albeit heuristic exploration strategy. Many popular algorithms lie within this class [65, 25, 1] and have demonstrated strong empirical performance in domains where efficient exploration is not a bottleneck.

Despite its popularity and convenience, the potential function of Approximation 1 *ignores epistemic uncertainty* [53]. Due to this shortcoming, the inference procedure produces invalid and arbitrarily poor posteriors. It can be shown that the resulting algorithms fail even in basic bandit-like problems (*e.g.*, [53, Problem 1]), as they underestimate the probability of optimal actions and take actions which have no probability of optimality under the true posterior. This implies that the resulting agents (*e.g.*, Soft Q-learning [25]) can perform poorly in simple domains that require deep exploration [57]. The fact that the 'RL as inference' framework does not perform valid inference on the optimality variables leads us to consider how a proper Bayesian inference approach might proceed. This brings us to the first contribution of this manuscript.

## 3 A Principled Bayesian Inference Approach to RL

In this section, we provide a principled Bayesian treatment of statistical inference over 'optimality' variables in an MDP. Although it is the main quantity of interest of the popular 'RL as inference' framework, a rigorous definition of whether a state-action pair is optimal has remained elusive. We provide a recursive one below.

**Definition 1** (State-action optimality). *For any step $l \in [L]$, state-action $(s, a) \in \mathcal{S}_l \times \mathcal{A}$, we define*

$$\Gamma_1^\star(s) := \{s_1 = s\}, \qquad \Gamma_{l+1}^\star(s') := \bigcup_{s \in \mathcal{S}_l, a \in \mathcal{A}} \Gamma_l^\star(s, a) \cap \{s_{l+1} = s'\},$$

$$\Gamma_1^\star(s, a) := \Gamma_1^\star(s) \cap \{\pi_1^\star(s) = a\}, \qquad \Gamma_{l+1}^\star(s', a') := \Gamma_{l+1}^\star(s') \cap \{\pi_{l+1}^\star(s') = a'\}.$$

In words, $\Gamma_l^\star(s)$ is the *random* event that state $s$ is reached at step $l$ after executing optimal actions from steps 1 to $l - 1$, and the event $\Gamma_l^\star(s, a)$ further requires that the action $a$ is optimal at state $s$ in step $l$. Equipped with this definition, 'optimality' in an MDP flows both in a *backward* way (via action optimality $\pi^\star$, which can be computed using dynamic programming) and in a *forward* way (via state-action optimality $\Gamma^\star$). We illustrate this bidirectional property in the simplified graphical model of Figure 5b (which isolates a single trajectory). In general, state-action optimality encapsulates all possible trajectories leading to a given state being optimal (and there may be exponentially many of them). Thanks to the MDP structure and Bayes' rule, we show in Lemma 1 that the *posterior probability* of state-action optimality lies in the space of occupancy measures. To simplify exposition, we first assume that the transition dynamics $P$ are known (only rewards are unknown), and we extend our analysis to unknown $P$ in Section 5.

**Lemma 1.** *For known $P$, it holds that* $\mathbb{P}_{\Gamma^\star} := \{\mathbb{P}_\phi(\Gamma_l^\star)\}_{l=1}^L \in \Lambda(P)$.

The significance of Lemma 1 is that we can readily *sample* from $\mathbb{P}_{\Gamma^\star}$ with the induced policy

$$\pi_l(s, a) = \frac{\mathbb{P}_\phi(\Gamma_l^\star(s, a))}{\sum_{a' \in \mathcal{A}} \mathbb{P}_\phi(\Gamma_l^\star(s, a'))} = \mathbb{P}_\phi(\pi_l^\star(s) = a \mid \Gamma_l^\star(s)), \tag{1}$$

Figure 1: Pair of MDPs $\{\mathcal{M}^+, \mathcal{M}^-\}$ with uniform prior $\phi = (\frac{1}{2}, \frac{1}{2})$. $\mathcal{M}^+$ and $\mathcal{M}^-$ only differ through their reward at state $s_L$, resp. $+1$ and $-1$. Two actions are available at states $s_1$ to $s_{L-1}$: $\downarrow$ exits the chain and moves to an absorbing, zero-reward state, and $\rightarrow$ moves right with small negative reward $-\epsilon$.

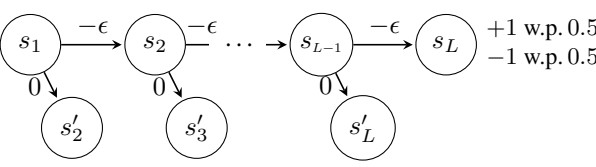

| Policy | Agent starts at $s_1$ $\pi_1(s_1, a)$ for $a \in \{\downarrow, \rightarrow\}$ | Agent reaches $s_l, l > 1$ $\pi_l(s_l, a)$ for $a \in \{\downarrow, \rightarrow\}$ | Expected # episodes to reach $s_L$ |
|---|---|---|---|
| Bayes-optimal | $(0, 1)$ | $(0, 1)$ | $1$ |
| Approximation 1 (e.g., Soft Q-Learning) | $(0.5, 0.5)$ | $(0.5, 0.5)$ | $\Omega(2^L)$ |
| Marginal $\mathbb{P}_\phi(\pi^\star)$ | $(0.5, 0.5)$ | $(0.5, 0.5)$ | $\Omega(2^L)$ |
| Conditional $\mathbb{P}_\phi(\pi^\star \mid \Gamma^\star)$ | $(0.5, 0.5)$ | $(0, 1)$ | $2$ |
| Thompson Sampling | $(0, 1)$ w.p. $0.5$ $(1, 0)$ w.p. $0.5$ | $(0, 1)$ | $2$ |
| VAPOR | $(0, 1)$ | $(0, 1)$ | $1$ |

Table 1: Comparison of algorithm performance on MDP in Figure 1. Unprincipled approaches like Soft Q-learning take time exponential in $L$ to reach the uncertain reward state. Naively using the marginal posterior probability of *action optimality* as the policy at each state also requires exponential time, however the policy derived from the posterior probability of *state-action optimality* $\mathbb{P}_{\Gamma^\star}$ takes *consistent* actions and is therefore efficient. Similar good performance is achieved by Thompson sampling (implicit approximation of $\mathbb{P}_{\Gamma^\star}$, Section 6) and VAPOR (explicit approximation of $\mathbb{P}_{\Gamma^\star}$, Section 4).

which is the probability under the posterior that action $a$ is optimal at state $s$ conditioned on all actions taken before timestep $l$ being optimal. It turns out that this policy explores efficiently, as measured by regret (see Corollary 1). Concretely, we have shown that the two facets of RL (inference and control) can be distilled into a single quantity: $\mathbb{P}_{\Gamma^\star}$. We use Bayesian inference to compute $\mathbb{P}_{\Gamma^\star}$ and then use that to derive a good control policy. This is a principled and consistent Bayesian 'RL as inference' approach. The remaining challenge thus lies in computing $\mathbb{P}_{\Gamma^\star}$. Unfortunately, doing so involves computing several complicated integrals with respect to the posterior and is intractable in most cases. In Section 4, we introduce the second contribution of this manuscript, which is a way to approximate $\mathbb{P}_{\Gamma^\star}$ using a computationally tractable variational approach.

### 3.1 Instructive Example

To gain intuition on the difference between *action optimality* (*i.e.*, $\mathbb{P}_\phi(\pi^\star)$) and *state-action optimality* (*i.e.*, $\mathbb{P}_{\Gamma^\star}$), we consider the simple decision problem of Figure 1. Each episode starts in the leftmost state and the agent can traverse a chain of states in order to reach the rightmost state, where the reward is *always* either $+1$ or $-1$ with equal probability. Once the agent reaches the rightmost state it will resolve its uncertainty. At each state the agent has the option to move $\rightarrow$, paying a cost of $\epsilon$ where $\epsilon L \ll 1$, or move $\downarrow$ with no cost. We see that there are only 2 consistent trajectories: either take action $\downarrow$ at state $s_1$ (optimal for $\mathcal{M}^-$), or repeat actions $\rightarrow$ to reach $s_L$ (optimal for $\mathcal{M}^+$). We detail how a selection of algorithms perform on this problem in Table 1.

It is worth examining in detail what has gone wrong for the policy that samples according to $\mathbb{P}_\phi(\pi^\star)$. Even if the agent has traveled to the right a few times it will exit the chain with probability $0.5$ at each new state, which leads to poor exploration. Taking action $\rightarrow$ a few times and then exiting the chain has zero probability of being optimal under the posterior, so this sequence of actions is *inconsistent* with the posterior. Conditioning on the event $\Gamma^\star(s)$ when at state $s$ forces the policy to be consistent with the set of the beliefs that would get the agent to state $s$ in the first place, which, as we prove later, yields deep exploration. In our chain example, the only belief that takes action $\rightarrow$ is the one in which the end state has reward $+1$, so the only consistent policy conditioned on having taken $\rightarrow$ once is to continue all the way to the end state. The same issue arises in Thompson sampling, where in order to achieve consistency the agent samples a policy from its posterior and keeps it fixed for the entire episode since resampling at every timestep leads to inefficient exploration [71]. As we shall show in Section 6, Thompson sampling is exactly equivalent to sampling from $\mathbb{P}_{\Gamma^\star}$. In conclusion:

| | | |
|---|---|---|
| action optimality: | $a_l \sim \mathbb{P}_\phi(\pi_l^\star(s_l) = \cdot)$ | $\Rightarrow$ Poor exploration, |
| state-action optimality: | $a_l \sim \mathbb{P}_\phi(\pi_l^\star(s_l) = \cdot \mid \Gamma_l^\star(s_l)) \propto \mathbb{P}_\phi(\Gamma_l^\star(s_l, \cdot))$ | $\Rightarrow$ Efficient exploration. |

# 4 A Variational Bayesian Approach

Having access to $\mathbb{P}_{\Gamma^\star}$ is sufficient to enable deep exploration, but computing this probability is intractable in general. In this section, we derive a *variational*, *i.e.*, optimization-based, approach to approximate $\mathbb{P}_{\Gamma^\star}$. In most variational Bayesian approaches [18], the optimization objective is to find a surrogate probability measure that minimizes some dissimilarity metric (*e.g.*, a KL-divergence) from the intractable probability measure of interest. Yet unlike standard variational inference techniques, we cannot minimize the KL-divergence to this distribution as we do not have access to samples. However, we do have a downstream *control* objective. That is to say, we are not simply interested in approximating the distribution, but also in our approximation doing well in the control problem. This insight provides an alternative objective that takes into account the rewards and uncertainties that the agent is likely to encounter. The optimal value function $V^\star$ is a random variable under the beliefs $\phi$. Our next lemma relates the expected value of $V^\star$ under $\phi$ to our state-action optimality event $\Gamma^\star$.

**Lemma 2.** *It holds that*
$$\mathbb{E}_{s\sim\rho}\mathbb{E}_\phi V_1^\star(s) = \sum_{l,s,a} \mathbb{P}_\phi(\Gamma_l^\star(s,a))\mathbb{E}_\phi\left[r_l(s,a) \mid \Gamma_l^\star(s,a)\right].$$

## 4.1 Information Theoretic Upper Bound

Lemma 2 depends on $\mathbb{P}_{\Gamma^\star}$, but the conditional expectation is not easy to deal with. To handle this, we upper bound each $\mathbb{E}_\phi[r_l(s,a) \mid \Gamma_l^\star(s,a)]$ in Lemma 2 using tools from information theory. To simplify the exposition, we consider the following standard sub-Gaussian assumption [70, 61, 47] and we defer to Appendix F.3 the treatment of the general case. We say that $X : \Omega \to \mathbb{R}$ is $\upsilon$-sub-Gaussian for $\upsilon > 0$ if $\mathbb{E}\exp(c(X - \mathbb{E}X)) \le \exp(c^2\upsilon^2/2)$, for all $c \in \mathbb{R}$.

**Lemma 3.** *If $r_l(s,a)$ is $\sigma_l(s,a)$-sub-Gaussian under $\phi$ for $l \in [L]$ and $(s,a) \in \mathcal{S}_l \times \mathcal{A}$, then*

$$\mathbb{E}_\phi\left[r_l(s,a) \mid \Gamma_l^\star(s,a)\right] \le \mathbb{E}_\phi r_l(s,a) + \min_{\tau_l(s,a)>0}\left(\frac{\sigma_l^2(s,a)}{2\tau_l(s,a)} - \tau_l(s,a)\log\mathbb{P}_\phi(\Gamma_l^\star(s,a))\right)$$

$$= \mathbb{E}_\phi r_l(s,a) + \sigma_l(s,a)\sqrt{-2\log\mathbb{P}_\phi(\Gamma_l^\star(s,a))}.$$

## 4.2 Optimization Problem

Finally, we combine Lemmas 2 and 3 to reveal a concave function in $\mathbb{P}_{\Gamma^\star}$ that upper bounds the value function objective $\mathbb{E}_{s\sim\rho}\mathbb{E}_\phi V_1^\star(s)$. For any $\tau \in \mathbb{R}_+^{L,S,A}$ and occupancy measure $\lambda \in \Lambda(P)$, we define the $\tau$-*weighted entropy* of $\lambda$ (summed over steps $l$) as $\mathcal{H}_\tau(\lambda) := -\sum_{l,s,a} \tau_l(s,a)\lambda_l(s,a)\log\lambda_l(s,a)$. In the following definition we take division and the square-root to be applied elementwise, and $\circ$ denotes elementwise multiplication. With this we can define the following *optimistic* value functions

$$\mathcal{V}_\phi(\lambda,\tau) := \lambda^\top\left(\mathbb{E}_\phi r + \frac{\sigma^2}{2\tau}\right) + \mathcal{H}_\tau(\lambda), \tag{2}$$

$$\mathcal{V}_\phi(\lambda) := \min_{\tau\in\mathbb{R}_+^{L,S,A}}\mathcal{V}_\phi(\lambda,\tau) = \lambda^\top\left(\mathbb{E}_\phi r + \sigma\circ\sqrt{-2\log\lambda}\right). \tag{3}$$

**Lemma 4.** *For known $P$ and $\sigma$-sub-Gaussian $r$, we have*

$$\mathbb{E}_{s\sim\rho}\mathbb{E}_\phi V_1^\star(s) \le \mathcal{V}_\phi(\mathbb{P}_{\Gamma^\star}) \le \underbrace{\max_{\lambda\in\Lambda(P)}\mathcal{V}_\phi(\lambda)}_{\text{VAPOR}}, \tag{4}$$

*where the* VAPOR *optimization problem is concave.*

The above optimization problem is our *variational* objective, the solution of which yields an occupancy measure that approximates $\mathbb{P}_{\Gamma^\star}$. We call the approach 'VAPOR', for variational approximation of the posterior probability of optimality in RL. We discuss various properties of the optimization problem in Appendix E (its unconstrained dual problem and message passing interpretation).

Since we are using variational inference a natural question to ask is how well our variational solution approximates $\mathbb{P}_{\Gamma^\star}$. Here we provide a bound quantifying the dissimilarity between $\mathbb{P}_{\Gamma^\star}$ and the solution of the VAPOR optimization problem, according to a weighted KL-divergence, where for any $\tau \in \mathbb{R}_+^{L,S,A}$, $\lambda, \lambda' \in \Lambda(P)$ we define $\mathrm{KL}_\tau(\lambda \,||\, \lambda') := \sum_{l,s,a} \tau_l(s,a)\lambda_l(s,a)\log(\lambda_l(s,a)/\lambda_l'(s,a))$.

**Lemma 5.** *Let $\lambda^*$ solve the* VAPOR *optimization problem* (4) *and $\tau^* \in \operatorname{argmin}_\tau \mathcal{V}_\phi(\lambda^*, \tau)$, then*

$$\mathrm{KL}_{\tau^*}(\mathbb{P}_{\Gamma^\star} \mid\mid \lambda^*) \leq \mathcal{V}_\phi(\lambda^*) - \mathbb{E}_{s \sim \rho}\mathbb{E}_\phi V_1^\star(s).$$

In other words, the (weighted) KL-divergence between $\mathbb{P}_{\Gamma^\star}$ and $\lambda^*$ is upper bounded by how loose our upper bound in (4) is. In practice however, we care less about a bound on the divergence metric than we do about performance in the control problem. Our next result shows that the variational policy also satisfies a strong sub-linear Bayesian regret bound, *i.e.*, the resulting policy performs well.

### 4.3 Bayesian Regret Analysis

**Learning problem.** We consider that the agent interacts with the MDP $\mathcal{M}$ over a (possibly unknown) number of $N$ episodes. We denote by $\mathcal{F}_t$ the sigma-algebra generated by all the history (*i.e.*, sequences of states, actions and rewards) before episode $t$, with $\mathcal{F}_1 = \emptyset$. We let $\mathbb{E}^t[\cdot] = \mathbb{E}[\cdot \mid \mathcal{F}_t]$, $\mathbb{P}^t(\cdot) = \mathbb{P}(\cdot \mid \mathcal{F}_t)$. We denote by $n_l^t(s, a)$ the visitation count to $(s, a)$ at step $l$ before episode $t$, and $(\cdot \vee 1) := \max(\cdot, 1)$. In the Bayesian approach where $\mathcal{M}$ is sampled from a known prior $\phi$, we want to minimize the *Bayes regret* over $T := NL$ timesteps of an algorithm alg producing policy $\pi^t$ at each episode $t$, which is defined as its expected regret under that prior distribution

$$\mathcal{R}_\mathcal{M}(\mathrm{alg}, T) := \sum_{t=1}^{N} \mathbb{E}_{s \sim \rho}\left(V_1^\star(s) - V_1^{\pi^t}(s)\right), \qquad \mathcal{BR}_\phi(\mathrm{alg}, T) := \mathbb{E}_{\mathcal{M} \sim \phi}\mathcal{R}_\mathcal{M}(\mathrm{alg}, T).$$

The VAPOR learning algorithm proceeds as follows: at the beginning of each episode, it solves the VAPOR optimization problem and executes the induced policy. We now show that it enjoys a sub-linear Bayesian regret bound under the following standard assumption [61, 47, 62].

**Assumption 1.** *The mean rewards are bounded in $[0, 1]$ almost surely with independent priors and the reward noise is additive $\nu$-sub-Gaussian for a constant $\nu > 0$.*

Assumption 1 implies that the mean rewards are sub-Gaussian under the posterior (see [70, App. D.2]), where we can upper bound the sub-Gaussian parameter $\sigma_l^t(s, a) \leq \sqrt{(\nu^2 + 1)/(n_l^t(s, a) \vee 1)}$ at the beginning of each episode $t$.

**Theorem 1.** *For known $P$ and under Assumption 1, it holds that*

$$\mathcal{BR}_\phi(\mathtt{VAPOR}, T) \leq \sqrt{2(\nu^2 + 1)TSA\log(SA)(1 + \log T/L)} = \widetilde{O}(\sqrt{SAT}).$$

In the above $\widetilde{O}$ suppresses log factors. The same regret bound holds for the (intractable) algorithm that uses (1) as the policy each episode.

**Corollary 1.** *Denote by $\mathrm{alg}_{\Gamma^\star}$ the algorithm that produces policies based on $\mathbb{P}_{\Gamma^\star}^t$ for each episode $t$ using* (1). *Then, under the same conditions as Theorem 1, we have $\mathcal{BR}_\phi(\mathrm{alg}_{\Gamma^\star}, T) \leq \widetilde{O}(\sqrt{SAT})$.*

### 4.4 Interpretation of VAPOR

Inspecting $\mathcal{V}_\phi$ (3) reveals that the VAPOR optimization problem is equivalent to solving a two-player zero-sum game between a 'policy' player $\lambda \in \Lambda(P)$ and a 'temperature' player $\tau \in \mathbb{R}_+^{L,S,A}$. The latter finds the tightest $\tau$ that best balances two exploration mechanisms on a per state-action basis: an *optimism* term $\sigma^2/\tau$ that augments the expected reward $\mathbb{E}_\phi r$, and a $\tau$-weighted *entropy regularization* term. The policy player maximizes the entropy-regularized optimistic reward.

The fact that entropy regularization falls out naturally from VAPOR is interesting because the standard 'RL as inference' framework conveniently reveals policy entropy regularization (Section 2.1), which has been widely studied theoretically [45, 20] and empirically for deep RL exploration [42, 26]. The specificity here is that it is with respect to the *occupancy measure* instead of the policy, and it is *adaptively weighted for each state-action pair*. This enables us to obtain an 'RL as inference' framework with entropy regularization that explores provably efficiently. Regularizing with the (weighted) entropy of the occupancy measure is harder to implement in practice, therefore in Section 7 we propose a principled, albeit looser, upper bound of the VAPOR optimization problem that regularizes the optimistic reward with only the (weighted) entropy of the *policy*, making the resulting objective amenable to online optimization with policy gradients.

---

**Algorithm 1** `VAPOR` learning algorithm

---

**For** episode $t = 1, 2, \ldots$ **do**

**1.** Compute expected rewards $\mathbb{E}^t r$, transitions $\mathbb{E}^t P$, uncertainty measure $\widehat{\sigma}^t$

**2.** Solve `VAPOR` optimization problem $\lambda^t \in \mathrm{argmax}_{\lambda \in \Lambda(\mathbb{E}^t P)} \mathcal{V}_{\widehat{\phi}^t}(\lambda)$ from Equation (3)

**3.** Execute policy $\pi_l^t(s, a) \propto \lambda_l^t(s, a)$, for $l = 1, \ldots, L$

---

## 5 Extension to Unknown Transition Dynamics

So far we have focused on the special case of known transition dynamics $P$. In this section, we derive a generic reduction from the case of unknown $P$ to known $P$, which may be of independent interest. We prove that *any* algorithm that enjoys a Bayesian regret bound in the known-$P$ special case can be easily converted into one that enjoys a regret bound for the more challenging unknown-$P$ case. Our analysis relies on the mean rewards being sub-Gaussian under the posterior (which holds *e.g.*, under Assumption 1) and on the following standard assumption [61, 47, 62].

**Assumption 2.** *The transition functions are independent Dirichlet under $\phi$, with parameter $\alpha_l(s, a) \in \mathbb{R}_+^{S_{l+1}}$ for each $(s, a) \in \mathcal{S}_l \times \mathcal{A}$ with $\sum_{s' \in \mathcal{S}_{l+1}} \alpha_l(s, a, s') \geq 1$.*

At a high level, our reduction transfers the uncertainty on the transitions to additional uncertainty on the rewards in the form of carefully defined zero-mean Gaussian noise. It extends the reward perturbation idea of the RLSVI algorithm [62], by leveraging a property of Gaussian-Dirichlet optimism [60] and deriving a new property of Gaussian sub-Gaussian optimism (Appendix G) which allows to bypass the assumption of binary rewards in $\{0, 1\}$ from [62, Asm. 3]. Our reduction implies the following 'dominance' property of the expected $V^\star$ under the transformed and original beliefs.

**Lemma 6.** *Define the mapping $\mathcal{T} : \phi \mapsto \widehat{\phi}$ that transforms the beliefs $\phi$ into a distribution $\widehat{\phi}$ on the same space, with transition dynamics equal to $\mathbb{E}_\phi P$ and rewards distributed as $\mathcal{N}(\mathbb{E}_\phi r, \widehat{\sigma}^2)$ with*

$$\widehat{\sigma}_l^2(s, a) := 3.6^2 \sigma_l^2(s, a) + \frac{(L - l)^2}{\sum_{s'} \alpha_l(s, a, s')}.$$

| | Original beliefs $\phi$ | Transformed beliefs $\widehat{\phi}$ |
|---|---|---|
| Mean reward $r$ | $\sigma$-sub-Gaussian | $\mathcal{N}\left(\mathbb{E}_\phi r, \widehat{\sigma}^2\right)$ |
| Transitions $P$ | Dirichlet$(\alpha)$ | $\mathbb{E}_\phi P$ |

*Then it holds that*

$$\mathbb{E}_{s \sim \rho} \mathbb{E}_\phi V_1^\star(s) \leq \mathbb{E}_{s \sim \rho} \mathbb{E}_{\widehat{\phi}} V_1^\star(s).$$

Now, let us return to the episodic interaction case and denote by $\phi^t := \phi(\cdot \mid \mathcal{F}_t)$ the posterior at the beginning of episode $t$. Under Assumptions 1 and 2, we can upper bound the uncertainty $\widehat{\sigma}^t$ of the transformed posteriors $\widehat{\phi}^t$ as $(\widehat{\sigma}_l^t)^2(s, a) \leq (3.6^2(\nu^2 + 1) + (L - l)^2)/(n_l^t(s, a) \vee 1)$. This brings us to a key result necessary for a general regret bound.

**Lemma 7.** *Let* alg *be any procedure that maps posterior beliefs to policies, and denote by $\mathcal{BR}_{\mathcal{T}, \phi}(\mathrm{alg}, T)$ the Bayesian regret of* alg *where at each episode $t = 1, \ldots, T$, the policy and regret are computed by replacing $\phi^t$ with $\mathcal{T}(\phi^t)$ (Lemma 6), then under Assumptions 1 and 2,*

$$\mathcal{BR}_\phi(\mathrm{alg}, T) \leq \mathcal{BR}_{\mathcal{T}, \phi}(\mathrm{alg}, T).$$

This tells us that if we have an algorithm with a Bayesian regret bound under known transitions $P$, then we can convert it directly into an algorithm that achieves a regret bound for the case of unknown $P$, simply by increasing the amount of uncertainty in the posterior of the unknown rewards and replacing the unknown $P$ with its mean when executing the algorithm. We now instantiate this generic reduction to `VAPOR` in Algorithm 1. Combining Lemma 7 and Theorem 1 yields the following regret bound.

**Theorem 2.** *Under Assumptions 1 and 2, it holds that $\mathcal{BR}_\phi(\mathtt{VAPOR}, T) \leq \widetilde{O}(L\sqrt{SAT})$.*

The Bayes regret bound in Theorem 2 is within a factor of $\sqrt{L}$ of the known information theoretic lower bound [32, 13], up to constant and log terms. It matches the best known bound for K-learning [47] and Thompson sampling [61], under the same set of standard assumptions (see Appendix C). There exists a complementary line of work deriving minimax-optimal regret bounds in the *frequentist* setting, both in the model-based [3, 41] and model-free [86, 38] cases. We refer to *e.g.*, [21] for discussion on the advantages and disadvantages of the Bayesian and frequentist approaches. Without

claiming superiority of either, it is worth highlighting that compared to frequentist algorithms that tend to be *deterministic* and *non-stationary* (*i.e.*, explicitly dependent on the episode number), `VAPOR` is naturally *stochastic* and *stationary* (*i.e.*, independent of the episode number).

## 6  Connections

**Thompson Sampling.**   Thompson sampling (TS) or posterior sampling for RL (PSRL) [78, 75, 59] first *samples* an environment (*i.e.*, rewards and transition dynamics) from the posterior and then computes the optimal policy for this sample to be executed during the episode. Our principled Bayesian inference approach to RL uncovers an alternative view of TS which

|  | TS / PSRL | VAPOR |
|---|---|---|
| $\mathbb{P}_{\Gamma^\star}$ approx. | Sample MDP $\mathcal{M}$ and compute $\pi^\star_{\mathcal{M}}$ $\Leftrightarrow \lambda^{\text{TS}} \sim \mathbb{P}_{\Gamma^\star}$ $\Rightarrow \mathbb{E}[\lambda^{\text{TS}}] = \mathbb{P}_{\Gamma^\star}$ | Variationally approximate $\lambda^* \approx \mathbb{P}_{\Gamma^\star}$ and compute $\pi^{\lambda^*}$ |
| Exploration mechanism | Stochastic optimism | Explicit optimism + Entropy regularization |

deepens our understanding of this popular algorithm.  The next lemma shows that TS *implicitly* approximates $\mathbb{P}_{\Gamma^\star}$ by sampling, thus tightly connecting it to `VAPOR`'s *explicit* approximation of $\mathbb{P}_{\Gamma^\star}$.

**Lemma 8.** *Let $\lambda^{\text{TS}}$ be the occupancy measure of the* TS *policy, it holds that $\mathbb{E}[\lambda^{\text{TS}}] = \mathbb{P}_{\Gamma^\star}$.*

Plugging this observation into Lemma 4 and retracing the proof of Theorem 2 immediately yields regret bounds for TS matching that of `VAPOR`. The explicit variational approximation of $\mathbb{P}_{\Gamma^\star}$ by `VAPOR` has several advantages over the implicit sampling-based approximation of $\mathbb{P}_{\Gamma^\star}$ by TS:

- Having direct (approximate) access to this probability can be desirable in some practical applications, for example to ensure safety constraints or to allocate budgets.
- TS suffers linear regret in the multi-agent and constrained cases [52], while we expect suitable modifications of `VAPOR` to be able to handle these cases, as suggested in the multi-armed bandit setting [51].
- The `VAPOR` optimization problem can naturally extend to parameterizing the occupancy measure to be in a certain family (*e.g.*, linear in some basis).
- The objective we use in `VAPOR` is *differentiable*, opening the door to differentiable computing architectures such as deep neural network. In contrast, performing TS requires to maintain an explicit model over MDP parameters [59], which becomes prohibitively expensive as the MDP becomes large. To alleviate this computational challenge of TS, some works have focused on cheaply generating approximate posterior samples [57, 55, 62, 54], but it is not yet clear if this approach is better than the variational approximation we propose.

**K-learning.**   K-learning [47] endows the agent with a risk-seeking exponential utility function and enjoys the same regret bound as Theorem 2.  Its update rule is similar to soft Q-learning [19, 25, 44, 22], which emerges from inference under Approximation 1 [37].  Although soft Q-learning ignores epistemic uncertainty and thus suffers linear regret [53], K-learning *learns* a *scalar* temperature parameter which balances both an optimistic reward and a policy entropy regularization term. In contrast, `VAPOR` naturally produces a *separate* risk-seeking parameter for *each state-action*. This enables a fine-grained control of the exploration-exploitation trade-off depending on the region of the state-action space, which can make a large difference in practice. In the multi-armed bandit case it was shown that K-learning and `VAPOR` coincide when `VAPOR` is constrained to have all temperature parameters (denoted $\tau$) equal [51]. It can be shown (see Appendix F.12) that the same observation holds in the more general MDP case, up to additional entropic terms added to the K-learning 'soft' value functions, which only contribute a small constant to the regret analysis. This sheds a new light on K-learning as a variational approximation of our probabilistic inference framework with the additional constraint that all 'temperature' parameters are equal.

**Maximum Entropy Exploration.**   A recent line of work called *maximum entropy exploration* suggests that in the absence of reward the agent should cover the state-action space as uniformly as possible by solving a maximum-entropy problem over occupancy measures $\max_{\lambda \in \Lambda(P)} \mathcal{H}(\lambda)$ [29, 9, 85, 79], where $\mathcal{H}$ denotes the entropy summed over steps $l$. `VAPOR` can thus be interpreted as regularizing a reward-driven objective with a maximum entropy exploration term, that is weighted with vanishing temperatures. In other words, the principle of (weighted) maximum-entropy exploration can be derived by considering a variational approximation to Bayesian state-action optimality.

# 7 A Policy-Gradient Approximation

Up to this point, we instantiated our new Bayesian 'RL as inference' approach with a tabular, model-based algorithm that exactly solves the variational optimization problem and has a guaranteed regret bound (Algorithm 1). In this section, we derive a principled, albeit looser, upper-bound approximation of the variational optimization problem, which can be solved using policy-gradient techniques by representing the policy using a deep neural network. This will yield a scalable, model-free algorithm that we call `VAPOR-lite`.

Denoting by $\mu^\pi$ the stationary state distribution of a policy $\pi$, note the relation $\lambda_l^\pi(s,a) = \pi_l(s,a)\mu_l^\pi(s)$. The regularization term of `VAPOR` can thus be decomposed in a (weighted) policy entropy regularization term and a (weighted) stationary state distribution entropy term, the latter of which is challenging to optimize in high-dimensional spaces [29, 36, 43]. In light of this, we introduce the following `VAPOR-lite` alternative optimization problem

$$\max_{\pi \in \Pi} \quad \sum_{l,s} \mu_l^\pi(s) \Big( \sum_a \pi_l(s,a) \big( r_l(s,a) + \widehat{\sigma}_l(s,a) + \mathcal{H}_{\widehat{\sigma}_l(s,\cdot)}(\pi_l(s,\cdot)) \big) \Big), \tag{5}$$

where we define the weighted policy entropy $\mathcal{H}_{\widehat{\sigma}_l(s,\cdot)}(\pi_l(s,\cdot)) := -\sum_a \widehat{\sigma}_l(s,a)\pi_l(s,a)\log\pi_l(s,a)$. Akin to policy-gradient methods, `VAPOR-lite` now optimizes over the policies $\pi$ which can be parametrized by a neural network. It depends on an uncertainty signal $\widehat{\sigma}$ which can also be parametrized by a neural network (*e.g.*, an ensemble of reward predictors, as we consider in our experiments in Section 8). Compared to `VAPOR`, `VAPOR-lite` accounts for a *weaker* notion of weighted entropy regularization (*i.e.*, in the policy space instead of the occupancy-measure space), which allows to solve the objective using policy-gradient techniques. For simplicity it also sets the temperatures $\tau$ to the uncertainties $\widehat{\sigma}$ instead of minimizing for the optimal ones. Importantly, `VAPOR-lite` remains a principled approach. We indeed prove the following relevant properties of `VAPOR-lite` in Appendix H: **(i)** although the problem is no longer concave in $\pi$, solving it in the space of occupancy measures $\lambda$ remains a computationally tractable, concave optimization problem, **(ii)** `VAPOR-lite` upper bounds the `VAPOR` objective (up to a multiplicative factor in the uncertainty measure and a negligible additive bias), and **(iii)** it yields the same $\widetilde{O}(L\sqrt{SAT})$ regret bound as `VAPOR` for a careful schedule of uncertainty measures $\widehat{\sigma}$ over episodes.

Essentially, `VAPOR-lite` endows a policy-gradient agent with **(i)** an uncertainty reward bonus and **(ii)** an uncertainty-weighted policy entropy regularization. The latter exploration mechanism is novel and has an appealing interpretation: Unlike standard policy entropy regularization, it adaptively accounts for epistemic uncertainty, by eliminating actions from the entropy term where the agent has low uncertainty for each given state. In the limit of $\widehat{\sigma} \to 0$ (*i.e.*, no uncertainty), the regularization of `VAPOR-lite` vanishes and we recover the original policy-gradient objective.

# 8 Numerical Experiments

**GridWorld.** We first study empirically how well `VAPOR` approximates $\mathbb{P}_{\Gamma^\star}$. Since $\mathbb{E}[\lambda^{\mathrm{TS}}] = \mathbb{P}_{\Gamma^\star}$, we estimate the latter by averaging over 1000 samples of the (random) TS occupancy measure (we denote it by $\overline{\lambda}_{(1000)}^{\mathrm{TS}}$). We design simple $10 \times 10$ GridWorld MDPs with four cardinal actions, known dynamics and randomly generated reward. Figure 2 suggests that `VAPOR` and the TS average output similar approximations of $\mathbb{P}_{\Gamma^\star}$, thus showing the accuracy of our variational approximation in this domain. Unlike TS which requires estimating it with many samples, `VAPOR` approximates $\mathbb{P}_{\Gamma^\star}$ by only solving a single convex optimization problem.

**DeepSea.** We now study the learning performance of the `VAPOR` Algorithm 1 in the hard-exploration environment of *DeepSea* [62]. In this $L \times L$ grid MDP, the agent starts at the top-left cell and must reach the lower-right cell. It can move left or right, always descending to the row below. Going left yields no reward, while going right incurs a small cost of $0.01/L$. The bottom-right cell yields a reward of $1$, so that the optimal policy is to always go right. Local dithering (*e.g.*, Soft Q-learning, $\epsilon$-greedy) takes time exponential in $L$, so the agent must perform deep exploration to reach the goal. Figure 3 shows the time required to 'solve' the problem as a function of the depth $L$, averaged over 10 seeds. The 'time to solve' is defined as the first episode where the rewarding state has been found at least in 10% of the episodes so far [47]. We compare `VAPOR` to TS / PSRL [59], K-learning [47] and a variant of RLSVI [62] that runs PSRL under the transformed posteriors of Lemma 6. (Several

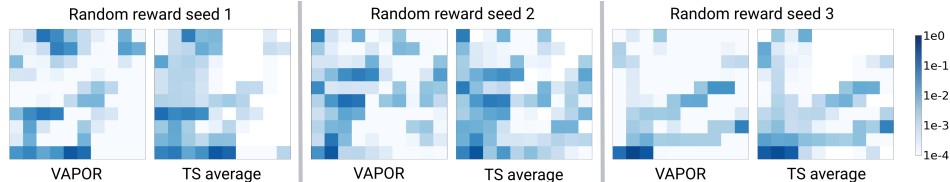

Figure 2: For 3 seeds of randomly generated reward means and noises in a GridWorld, visualization of the timestep-averaged stationary state distribution (*i.e.*, $L^{-1} \sum_{l,a} \lambda_l(s,a)$) for $\lambda^{\texttt{VAPOR}}$ *(left)* and $\overline{\lambda}^{\text{TS}}_{(1000)}$ *(right)*.

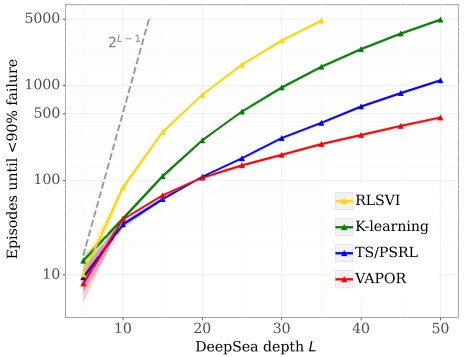

Figure 3: Learning time on DeepSea

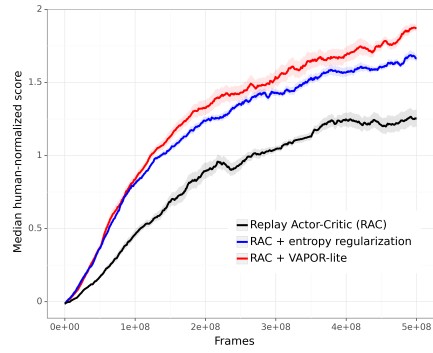

Figure 4: Median human normalized score across 57 Atari games (with standard errors across 5 seeds).

optimistic methods like UCBVI [3] or Optimistic Q-learning [32] were additionally compared by [47] but they performed worse.) We see that `VAPOR` achieves the lowest learning time as $L$ increases, thus displaying its ability to perform deep exploration.

**Atari.** Finally, we investigate the performance of `VAPOR-lite` on the Atari benchmark [5]. We consider a replay-based actor-critic agent with V-trace off-policy corrections [15] and an actor-learner decomposition [30] (see Appendix I for the full experimental details). We compare this agent without and with added *fixed* policy entropy regularization as commonly used [83, 42], to the same agent with the `VAPOR-lite` objective. We stress that unlike simply adding entropy regularization, which leads to local dithering and suffers linear regret, `VAPOR-lite` relies on a principled approach of casting RL as Bayesian inference. Figure 4 shows the performance advantage of augmenting a policy-gradient agent with `VAPOR-lite` on Atari. It reaches the peak performance of the replay actor-critic agent (resp. with entropy regularization) in about $2\times$ (resp. $1.5\times$) fewer environment frames, for essentially the same computational cost. This advantage comes from the fact that `VAPOR-lite` leads to deep exploration, which results in finding higher rewarding states and in better cumulative performance. In our ablation in Appendix I, we illustrate the isolated benefits of tuning entropy regularization on a per state-action basis with `VAPOR-lite` compared to tuning a single scalar for entropy regularization [49]. Our promising empirical results suggest that further gains could come from deriving a more accurate practical approximation of `VAPOR`.

## 9   Conclusion

RL is a statistical inference problem wrapped in a control problem. In this paper, we demonstrated that a single quantity can be used to handle both aspects of the problem — the posterior probability of each state-action pair being visited under the optimal policy, $\mathbb{P}_{\Gamma^\star}$. We can perform Bayesian inference to compute this, and then use it to generate a control policy that performs efficient exploration. This is a coherent and principled approach to 'RL as inference', rather than the heuristic approaches in prior work which could not compute valid posteriors. Unfortunately, computing $\mathbb{P}_{\Gamma^\star}$ is intractable in practice so we derived a variational approximation that we showed also explores efficiently. We concluded with some numerical experiments showing improved performance in the challenging 'DeepSea' unit test, as well as on the Atari suite. We discuss some limitations of our work and directions for future investigation in Appendix D.

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

# Appendix

## Table of Contents

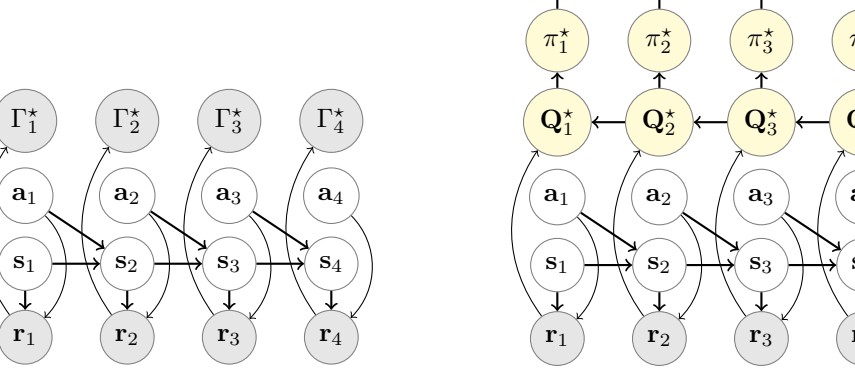

(a) graphical model of the standard
'RL as inference' framework [37]

(b) graphical model for principled
Bayesian 'RL as inference'

Figure 5: **(a)** In the standard 'RL as inference' framework, the binary random variables[2] of state-action optimality $\mathbf{\Gamma}^\star$ are *independent* and *observed*, equal one with probability proportional to the exponentiated reward, i.e., $\mathbb{P}(\mathbf{\Gamma}_l^\star = 1) \propto \exp(\mathbf{r}_l)$. This assumption is arbitrary and ignores the role of uncertainty and exploration. **(b)** In our model, only rewards $\mathbf{r}$ are observed, with *unobserved* binary optimality variables $\mathbf{\Gamma}^\star$ and unobserved optimal values $\mathbf{Q}^\star$. $\mathbf{Q}_l^\star$ depends on current reward $\mathbf{r}_l$ and future $\mathbf{Q}_{l+1}^\star$, and determines action optimality $\pi_l^\star$. $\mathbf{\Gamma}_l^\star$ depends on prior state-action optimality $\mathbf{\Gamma}_{l-1}^\star$ and current action optimality $\pi_l^\star$. 'Optimality' thus propagates both in a *backward* way (via $\mathbf{Q}_l^\star$) and a *forward* way (via $\mathbf{\Gamma}_l^\star$). Refraining from any modeling assumption on $\mathbf{\Gamma}^\star$, we derive a variational approximation of the posterior probability of $\mathbf{\Gamma}^\star$ which yields an algorithm that explores efficiently.

## A   Graphical Models

We illustrate in Figure 5 the graphical models representing the standard[1] 'RL as inference' framework [37] and our principled, Bayesian 'RL as inference' framework. For ease of presentation, we consider here a simplified representation where we isolate a single trajectory $(s_1, a_1, \ldots, s_4, a_4)$ and consider that $\Gamma^\star$ does not encompass the randomness in the transition dynamics (which is *e.g.*, the case for deterministic dynamics).

---

[1]There exist other approaches to the 'RL as inference' perspective, such as the VIREL framework [17], that provide alternative benefits yet do not offer any principled guarantees either.

[2]In the main text, we use the terminology of *events* but note that events can be seen as *binary random variables* by taking the indicator function of the event.

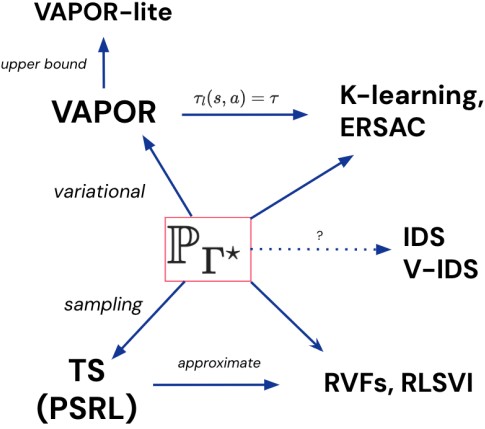

Figure 6: VAPOR is an explicit, variational approximation of $\mathbb{P}_{\Gamma^\star}$, while Thompson sampling (TS) implicitly approximates $\mathbb{P}_{\Gamma^\star}$ by sampling, similarly to the TS approximations Randomized value functions (RVFs), RLSVI [62]. A special case of VAPOR with equal temperatures approximately recovers K-learning and ERSAC [49]. We conjecture that there also exist connections between $\Gamma^\star$ and information-theoretic approaches such as information-directed sampling (IDS), variance-IDS (V-IDS) [69, 28, 39].

## B  Algorithmic Connections to $\Gamma^\star$

Figure 6 illustrates how numerous exploration algorithms can be linked to $\Gamma^\star$. This sheds a new light on existing algorithms, tightly connects them to our variational approach, and unifies these approaches within our principled Bayesian framework.

## C  Assumptions

In this section, we review and discuss the assumptions made throughout the paper. We stress that these assumptions are *not* required *computationally* for our variational Bayesian approach, but only for the *analysis* to obtain VAPOR's regret bound (Theorem 2). These are:

(A)  Layered, time-inhomogoneous MDP,
(B)  The reward noise is additive sub-Gaussian and the mean rewards are bounded almost surely with independent priors,
(C)  The prior over transition functions is independent Dirichlet.

(A) This assumption is non-restrictive in the sense that any finite-horizon MDP with cycles can be transformed into an equivalent MDP without cycles by adding a timestep variable $l \in [L]$ to the state space (picking up an additional factor of $\sqrt{L}$ in the regret bound). It implies the transition function and the value function at the next state are conditionally independent, which is a technical property that simplifies the regret analysis.

(B) The sub-Gaussian assumption is standard in the literature and arises commonly in practice. For instance, it holds in any environment with bounded rewards (e.g., DeepSea, Atari). In Appendix F.3 (see Lemma 11) we extend Lemma 3 to the general case, and we refer to Appendix F.5 for the resulting VAPOR optimization problem (this would give a generic regret bound for the VAPOR learning algorithm which would depend on how fast the posteriors concentrate).

(C) The Dirichlet prior over the transition functions is the canonical prior for transitions in Bayesian RL, see [21]. This is because visiting a state-action pair simply increments by 1 the appropriate entry of the Dirichlet vector — note the ease of the update because the Dirichlet is the conjugate prior of the categorical distribution (which models transition probabilities of discrete-state-action-MDPs).

Note that we make the same assumptions as existing state-of-the-art Bayesian regret analyses, including PSRL [61][3], K-learning [47], RLSVI [62]. To the best of our knowledge, it is an open question how to obtain an algorithm with a $\widetilde{O}(L\sqrt{SAT})$ Bayes regret bound without these assumptions.

---

[3]We point out that [61] originally considers time-homogeneous dynamics but there is a known mistake in the regret analysis, specifically in Lemma 3. We refer to e.g., these tutorial slides (see footnote of slide 65/91) and [67] for details. This issue can be fixed by assuming time-inhomogeneous dynamics.

# D    Limitations

We now discuss some limitations of our work, which constitute relevant directions for future investigation.

- The `VAPOR` convex optimization problem grows with $L, S, A$. Although large-scale exponential cone solvers exist that can handle this type of problem, it does not immediately lend itself to an online approximation.
- The regret guarantee of `VAPOR` relies on Assumptions 1 and 2 which, although standard (Appendix C), can be strong and it would be relevant to (partly) relax them.
- `VAPOR-lite` makes some upper-bound approximations of our variational optimization problem to use policy-gradient techniques, and there may be a tighter way to approximate `VAPOR` in the case where the policy is parameterized using a deep neural network that is updated using online stochastic gradients.

# E    Properties of the `VAPOR` Optimization Problem

The `VAPOR` optimization problem is an exponential cone program that can be solved efficiently using modern optimization methods [50, 12, 74, 46]. In Appendix E.1, we show that its dual is a convex optimization problem that is *unconstrained* in a 'value' variable and a 'temperature' variable. In Appendix E.2, we relate `VAPOR` to *message passing*, a popular technique to perform inference in graphical models [82], by showing that `VAPOR` can be cast as passing some forward and backward messages back and forth.

## E.1    Dual Problem of the `VAPOR` Optimization Problem

We derive the dual problem of the `VAPOR` optimization problem (4) and show that it is a convex optimization problem in a 'value' variable $V \in \mathbb{R}^{L,S} := \{\mathbb{R}^{S_l}\}_{l=1}^L$ and a 'temperature' variable $\tau \in \mathbb{R}_+^{L,S,A}$. Even though the 'primal' admits flow constraints in $\lambda$, this dual is *unconstrained* in $V$ (the only constraint is the non-negativity of $\tau$). In the case of unknown $P$ (Section 5), the dual is the same as in Lemma 9 except that $P$ is replaced by $\mathbb{E}_\phi P$ and the uncertainty $\sigma$ is replaced by $\hat{\sigma}$.

**Lemma 9.** *The $\lambda^* \in \Lambda(P)$ solving the `VAPOR` optimization problem (4) is unique and satisfies*

$$\lambda_l^*(s, a) = \exp\left( \frac{\delta_{V^*, \tau^*, l}(s, a)}{\tau_l(s, a)} - 1 \right),$$

*where we define the 'advantage' function*

$$\delta_{V^*, \tau^*, l}(s, a) = \mathbb{E}_\phi r_l(s, a) + \frac{\sigma_l^2(s, a)}{2\tau_l^*(s, a)} + \sum_{s'} P_l(s' \mid s, a) V_{l+1}^*(s') - V_l^*(s),$$

*and where $V^*, \tau^*$ minimize the (convex) dual function*

$$\min_{V \in \mathbb{R}^{L,S}} \min_{\tau \in \mathbb{R}_+^{L,S,A}} \left\{ \sum_s \rho(s) V_1(s) + \sum_{l,s,a} \tau_l(s, a) \exp\left( \frac{\delta_{V, \tau, l}(s, a)}{\tau_l(s, a)} - 1 \right) \right\}. \tag{6}$$

*The induced policy $\pi^*$ is uniquely defined as*

$$\pi_l^*(s, a) = \frac{\exp\left( \frac{\delta_{V^*, \tau^*, l}(s, a)}{\tau_l^*(s, a)} - 1 \right)}{\sum_{a'} \exp\left( \frac{\delta_{V^*, \tau^*, l}(s, a')}{\tau_l^*(s, a')} - 1 \right)}$$

*if the denominator is positive, otherwise $\pi_l^*(s, \cdot)$ can be any distribution.*

*Proof.* The proof follows from applying the method of Lagrange multipliers to the `VAPOR` optimization problem using the definition of $\mathcal{V}$ in (3). Derivations of dual problems of related optimization problems in the space of occupancy measures can be found in *e.g.*, [65, 24]. Note that for a fixed temperature function $\tau \in \mathbb{R}_+^{L,S,A}$, `VAPOR` is an instance of linear programming with weighted entropic perturbation, which is a class of optimization problems shown to admit an unconstrained convex dual [8, 16]. □

---

**Algorithm 2** Frank-Wolfe algorithm for the `VAPOR` optimization problem (Lemma 10)

---

**Require:** Accuracy level $\varepsilon > 0$

   Set the smoothing parameter $\delta = \varepsilon/(\sigma_{\max}LSA)$

   Define $\mathcal{V}_\delta(\lambda) \coloneqq \sum_{l,s,a} \lambda_l(s,a)\left[\mathbb{E}r_l(s,a) + \sigma_l(s,a)\sqrt{-2(\log(\lambda_l(s,a) + \delta) + \delta)}\right]$

   Set the number of iterations $K = (\sigma_{\max}L\varepsilon^{-1})^5(SA)^4$

   Initialize any $\lambda^{(0)} \in \Lambda$

   **for** iterations $k = 1, 2, \ldots, K$ **do**

      *Backward message pass:* Compute the optimal policy $\pi^{(k-1)}$ for the reward $\nabla\mathcal{V}_\delta(\lambda^{(k-1)})$

      *Forward message pass:* Compute $d^{(k-1)}$ the stationary state distribution induced by $\pi^{(k-1)}$, *i.e.*,

$$d_1^{(k-1)}(s) = \rho(s), \quad d_{l+1}^{(k-1)}(s') = \sum_{s,a} P_l(s' \mid s,a)\pi_l^{(k-1)}(s,a)d_l(s)$$

      *Combine:* Compute for step size $\gamma_k = \frac{2}{k+1}$ the occupancy measure

$$\lambda_l^{(k)}(s,a) = (1 - \gamma_k)\lambda_l^{(k-1)}(s,a) + \gamma_k d_l^{(k-1)}(s)\pi_l^{(k-1)}(s,a)$$

   **end for**

   **return** $\lambda^{(K)}$

---

### E.2 Message Passing View of the `VAPOR` Optimization Problem

Message passing is a popular technique to perform inference in graphical models [82]. In fact, as shown in [37, Section 2.3], the standard 'RL as inference' framework (Section 2.1) can recover the optimal policy using standard sum-product message passing, relying on a single backward message pass. While `VAPOR` does not have such a simplified derivation, the next lemma shows that it can be cast as passing some forward and backward messages back and forth. In this view, a backward message amounts to computing the optimal policy for a given reward (via backward induction), and a forward message amounts to computing the associated stationary state distribution (via the flow equations). This view stems from iteratively solving `VAPOR` using the Frank-Wolfe (a.k.a. conditional gradient) algorithm, where an update is equivalent to solving an MDP [29, 77, 85].

**Lemma 10.** *For any accuracy $\varepsilon > 0$, there exist forward-backward messages such that after* $\mathrm{poly}(S, A, L, \sigma_{\max}, \varepsilon^{-1})$ *passes they output a* $\lambda^{MP} \in \Lambda(P)$ *with* $\max_{\lambda\in\Lambda(P)} \mathcal{V}_\phi(\lambda) - \mathcal{V}_\phi(\lambda^{MP}) \leq \varepsilon$.

*Proof.* We consider in Algorithm 2 the Frank-Wolfe algorithm applied to maximizing the smoothed proxy $V_\delta$ over $\lambda \in \Lambda$ (throughout the proof we write $\Lambda = \Lambda(P)$ and omit the dependence on the beliefs $\phi$ for ease of notation). Each iteration represents one forward-backward message pass: the backward message computes the optimal policy for the reward given by the gradient of the objective $V_\delta$ evaluated at the current iterate (via backward induction), while the forward message computes the associated stationary state distribution (via the flow equations). Note that the chosen number of iterations $K$ is polynomial in the inverse of the accuracy level $\varepsilon^{-1}$, the maximum uncertainty $\sigma_{\max} \coloneqq \max_{l,s,a} \sigma_l(s,a)$ and the MDP parameters $L, S, A$. To prove the $\varepsilon$-convergence in the original objective $\mathcal{V}$, we decompose

$$\mathcal{V}(\lambda^*) - \mathcal{V}(\lambda^{(K)}) = \mathcal{V}(\lambda^*) - \max_{\lambda\in\Lambda}V_\delta(\lambda) + \max_{\lambda\in\Lambda}V_\delta(\lambda) - \mathcal{V}_\delta(\lambda^{(K)}) + \mathcal{V}_\delta(\lambda^{(K)}) - \mathcal{V}(\lambda^{(K)})$$

$$\leq \underbrace{\mathcal{V}(\lambda^*) - V_\delta(\lambda^*)}_{①} + \underbrace{\max_{\lambda\in\Lambda}V_\delta(\lambda) - \mathcal{V}_\delta(\lambda^{(K)})}_{②} + \underbrace{\mathcal{V}_\delta(\lambda^{(K)}) - \mathcal{V}(\lambda^{(K)})}_{③}.$$

① and ③ can be bounded using the fact that for any $\lambda \in \Lambda$, $| \mathcal{V}_\delta(\lambda) - \mathcal{V}(\lambda) | \leq \sigma_{\max}LSA\delta$. ② can be bounded from the Frank-Wolfe convergence rate for smooth objective functions [31], since the proxy $\mathcal{V}_\delta$ is $C_\delta$-smooth by construction, with $C_\delta = O(\sigma_{\max}\delta^{-4})$. As a result, ② $\leq M_\delta/(K+2)$, where $M_\delta$ is the curvature constant of $\mathcal{V}_\delta$ over $\Lambda$, with $M_\delta \leq C_\delta \operatorname{diam}^2(\Lambda)$, where $\operatorname{diam}^2(\Lambda) \leq 2L$. Therefore, choosing $\delta = \varepsilon/(\sigma_{\max}LSA)$ and $K = (\sigma_{\max}L\varepsilon^{-1})^5(SA)^4$ gives

$$\mathcal{V}(\lambda^*) - \mathcal{V}(\lambda^{(K)}) \leq O\left(\sigma_{\max}LSA\delta + \frac{\sigma_{\max}L}{\delta^4 K}\right) = O(\varepsilon).$$

□

# F  Proofs

## F.1  Proof of Lemma 1

**Lemma 1.** *For known $P$, it holds that* $\mathbb{P}_{\Gamma^\star} := \{\mathbb{P}_\phi\left(\Gamma_l^\star\right)\}_{l=1}^L \in \Lambda(P)$.

*Proof.* We omit the dependence on the beliefs $\phi$ for ease of notation. At the initial step,

$$\mathbb{P}\left(\Gamma_1^\star(s,a)\right) = \mathbb{P}\left(\{s_1 = s\} \cap \{\pi_1^\star(s) = a\}\right) = \rho(s)\mathbb{P}\left(\pi_1^\star(s) = a\right),$$

and from Bayes' rule and the fact that $P$ is assumed known, we have the recursive equation

$$
\begin{aligned}
\mathbb{P}\left(\Gamma_{l+1}^\star(s',a')\right) &= \mathbb{P}\left(\Gamma_{l+1}^\star(s') \cap \{\pi_{l+1}^\star(s') = a'\}\right) \\
&= \mathbb{P}\left(\pi_{l+1}^\star(s') = a' \mid \Gamma_{l+1}^\star(s')\right) \mathbb{P}\left(\Gamma_{l+1}^\star(s')\right) \\
&= \mathbb{P}\left(\pi_{l+1}^\star(s') = a' \mid \Gamma_{l+1}^\star(s')\right) \sum_{s,a} \mathbb{P}\left(\Gamma_l^\star(s,a) \cap \{s_{l+1} = s'\}\right) \\
&= \mathbb{P}\left(\pi_{l+1}^\star(s') = a' \mid \Gamma_{l+1}^\star(s')\right) \sum_{s,a} \mathbb{P}\left(\{s_{l+1} = s'\} \mid \Gamma_l^\star(s,a)\right) \mathbb{P}\left(\Gamma_l^\star(s,a)\right) \\
&= \mathbb{P}\left(\pi_{l+1}^\star(s') = a' \mid \Gamma_{l+1}^\star(s')\right) \sum_{s,a} P_l(s' \mid s,a)\mathbb{P}\left(\Gamma_l^\star(s,a)\right). \quad (7)
\end{aligned}
$$

Summing (7) over $a' \in \mathcal{A}$ implies the non-negativity and flow conservation properties of $\mathbb{P}_{\Gamma^\star}$. Moreover, when $\sum_{a' \in \mathcal{A}} \mathbb{P}\left(\Gamma_l^\star(s,a')\right) > 0$, it holds that

$$\mathbb{P}\left(\pi_l^\star(s) = a \mid \Gamma_l^\star(s)\right) = \frac{\mathbb{P}\left(\Gamma_l^\star(s,a)\right)}{\sum_{a' \in \mathcal{A}} \mathbb{P}\left(\Gamma_l^\star(s,a')\right)},$$

which corresponds to (1). $\qquad\square$

## F.2  Proof of Lemma 2

**Lemma 2.** *It holds that*

$$\mathbb{E}_{s\sim\rho}\mathbb{E}_\phi V_1^\star(s) = \sum_{l,s,a} \mathbb{P}_\phi(\Gamma_l^\star(s,a))\mathbb{E}_\phi\left[r_l(s,a) \mid \Gamma_l^\star(s,a)\right].$$

*Proof.* Denote by $Z^\star$ the random variable of the total reward accumulated along a single rollout of the optimal policy $\pi^\star$, and let $Z_l^\star(s)$ be the total reward accumulated by $\pi^\star$ from state $s$ at layer $l$ and $Z_l^\star(s,a)$ be the total reward accumulated by $\pi^\star$ from state $s$ at layer $l$ after taking action $a$. We can write the total reward accumulated as

$$Z^\star = \sum_s Z_1^\star(s)\mathbb{1}\{s_1 = s\} = \sum_s Z_1^\star(s)\mathbb{1}\{\Gamma_1^\star(s)\}.$$

Now for arbitrary $l \in \{1, \ldots, L-1\}$,

$$
\begin{aligned}
\sum_s Z_l^\star(s)\mathbb{1}\{\Gamma_l^\star(s)\} &= \sum_{s,a} Z_l^\star(s,a)\mathbb{1}\{\pi_l^\star(s) = a\}\mathbb{1}\{\Gamma_l^\star(s)\} \\
&= \sum_{s,a} Z_l^\star(s,a)\mathbb{1}\{\Gamma_l^\star(s,a)\} \\
&= \sum_{s,a} R_l(s,a)\mathbb{1}\{\Gamma_l^\star(s,a)\} + \sum_{s,a} \mathbb{1}\{\Gamma_l^\star(s,a)\} \sum_{s'} Z_{l+1}^\star(s')\mathbb{1}\{s_{l+1} = s'\} \\
&= \sum_{s,a} R_l(s,a)\mathbb{1}\{\Gamma_l^\star(s,a)\} + \sum_{s'} Z_{l+1}^\star(s')\mathbb{1}\{\Gamma_{l+1}^\star(s')\},
\end{aligned}
$$

and unrolling we obtain

$$Z^\star = \sum_{l,s,a} R_l(s,a)\mathbb{1}\{\Gamma_l^\star(s,a)\}.$$

Denoting by $\mathcal{F}_r$ the sigma-algebra generated by $r$, it holds from the tower property of conditional expectation that

$$
\begin{aligned}
\mathbb{E}_\phi[R_l(s,a)\mathbb{1}\{\Gamma_l^\star(s,a)\}] &= \mathbb{E}_\phi[\mathbb{E}_\phi[R_l(s,a)\mathbb{1}\{\Gamma_l^\star(s,a)\} \mid \mathcal{F}_r]] \\
&= \mathbb{E}_\phi[\mathbb{E}_\phi[R_l(s,a) \mid \mathcal{F}_r]\mathbb{E}_\phi[\mathbb{1}\{\Gamma_l^\star(s,a)\} \mid \mathcal{F}_r]] \\
&= \mathbb{E}_\phi[r_l(s,a)\mathbb{1}\{\Gamma_l^\star(s,a)\}].
\end{aligned}
$$

Therefore it holds that

$$
\mathbb{E}_{s\sim\rho}\mathbb{E}_\phi V^\star(s) = \sum_{l,s,a} \mathbb{E}_\phi\left[r_l(s,a)\mathbb{1}\{\Gamma_l^\star(s,a)\}\right].
$$

Now note that

$$
\begin{aligned}
\mathbb{E}_\phi\left[r_l(s,a)\mathbb{1}\{\Gamma_l^\star(s,a)\}\right] &= \int_\mathbb{R} r\mathbb{P}_\phi(r_l(s,a) = r, \Gamma_l^\star(s,a)) \\
&= \int_\mathbb{R} r\mathbb{P}_\phi(r_l(s,a) = r \mid \Gamma_l^\star(s,a))\mathbb{P}_\phi(\Gamma_l^\star(s,a)) \\
&= \mathbb{E}_\phi\left[r_l(s,a) \mid \Gamma_l^\star(s,a)\right]\mathbb{P}_\phi(\Gamma_l^\star(s,a)).
\end{aligned}
$$

Putting it all together,

$$
\mathbb{E}_{s\sim\rho}\mathbb{E}_\phi V^\star(s) = \sum_{l,s,a} \mathbb{P}_\phi(\Gamma_l^\star(s,a))\mathbb{E}_\phi\left[r_l(s,a) \mid \Gamma_l^\star(s,a)\right].
$$

$\square$

### F.3 Proof of Lemma 3

We prove in Lemma 11 a generalization of Lemma 3 that does not rely on the assumption of $\sigma$-sub-Gaussian mean rewards. It builds on the Donsker-Varadhan representation [23, 70, 69, 28] (Proposition 1), see also [51, Theorem 1]. We denote by $\mathrm{KL}(\cdot \,||\, \cdot)$ the Kullback–Leibler divergence, by $\Psi_X : \mathbb{R} \to \mathbb{R}$ the cumulant generating function of random variable $X - \mathbb{E}X$ and by $f^\star : \mathbb{R} \to \mathbb{R}$ the convex conjugate of function $f : \mathbb{R} \to \mathbb{R}$, *i.e.*,

$$
\Psi_X(\beta) := \log\mathbb{E}\exp(\beta(X - \mathbb{E}X)), \qquad f^\star(y) := \sup_{x\in\mathbb{R}}\{xy - f(x)\}.
$$

**Lemma 11.** *Let $X : \Omega \to \mathbb{R}$ be a random variable on $(\Omega, \mathcal{F}, \mathbb{P})$ satisfying $X \in L_1$ such that the interior of the domain of $\Psi_X$ is non-empty, and let $B \in \mathcal{F}$ be an event with $\mathbb{P}(B) > 0$. Then,*[4]

$$
\begin{aligned}
\mathbb{E}\left[X \mid B\right] &\le \mathbb{E}X + (\Psi_X^*)^{-1}\left(\mathrm{KL}(\mathbb{P}(X \in \cdot \mid B) \,||\, \mathbb{P}(X \in \cdot))\right) \\
&\le \mathbb{E}X + (\Psi_X^*)^{-1}\left(-\log\mathbb{P}(B)\right).
\end{aligned}
$$

*Proof.* Applying Proposition 1 to $\lambda(X - \mathbb{E}X)$ for arbitrary $\lambda \in \mathbb{R}_+$ with the choices $Q := \mathbb{P}(X \in \cdot)$, $P := \mathbb{P}(X \in \cdot \mid B)$ gives

$$
\begin{aligned}
\mathrm{KL}(\mathbb{P}(X \in \cdot \mid B) \,||\, \mathbb{P}(X \in \cdot)) &\ge \sup_{\lambda\in\mathbb{R}_+}\left\{\lambda\mathbb{E}[(X - \mathbb{E}X) \mid B] - \log\mathbb{E}[\exp\lambda(X - \mathbb{E}X)]\right\} \\
&= \sup_{\lambda\in\mathbb{R}_+}\left\{\lambda(\mathbb{E}[X \mid B] - \mathbb{E}X) - \Psi_X(\lambda)\right\} \\
&= \Psi_X^\star(\mathbb{E}[X \mid B] - \mathbb{E}X),
\end{aligned}
$$

by definition of $\Psi_X$ the cumulant generating function of $X - \mathbb{E}X$ and by definition of the convex conjugate of $\Psi_X$ which satisfies $\Psi_X^\star(y) = \sup_{\lambda\ge0}\{\lambda y - \Psi_X(\lambda)\}$. Since $\Psi_X^\star$ is strictly increasing, its inverse $(\Psi_X^*)^{-1}$ is also strictly increasing, so

$$
\mathbb{E}[X \mid B] - \mathbb{E}X \le (\Psi_X^*)^{-1}\left(\mathrm{KL}(\mathbb{P}(X \in \cdot \mid B) \,||\, \mathbb{P}(X \in \cdot))\right),
$$

which proves the first inequality. We now derive the second inequality. Note the definition of the KL divergence

$$
\mathrm{KL}(P \,||\, Q) = \int \log\left(\frac{dP}{dQ}\right) dP,
$$

---

[4]If $\Psi_X = 0$, then we take $(\Psi_X^*)^{-1} = 0$.

where $\frac{dP}{dQ}$ is the Radon-Nikodym derivative of $P$ with respect to $Q$. We can then write

$$
\begin{aligned}
\mathrm{KL}(\mathbb{P}(X \in \cdot \mid B) \,||\, \mathbb{P}(X \in \cdot)) &= \int \log \frac{d\mathbb{P}(X \in \cdot \mid B)}{d\mathbb{P}(X \in \cdot)} d\mathbb{P}(X \in \cdot \mid B) \\
&\overset{(i)}{=} \int \log \frac{\mathbb{P}(X \in \cdot \mid B)}{\mathbb{P}(X \in \cdot)} d\mathbb{P}(X \in \cdot \mid B) \\
&\overset{(ii)}{=} \int \log \frac{\mathbb{P}(B \mid X \in \cdot)}{\mathbb{P}(B)} d\mathbb{P}(X \in \cdot \mid B) \\
&= \int \underbrace{\log \mathbb{P}(B \mid X \in \cdot)}_{\leq 0} d\mathbb{P}(X \in \cdot \mid B) - \log \mathbb{P}(B) \\
&\leq -\log \mathbb{P}(B),
\end{aligned}
$$

where (i) is from the formulation of Bayes' theorem stated in [73, Theorem 1.31] and (ii) applies Bayes' rule. $\qquad \square$

**Proposition 1** (Variational form of the KL-Divergence given in Theorem 5.2.1 of [23])**.** *Fix two probability distributions $P$ and $Q$ such that $P$ is absolutely continuous with respect to $Q$. Then,*

$$
\mathrm{KL}(P \,||\, Q) = \sup_X \left\{ \mathbb{E}_P[X] - \log \mathbb{E}_Q[\exp X] \right\},
$$

*where $\mathbb{E}_P$ and $\mathbb{E}_Q$ denote the expectation operator under $P$ and $Q$ respectively, and the supremum is taken over all real valued random variables $X$ such that $\mathbb{E}_P[X]$ is well defined and $\mathbb{E}_Q[\exp X] < \infty$.*

*Proof of Lemma 3.* Under the $\sigma_l(s,a)$-sub-Gaussian assumption of $r_l(s,a)$, we can bound the cumulant generating function of $r_l(s,a)$ as

$$
\Psi_{r_l(s,a)}(\beta) \leq \frac{\beta^2 \sigma_l(s,a)^2}{2}.
$$

We therefore have that for any $y \geq 0$,

$$
(\Psi^*_{r_l(s,a)})^{-1}(y) = \inf_{\tau_l(s,a) > 0} \left\{ \tau_l(s,a) \Psi_{r_l(s,a)}(1/\tau_l(s,a)) + \tau_l(s,a)y \right\} \leq \sigma_l(s,a)\sqrt{2y}. \quad (8)
$$

Plugging this bound into Lemma 11 yields Lemma 3. $\qquad \square$

### F.4 Proof of Lemma 4

**Lemma 4.** *For known $P$ and $\sigma$-sub-Gaussian $r$, we have*

$$
\mathbb{E}_{s \sim \rho} \mathbb{E}_\phi V_1^\star(s) \leq \mathcal{V}_\phi(\mathbb{P}_{\Gamma^\star}) \leq \underbrace{\max_{\lambda \in \Lambda(P)} \mathcal{V}_\phi(\lambda)}_{\text{VAPOR}}, \quad (4)
$$

*where the* `VAPOR` *optimization problem is concave.*

*Proof.* The first inequality in (4) combines Lemmas 2 and 3. The second inequality comes from the fact that $\mathbb{P}_{\Gamma^\star} \in \Lambda(P)$ from Lemma 1. The `VAPOR` optimization problem is concave since (i) the objective function $\mathcal{V}$ is concave in $\lambda$ by concavity of the function $x \in (0,1) \to x\sqrt{-\log x}$, and (ii) the constraints are a convex set due to the fact that the state-action polytope $\Lambda(P)$ is closed, bounded and convex [66, Theorem 8.9.4]. Finally, note that the maximum exists and is attained, since $\Lambda(P)$ is closed, bounded and nonempty due to the fact that $\mathbb{P}_{\Gamma^\star} \in \Lambda(P)$. $\qquad \square$

### F.5 `VAPOR` without the Sub-Gaussian Assumption

From (8), for *any* set of reward beliefs, the `VAPOR` optimization problem (Lemma 4) becomes

$$
\max_{\lambda \in \Lambda(P)} \lambda^\top \left( \mathbb{E}_\phi r + (\Psi_r^*)^{-1}(-\log \lambda) \right) = \sum_{l,s,a} \lambda_l(s,a) \left( \mathbb{E}_\phi r_l(s,a) + (\Psi^*_{r_l(s,a)})^{-1}(-\log \lambda_l(s,a)) \right).
$$

This objective remains a concave optimization problem since the function $(\Psi^*_{r_l(s,a)})^{-1}$ is always concave (as the inverse of the strictly increasing convex function $\Psi_{r_l(s,a)}$).

### F.6 Proof of Lemma 5

**Lemma 5.** *Let $\lambda^*$ solve the* `VAPOR` *optimization problem* (4) *and* $\tau^* \in \operatorname{argmin}_\tau \mathcal{V}_\phi(\lambda^*, \tau)$, *then*

$$\mathrm{KL}_{\tau^*}(\mathbb{P}_{\Gamma^\star} \| \lambda^*) \leq \mathcal{V}_\phi(\lambda^*) - \mathbb{E}_{s\sim\rho}\mathbb{E}_\phi V_1^\star(s).$$

*Proof.* The proof extends the relationship between Bregman divergence and value in policy-regularized MDPs [48] to the setting of occupancy-measure-regularized MDPs. The Bregman divergence generated by $\Omega$ between two points $\lambda, \lambda' \in \Lambda(P)$ is defined as

$$D_\Omega(\lambda, \lambda') := \Omega(\lambda) - \Omega(\lambda') - \nabla\Omega(\lambda')^\top(\lambda - \lambda').$$

For ease of notation, we omit the dependence on the beliefs $\phi$ throughout the proof. We write

$$\mathcal{V}(\lambda) := \sum_{l=1}^{L} \sum_{s,a} \lambda_l(s,a)\mathbb{E}_\phi r_l(s,a) - \Omega(\lambda), \qquad \mathcal{V}^* := \max_{\lambda \in \Lambda(P)} \mathcal{V}(\lambda), \qquad \lambda^* \in \operatorname*{argmax}_{\lambda \in \Lambda(P)} \mathcal{V}(\lambda),$$

where $\Omega : \Lambda(P) \to \mathbb{R}$ is a continuously differentiable strictly convex regularizer defined as

$$\Omega(\lambda) := -\sum_{l=1}^{L} \sum_{s,a} \sigma_l(s,a)\lambda_l(s,a)\sqrt{-2\log\lambda_l(s,a)}.$$

By definition, $\lambda^*$ must satisfy the first-order optimality conditions for the maximum of $\mathcal{V}$, and since $\lambda^* \in \operatorname{relint}(\Lambda(P))$ the following holds

$$(\mathbb{E}_\phi r - \nabla\Omega(\lambda^*))^\top(\lambda - \lambda^*) = 0, \quad \forall\lambda \in \Lambda(P).$$

Then,

$$\begin{aligned}
D_\Omega(\mathbb{P}_{\Gamma^\star}, \lambda^*) &= \Omega(\mathbb{P}_{\Gamma^\star}) - \Omega(\lambda^*) - \nabla\Omega(\lambda^*)^\top(\mathbb{P}_{\Gamma^\star} - \lambda^*) \\
&= \Omega(\mathbb{P}_{\Gamma^\star}) - \Omega(\lambda^*) - \mathbb{E}_\phi r^\top(\mathbb{P}_{\Gamma^\star} - \lambda^*) \\
&= \mathcal{V}^* - \mathbb{E}_\phi r^\top \mathbb{P}_{\Gamma^\star} + \Omega(\mathbb{P}_{\Gamma^\star}) \\
&= \mathcal{V}^* - \mathcal{V}(\mathbb{P}_{\Gamma^\star}).
\end{aligned} \tag{9}$$

We now notice that

$$\Omega(\lambda) = -\sum_{l=1}^{L} \sum_{s,a} \min_{\tau_l(s,a)>0} \left\{ \frac{\sigma_l^2(s,a)}{2\tau_l(s,a)} - \tau_l(s,a)\lambda_l(s,a)\log\lambda_l(s,a) \right\}.$$

For any positive $\tau \in \mathbb{R}_+^{L,S,A}$, we define

$$\Omega_\tau(\lambda) := \sum_{l=1}^{L} \sum_{s,a} -\frac{\sigma_l^2(s,a)}{2\tau_l(s,a)} + \tau_l(s,a)\lambda_l(s,a)\log\lambda_l(s,a)$$

$$\mathcal{V}_\tau(\lambda) := \sum_{l=1}^{L} \sum_{s,a} \lambda_l(s,a)\mathbb{E}_\phi r_l(s,a) - \Omega_\tau(\lambda),$$

$$\mathcal{V}_\tau^* := \max_{\lambda \in \Lambda(P)} \mathcal{V}_\tau(\lambda), \quad \lambda_\tau^* \in \operatorname*{argmax}_{\lambda \in \Lambda(P)} \mathcal{V}_\tau(\lambda).$$

Using that the KL-divergence is the Bregman divergence generated by the negative-entropy function, (9) implies that

$$\mathrm{KL}_\tau(\mathbb{P}_{\Gamma^\star} \| \lambda_\tau^*) = \mathcal{V}_\tau^* - \mathcal{V}_\tau(\mathbb{P}_{\Gamma^\star}).$$

Moreover, from strong duality (which holds because $\Lambda(P)$ is convex),

$$\min_{\tau \in \mathbb{R}_+^{L,S,A}} \max_{\lambda \in \Lambda(P)} \mathcal{V}_\tau(\lambda) = \max_{\lambda \in \Lambda(P)} \min_{\tau \in \mathbb{R}_+^{L,S,A}} \mathcal{V}_\tau(\lambda).$$

Denote $\tau^* \in \operatorname{argmin}_\tau \mathcal{V}_\tau^*$, then it holds that

$$\mathrm{KL}_{\tau^*}(\mathbb{P}_{\Gamma^\star} \| \lambda^*) = \mathcal{V}^* - \mathcal{V}_{\tau^*}(\mathbb{P}_{\Gamma^\star}) \leq \mathcal{V}^* - \mathcal{V}(\mathbb{P}_{\Gamma^\star}),$$

which concludes the proof. $\square$

### F.7 Proof of Theorem 1 and Corollary 1

Below we prove a more general statement that will imply Theorem 1 and Corollary 1.

**Lemma 12.** *Let* alg *produce any sequence of $\mathcal{F}_t$-measurable policies $\pi^t, t = 1, \ldots, N$, whose induced occupancy measures $\lambda^{\pi^t}$ satisfy*

$$\mathbb{E}_{s \sim \rho} \mathbb{E}_\phi V_1^\star(s) \leq \mathcal{V}_{\phi^t}(\mathbb{E}^t \lambda^{\pi^t}), \tag{10}$$

*where we assume that the uncertainty measure $\sigma^t$ of the $\mathcal{V}_{\phi^t}$ function (3) satisfies*

$$\sigma_l^t(s, a) \leq \frac{c_0}{\sqrt{(n_l^t(s, a) \vee 1)}}, \tag{11}$$

*for some $c_0 > 0$ and any $t \in [N]$, $l \in [L]$, $(s, a) \in \mathcal{S}_l \times \mathcal{A}$, where we recall that $n_l^t(s, a)$ denotes the visitation count to $(s, a)$ at step $l$ before episode $t$ and $(\cdot \vee 1) := \max(\cdot, 1)$. Then it holds that*

$$\mathcal{BR}_\phi(\mathrm{alg}, T) \leq \sqrt{2c_0^2 TSA \log(SA)(1 + \log T/L)}$$
$$= \widetilde{O}(c_0 \sqrt{SAT}).$$

*Proof.* Note that $\mathbb{E}^t \lambda^{\pi^t}$ corresponds to the occupancy measure of policy $\pi^t$ on the expected transition dynamics $\mathbb{E}^t P$, *i.e.*, $\mathbb{E}^t \lambda^{\pi^t} \in \Lambda(\mathbb{E}^t P)$. We bound the Bayes regret of alg as

$$\mathcal{BR}_\phi(\mathrm{alg}, T) \overset{(i)}{=} \mathbb{E} \sum_{t=1}^N \mathbb{E}_{s \sim \rho} \mathbb{E}^t \left[ V_1^\star(s) - V_1^{\pi^t}(s) \right]$$

$$\overset{(ii)}{=} \mathbb{E} \sum_{t=1}^N \left[ \mathbb{E}_{s \sim \rho} \mathbb{E}^t V_1^\star(s) - \sum_{l=1}^L \sum_{s,a} \mathbb{E}^t \lambda_l^{\pi^t}(s, a) \mathbb{E}^t r_l(s, a) \right]$$

$$\overset{(iii)}{\leq} \mathbb{E} \sum_{t=1}^N \left[ \mathcal{V}_{\phi^t}(\mathbb{E}^t \lambda^{\pi^t}) - \sum_{l=1}^L \sum_{s,a} \mathbb{E}^t \lambda_l^{\pi^t}(s, a) \mathbb{E}^t r_l(s, a) \right]$$

$$\overset{(iv)}{=} \mathbb{E} \sum_{t=1}^N \sum_{l=1}^L \sum_{s,a} \mathbb{E}^t \lambda_l^{\pi^t}(s, a) \sigma_l^t(s, a) \sqrt{-2 \log \mathbb{E}^t \lambda_l^{\pi^t}(s, a)}$$

$$\overset{(v)}{\leq} \mathbb{E} \sqrt{\sum_{t=1}^N \sum_{l=1}^L \mathcal{H}(\mathbb{E}^t \lambda_l^{\pi^t})} \sqrt{\sum_{t=1}^N \sum_{l=1}^L \sum_{s,a} 2 \mathbb{E}^t \lambda_l^{\pi^t}(s, a) \sigma_l^t(s, a)^2}$$

$$\overset{(vi)}{\leq} \sqrt{\mathbb{E} \sum_{t=1}^N \sum_{l=1}^L \mathcal{H}(\mathbb{E}^t \lambda_l^{\pi^t})} \sqrt{\mathbb{E} \sum_{t=1}^N \sum_{l=1}^L \sum_{s,a} 2 \mathbb{E}^t \lambda_l^{\pi^t}(s, a) \sigma_l^t(s, a)^2},$$

where (i) follows from the tower property of conditional expectation where the outer expectation is with respect to $\mathcal{F}_1, \mathcal{F}_2, \ldots$, (ii) is from the dual formulation of the value function averaged with respect to the initial state distribution [66] and the fact that $\lambda_l^{\pi^t}(s, a)$ and $r_l(s, a)$ are conditionally independent given $\mathcal{F}_t$, (iii) is due to condition (10), (iv) is from the definition of $\mathcal{V}_{\phi^t}$, (v) and (vi) are by applying the Cauchy-Schwarz inequality, where $\mathcal{H}$ denotes the entropy function [10].

We bound the left term by using the fact that $\mathcal{H}(\mathbb{E}^t \lambda_l^{\pi^t}) \leq \log SA$. We bound the right term using condition (11) and by applying the pigeonhole principle (*e.g.*, [47, Lemma 6]) which gives

$$\mathbb{E} \sum_{t=1}^N \sum_{l=1}^L \sum_{s,a} \frac{\mathbb{E}^t \lambda_l^{\pi^t}(s, a)}{n_l^t(s, a) \vee 1} = \mathbb{E} \sum_{t=1}^N \sum_{l=1}^L \mathbb{E}^t \left( \sum_{s,a} \frac{\mathbb{E}^t \lambda_l^{\pi^t}(s, a)}{n_l^t(s, a) \vee 1} \right)$$

$$= \sum_{l=1}^L \mathbb{E} \left( \sum_{t=1}^N \sum_{s,a} \frac{\lambda_l^{\pi^t}(s, a)}{n_l^t(s, a) \vee 1} \right)$$

$$\leq \sum_{l=1}^L S_l A(1 + \log N)$$

$$= SA(1 + \log N),$$

which follows from the tower property of conditional expectation and since the counts at time $t$ are $\mathcal{F}_t$-measurable. Taking $T = NL$ finally yields

$$\mathcal{BR}_\phi(\text{alg}, T) \leq \sqrt{2c_0^2 NLSA \log(SA)(1 + \log N)}$$
$$= \widetilde{O}(c_0\sqrt{SAT}).$$

$\square$

**Theorem 1.** *For known $P$ and under Assumption 1, it holds that*

$$\mathcal{BR}_\phi(\text{VAPOR}, T) \leq \sqrt{2(\nu^2 + 1)TSA \log(SA)(1 + \log T/L)} = \widetilde{O}(\sqrt{SAT}).$$

*Proof.* Assumption 1 implies that the mean rewards are sub-Gaussian under the posterior (see [70, App. D.2]), where we can upper bound the sub-Gaussian parameter $\sigma_l^t(s, a) \leq \sqrt{(\nu^2 + 1)/(n_l^t(s, a) \vee 1)}$ at the beginning of each episode $t$. Thus the condition (11) holds for $c_0 = \sqrt{\nu^2 + 1}$. Moreover, Lemma 4 implies that the occupancy measures that solve the VAPOR optimization problem satisfy condition (10). The result thus follows from applying Lemma 12. $\square$

**Corollary 1.** *Denote by $\text{alg}_{\Gamma^\star}$ the algorithm that produces policies based on $\mathbb{P}_{\Gamma^\star}^t$ for each episode $t$ using (1). Then, under the same conditions as Theorem 1, we have $\mathcal{BR}_\phi(\text{alg}_{\Gamma^\star}, T) \leq \widetilde{O}(\sqrt{SAT})$.*

*Proof.* The result comes from combining Lemmas 4 and 12, as done in the proof of Theorem 1. $\square$

### F.8 Proof of Lemma 6

**Definition 2** (Transformed beliefs). *Consider any beliefs $\phi$ such that the mean rewards are $\sigma$-sub-Gaussian and the transition dynamics are $\alpha$-Dirichlet. We define the transformed beliefs $\widehat{\phi}$ according to which the transition dynamics are known and equal to $\mathbb{E}_\phi P$ and the rewards are distributed as $\mathbb{E}_\phi r + \omega$, where $\omega_l(s, a)$ follows an independent zero-mean Gaussian distribution with variance $\widehat{\sigma}^2$ defined for each step $l \in [L]$ and state-action pair $(s, a) \in \mathcal{S}_l \times \mathcal{A}$ as*

$$\widehat{\sigma}_l(s, a)^2 := 3.6^2 \sigma_l^2(s, a) + \frac{(L - l)^2}{\sum_{s' \in \mathcal{S}_{l+1}} \alpha_l(s, a, s')}.$$

We prove in Appendix F.8.1 the following stronger result which implies Lemma 6.

**Lemma 13.** *For any step $l \in [L]$ and state $s \in \mathcal{S}_l$, it holds that*

$$\mathbb{E}_\phi V_l^\star(s) \leq \mathbb{E}_{\widehat{\phi}} V_l^\star(s).$$

#### F.8.1 Proof of Lemma 13

**Overview.** The proof relies on studying properties of two stochastic Bellman operators, drawing inspiration from [62, Section 6.5]. In particular we will make use of the notion of stochastic optimism (Definition 3). To deal with the unknown transitions, we will use the property of Gaussian-Dirichlet optimism [60]. To deal with unknown rewards, we will derive a property of Gaussian-sub-Gaussian optimism (Lemma 19) which holds when the variance of the Gaussian distribution is multiplied by a small universal constant.

**Definition 3.** *A random variable $X$ is stochastically optimistic with respect to another random variable $Y$, written $X \geq_{SO} Y$, if $\mathbb{E}u(X) \geq \mathbb{E}u(Y)$ for all convex increasing functions $u : \mathbb{R} \to \mathbb{R}$.*

Likewise, we will say that $X$ is stochastically optimistic with respect to $Y$ under $\phi$ and write that $X \mid \phi \geq_{SO} Y \mid \phi$ if $\mathbb{E}_\phi u(X) \geq \mathbb{E}_\phi u(Y)$ for all convex increasing functions $u : \mathbb{R} \to \mathbb{R}$.

For any step $l \in [L]$, we introduce the Bellman operator $\mathcal{B}_l : \mathbb{R}^{S_{l+1} \times A} \to \mathbb{R}^{S_{l+1} \times A}$ which is defined for any $Q_{l+1} \in \mathbb{R}^{S_{l+1} \times A}$ and state-action pair $(s, a) \in \mathcal{S}_l \times \mathcal{A}$ as

$$\mathcal{B}_l Q_{l+1}(s, a) := r_l(s, a) + P_l(\cdot \mid s, a)^\top \max_{a'} Q_{l+1}(\cdot, a')$$

$$:= r_l(s, a) + \sum_{s' \in \mathcal{S}_{l+1}} P_l(s' \mid s, a) \max_{a'} Q_{l+1}(s', a').$$

We also define the 'transformed' Bellman operator $\widehat{\mathcal{B}}_l : \mathbb{R}^{S_{l+1} \times A} \to \mathbb{R}^{S_{l+1} \times A}$ as

$$\widehat{\mathcal{B}}_l Q_{l+1}(s,a) := \mathbb{E}_\phi r_l(s,a) + \omega_l(s,a) + \mathbb{E}_\phi P_l(\cdot \mid s,a)^\top \max_{a'} Q_{l+1}(\cdot, a'),$$

where we recall that $\omega_l(s,a) \sim \mathcal{N}\left(0, \widehat{\sigma}_l(s,a)^2\right)$ is distributed independently across the state-action pairs $(s,a) \in \mathcal{S}_l \times \mathcal{A}$ under $\phi$. Note that $\mathcal{B}_l$ and $\widehat{\mathcal{B}}$ can be viewed as randomized Bellman operators due to the randomness in the MDP $\mathcal{M}$ and in the distribution $\omega$, respectively. We rely on two key properties.

**Lemma 14** (Property 1: monotonicity). *For any step $l \in [L]$, consider two random Q functions $Q_{l+1}, Q'_{l+1} \in \mathbb{R}^{\mathcal{S}_{l+1} \times \mathcal{A}}$ such that conditioned on $\phi$ the entries of $Q_{l+1}$ (respectively $Q'_{l+1}$) are drawn independently across state-action pairs $(s,a) \in \mathcal{S}_{l+1} \times \mathcal{A}$ and drawn independently of the noise terms $\omega_l(s,a)$. Then*

$$Q_{l+1}(s,a) \mid \phi \geq_{SO} Q'_{l+1}(s,a) \mid \phi \quad \forall(s,a) \in \mathcal{S}_{l+1} \times \mathcal{A}$$

*implies*

$$\widehat{\mathcal{B}}_l Q_{l+1}(s,a) \mid \widehat{\phi} \geq_{SO} \widehat{\mathcal{B}}_l Q'_{l+1}(s,a) \mid \widehat{\phi} \quad \forall(s,a) \in \mathcal{S}_l \times \mathcal{A}, \ l \in [L].$$

*Proof.* The proof follows exactly the steps of [62, Lemma 3]. It relies on the fact that conditioned on $\widehat{\phi}$, $\widehat{\mathcal{B}}_l Q_{l+1}(s,a)$ is a convex increasing function of $(Q_{l+1}(s,a))_{s,a}$ convolved with the independent noise term $\omega_l(s,a)$. The result therefore follows from the fact that stochastic optimism is preserved under convex increasing operations [62, Lemma 2]. $\square$

**Lemma 15** (Property 2: stochastic optimism). *For any $l \in [L]$ and $(s,a) \in \mathcal{S}_{l+1} \times \mathcal{A}$, it holds that*

$$\widehat{\mathcal{B}}_l Q_{l+1}(s,a) \mid \widehat{\phi} \geq_{SO} \mathcal{B}_l Q_{l+1}(s,a) \mid \phi$$

*for any fixed $Q_{l+1} \in \mathbb{R}^{\mathcal{S}_{l+1} \times \mathcal{A}}$ such that $\mathrm{Span}(Q_{l+1}) \leq L - l$.*

*Proof.* For any fixed $Q_{l+1}$, it holds that

$$\widehat{\mathcal{B}}_l Q_{l+1}(s,a) \mid \widehat{\phi} \sim \mathcal{N}\left(b_l(s,a), \widehat{\sigma}_l(s,a)^2\right) \mid \phi,$$

where we define

$$b_l(s,a) := \mathbb{E}_\phi r_l(s,a) + \mathbb{E}_\phi P_l(\cdot \mid s,a)^\top \max_{a'} Q_{l+1}(\cdot, a').$$

**Lemma 16** ([60]). *Let $Y = \sum_{i=1}^n A_i b_i$ for fixed $b \in \mathbb{R}^n$ and random variable $A$, where $A$ is Dirichlet with parameter $\alpha \in \mathbb{R}^n$, and let $X \sim \mathcal{N}(\mu_X, \sigma_X^2)$ with $\mu_X \geq \frac{\sum_i \alpha_i b_i}{\sum_i \alpha_i}$ and $\sigma_X^2 \geq (\sum_i \alpha_i)^{-1} \mathrm{Span}(b)^2$, where $\mathrm{Span}(b) = \max_i b_i - \min_j b_j$, then $X \geq_{SO} Y$.*

In our case, in the notation of Lemma 16, $A$ will represent the transition function probabilities, and $b$ will represent $V := \max_a Q(\cdot, a)$, *i.e.*, for a given $(s,a) \in \mathcal{S}_l \times \mathcal{A}$ let $X$ be a random variable distributed as $\mathcal{N}(\mu_X, \sigma_X^2)$ where

$$\mu_X = \sum_{s' \in \mathcal{S}_{l+1}} \left( \alpha_l(s,a,s') V_{l+1}(s') / \sum_x \alpha_l(s,a,x) \right) = \sum_{s' \in \mathcal{S}_{l+1}} \mathbb{E}_\phi[P_l(s' \mid s,a)] V_{l+1}(s')$$

due to Assumption 2. Moreover, the lemma assumes that $\mathrm{Span}(V_{l+1}) \leq L - l$, so we choose $\sigma_X^2 = (L-l)^2/(\sum_{s'} \alpha_l(s,a,s'))$. As a result, it holds that

$$\mathcal{N}\left( \mathbb{E}_\phi P_l(\cdot \mid s,a)^\top \max_{a'} Q_{l+1}(\cdot, a'), \frac{(L-l)^2}{\sum_{s'} \alpha_l(s,a,s')} \right) \mid \phi \geq_{SO} P_l(\cdot \mid s,a)^\top \max_{a'} Q_{l+1}(\cdot, a') \mid \phi.$$

Moreover, from the assumption of $\sigma_l(s,a)$-sub-Gaussian $r_l(s,a)$ and from the Gaussian Sub-Gaussian stochastic optimism property of Lemma 19,

$$\mathcal{N}\left(0, 3.6^2 \sigma_l^2(s,a)\right) \geq_{SO} r_l(s,a) - \mathbb{E}_\phi r_l(s,a) \mid \phi,$$

which means that

$$\mathcal{N}\left(\mathbb{E}_\phi r_l(s,a), 3.6^2\sigma_l^2(s,a)\right) \mid \phi \ \geq_{SO} \ r_l(s,a) \mid \phi.$$

As a result, since the random variables $r_l(s,a)$, $P_l(\cdot \mid s,a)^\top \max_{a'} Q_{l+1}(\cdot, a')$, $\mathcal{N}\left(\mathbb{E}_\phi r_l(s,a), 3.6^2\sigma_l^2(s,a)\right)$ and $\mathcal{N}\left(\mathbb{E}_\phi P_l(\cdot \mid s,a)^\top \max_{a'} Q_{l+1}(\cdot, a'), \frac{(L-l)^2}{\sum_{s'} \alpha_l(s,a,s')}\right)$ are mutually conditionally independent, we have

$$
\begin{aligned}
\mathcal{N}\left(b_l(s,a), \widehat{\sigma}_l(s,a)^2\right) \mid \phi \ &\sim \ \mathcal{N}\left(\mathbb{E}_\phi r_l(s,a), 3.6^2\sigma_l^2(s,a)\right) \\
&\quad + \mathcal{N}\left(\mathbb{E}_\phi P_l(\cdot \mid s,a)^\top \max_{a'} Q_{l+1}(\cdot, a'), \frac{(L-l)^2}{\sum_{s'} \alpha_l(s,a,s')}\right) \mid \phi \\
&\geq_{SO} \ r_l(s,a) + P_l(\cdot \mid s,a)^\top \max_{a'} Q_{l+1}(\cdot, a') \mid \phi \\
&\sim \ \mathcal{B}_l Q_{l+1}(s,a) \mid \phi,
\end{aligned}
$$

which concludes the proof. $\qquad\square$

**Lemma 17.** *For any $l \in [L]$ and $(s,a) \in \mathcal{S}_l \times \mathcal{A}$,*

$$Q_l^\star(s,a) \mid \widehat{\phi} \ \geq_{SO} \ Q_l^\star(s,a) \mid \phi.$$

*Proof.* Note that under $\phi$, $Q_1^\star = \mathcal{B}_1\mathcal{B}_2\ldots\mathcal{B}_L 0$ and under $\widehat{\phi}$, $Q_1^\star = \widehat{\mathcal{B}}_1\widehat{\mathcal{B}}_2\ldots\widehat{\mathcal{B}}_L 0$. By Lemma 15,

$$(\widehat{\mathcal{B}}_L 0)(s,a) \mid \widehat{\phi} \ \geq_{SO} \ (\mathcal{B}_L 0)(s,a) \mid \phi.$$

Proceeding by induction, suppose that for some $l \leq L$,

$$(\widehat{\mathcal{B}}_{l+1}\ldots\widehat{\mathcal{B}}_L 0)(s,a) \mid \widehat{\phi} \ \geq_{SO} \ (\mathcal{B}_{l+1}\ldots\mathcal{B}_L 0)(s,a) \mid \phi.$$

Combining this with Lemma 14 shows

$$
\begin{aligned}
\widehat{\mathcal{B}}_l(\widehat{\mathcal{B}}_{l+1}\ldots\widehat{\mathcal{B}}_L 0)(s,a) \mid \widehat{\phi} \ &\geq_{SO} \ \widehat{\mathcal{B}}_l(\mathcal{B}_{l+1}\ldots\mathcal{B}_L 0)(s,a) \mid \widehat{\phi} \\
&\geq_{SO} \ \mathcal{B}_l(\mathcal{B}_{l+1}\ldots\mathcal{B}_L 0)(s,a) \mid \phi,
\end{aligned}
$$

where the final step uses Lemma 15 and the fact that $\mathrm{Span}(\mathcal{B}_{l+1}\ldots\mathcal{B}_L 0) \leq L - l$. $\qquad\square$

We are now ready to prove Lemma 13. We apply the property of stochastic optimism derived in Lemma 17 to the convex increasing function $u_s(Q) := \max_{a \in \mathcal{A}} Q(s,a)$ for every step $l \in [L]$ and state $s \in \mathcal{S}_l$, which yields the desired inequality $\mathbb{E}_\phi V_l^\star(s) \leq \mathbb{E}_{\widehat{\phi}} V_l^\star(s)$.

### F.9 Proof of Lemma 7

**Lemma 7.** *Let* alg *be any procedure that maps posterior beliefs to policies, and denote by* $\mathcal{BR}_{\mathcal{T},\phi}(\mathrm{alg}, T)$ *the Bayesian regret of* alg *where at each episode $t = 1, \ldots, T$, the policy and regret are computed by replacing $\phi^t$ with $\mathcal{T}(\phi^t)$ (Lemma 6), then under Assumptions 1 and 2,*

$$\mathcal{BR}_\phi(\mathrm{alg}, T) \leq \mathcal{BR}_{\mathcal{T},\phi}(\mathrm{alg}, T).$$

*Proof.* We recall that we denote by $\phi$ the prior on the MDP, by $\phi^t := \phi(\cdot \mid \mathcal{F}_t)$ the posterior beliefs at the beginning of each episode $t$ and by $\mathcal{T}$ the mapping that transforms the posteriors according to Lemma 6. Using the tower property of conditional expectation (where the outer expectation is with respect to $\mathcal{F}_1, \mathcal{F}_2, \ldots$), we can define the Bayes regret over $T$ timesteps of alg under the sequence of original and transformed posteriors respectively as

$$\mathcal{BR}_\phi(\mathrm{alg}, T) := \mathbb{E} \sum_{t=1}^N \mathbb{E}_{s \sim \rho} \mathbb{E}_{\phi^t}\left[V_1^\star(s) - V_1^{\pi^t}(s)\right],$$

$$\mathcal{BR}_{\mathcal{T},\phi}(\mathrm{alg}, T) := \mathbb{E} \sum_{t=1}^N \mathbb{E}_{s \sim \rho} \mathbb{E}_{\mathcal{T}(\phi^t)}\left[V_1^\star(s) - V_1^{\pi^t}(s)\right].$$

The result then immediately comes from combining Lemmas 13 and 18. $\qquad\square$

**Lemma 18.** *For any fixed policy $\pi$, it holds that*
$$\mathbb{E}_\phi V_1^\pi(s) = \mathbb{E}_{\hat\phi} V_1^\pi(s).$$

*Proof.* We prove by induction on the step $l$ that $\mathbb{E}_\phi Q_l^\pi(s,a) = \mathbb{E}_{\hat\phi} Q_l^\pi(s,a)$ for any $(s,a) \in \mathcal{S}_l \times \mathcal{A}$. This is true at step $L+1$. Assuming that it is true at step $l+1$, then

$$\mathbb{E}_\phi Q_l^\pi(s,a) - \mathbb{E}_{\hat\phi} Q_{l+1}^\pi(s',a')$$

$$= \mathbb{E}_\phi \left[ r_l(s,a) + \sum_{s'} P_l(s' \mid s,a) \sum_{a'} \pi_{l+1}(s',a') Q_{l+1}^\pi(s',a') \right]$$

$$- \mathbb{E}_{\hat\phi} \left[ \mathbb{E}_\phi r_l(s,a) + \omega_l(s,a) + \sum_{s'} \mathbb{E}_\phi P_l(s' \mid s,a) \sum_{a'} \pi_{l+1}(s',a') Q_{l+1}^\pi(s',a') \right]$$

$$= \sum_{s'} \mathbb{E}_\phi P_l(s' \mid s,a) \sum_{a'} \pi_{l+1}(s',a') \left[ \mathbb{E}_\phi Q_{l+1}^\pi(s',a') - \mathbb{E}_{\hat\phi} Q_{l+1}^\pi(s',a') \right]$$

$$= 0,$$

because $\mathbb{E}_\phi Q_l^\pi(s',a') = \mathbb{E}_{\hat\phi} Q_l^\pi(s',a')$ for every $(s',a') \in \mathcal{S}_{l+1} \times \mathcal{A}$ by induction hypothesis. In the above we used that $\mathbb{E}_\phi[P_l(s' \mid s,a) Q_{l+1}^\pi(s',a')] = \mathbb{E}_\phi P_l(s' \mid s,a) \mathbb{E}_\phi Q_{l+1}^\pi(s',a')$, which holds due to the time-inhomogeneity of the MDP, since the future return from a fixed state-action pair cannot be influenced by the dynamics that gets the agent to that state-action. As a result,

$$\mathbb{E}_\phi V_1^\pi(s) - \mathbb{E}_{\hat\phi} V_1^\pi(s) = \sum_a \pi_1(s,a) \mathbb{E}_\phi Q_1^\pi(s,a) - \sum_a \pi_1(s,a) \mathbb{E}_{\hat\phi} Q_1^\pi(s,a) = 0.$$

$\square$

### F.10  Proof of Theorem 2

**Theorem 2.** *Under Assumptions 1 and 2, it holds that $\mathcal{BR}_\phi(\mathtt{VAPOR}, T) \leq \widetilde{O}(L\sqrt{SAT})$.*

*Proof.* We first explicitly bound the uncertainty measure $\hat\sigma^t$ given our assumptions. The prior over the transition function $P_l(\cdot \mid s,a)$ is assumed Dirichlet, and let us denote the parameter of the Dirichlet distribution $\alpha_l^0(s,a) \in \mathbb{R}_+^{S_{l+1}}$ for each $(s,a)$, where $\sum_{s' \in \mathcal{S}_{l+1}} \alpha_l^0(s,a,s') \geq 1$, *i.e.*, we start with a total pseudo-count of at least one for every state-action (as done in [47]). Since the likelihood for the transition function is a Categorical distribution, conjugacy of the categorical and Dirichlet distributions implies that the posterior over $P_l(\cdot \mid s,a)$ at time $t$ is Dirichlet with parameter $\alpha_l^t(s,a)$, where $\alpha_l^t(s,a,s') = \alpha_l^0(s,a,s') + n_l^t(s,a,s')$ for each $s' \in \mathcal{S}_{l+1}$, where $n_l^t(s,a,s') \in \mathbb{N}$ is the number of times the agent has been in state $s$, taken action $a$, and transitioned to state $s'$ at timestep $l$, and note that $\sum_{s' \in \mathcal{S}_{l+1}} n_l^t(s,a,s') = n_l^t(s,a)$, the total visit count to $(s,a)$. Meanwhile, the reward noise is assumed additive $\nu$-sub-Gaussian, so we can upper bound the uncertainty measure $\sigma^t \in \mathbb{R}_+^{L,S,A}$ as $\sigma_l^t(s,a) \leq \sqrt{(\nu^2 + 1)/(n_l^t(s,a) \vee 1)}$. As a result, we can bound

$$\hat\sigma_l^t(s,a)^2 \leq \frac{3.6^2(\nu^2 + 1) + (L-l)^2}{(n_l^t(s,a) \vee 1)}. \tag{12}$$

The result follows from

$$\mathcal{BR}_\phi(\mathtt{VAPOR}, T) \leq \mathcal{BR}_{\mathcal{T},\phi}(\mathtt{VAPOR}, T) \leq \sqrt{2(3.6^2(\nu^2+1)+L^2) TSA \log(SA)(1 + \log T/L)}$$
$$= \widetilde{O}(L\sqrt{SAT}),$$

where the first inequality stems from Lemma 7 and the second inequality applies Lemma 12 whose condition (11) holds for $c_0 = \sqrt{3.6^2(\nu^2+1)+L^2}$. $\square$

### F.11  Proof of Lemma 8

**Lemma 8.** *Let $\lambda^{\mathrm{TS}}$ be the occupancy measure of the TS policy, it holds that $\mathbb{E}[\lambda^{\mathrm{TS}}] = \mathbb{P}_{\Gamma^\star}$.*

*Proof.* Given any environment $\mathcal{M}$ sampled from the posterior $\phi$, the optimal policy $\pi_{\mathcal{M}}^\star$ induces the occupancy measure $\lambda_l^{\pi_{\mathcal{M}}^\star}(s,a) = \mathbb{P}_{\mathcal{M}}(\Gamma_l^\star(s,a))$. Marginalizing over the environment $\mathcal{M}$ yields $\mathbb{E}[\lambda_l^{\mathrm{TS}}(s,a)] = \int_{\mathcal{M}} \lambda_l^{\pi_{\mathcal{M}}^\star}(s,a) d\mathbb{P}(\mathcal{M}) = \mathbb{P}_\phi(\Gamma_l^\star(s,a))$. $\square$

## F.12 Connection between K-learning and VAPOR

We sketch a proof of the connection between K-learning and VAPOR here. It proceeds via the dual in Lemma 9, by setting $\tau_l(s, a) = \tau$ for all $l, s, a$ and explicitly minimizing over the $V \in \mathbb{R}^{L,S}$ variable. First, we define the following $K$-function as done in the original K-learning paper [47]

$$K_l(s, a) = \mathbb{E}_\phi r_l(s, a) + \frac{\sigma_l^2(s, a)}{2\tau} + \sum_{s'} P_l(s' \mid s, a) V_{l+1}(s'),$$

for each $l, s, a$. After setting the $\tau$ parameters to be equal, we obtain the following dual over $K \in \mathbb{R}^{L, S \times A}, V \in \mathbb{R}^{L,S}, \tau \in \mathbb{R}$,

$$\text{minimize} \quad \sum_s \rho(s) V_1(s) + \tau \sum_{l,s,a} \exp\left(\frac{K_l(s,a) - V_l(s)}{\tau} - 1\right)$$
$$\text{subject to} \quad K_l(s, a) \geq \mathbb{E}_\phi r_l(s, a) + \frac{\sigma_l^2(s,a)}{2\tau} + \sum_{s'} P_l(s' \mid s, a) V_{l+1}(s')$$

where we have introduced the $K$-function as an inequality constraint that becomes tight at the optimum and preserves convexity. Explicitly minimizing over $V_1$ we obtain

$$V_1(s) = \tau \log \sum_a \exp\left(\frac{K_1(s, a)}{\tau}\right) - \tau \log \rho(s) - \tau$$

for each $s \in \mathcal{S}_1$, and doing the same for $V_l(s), l > 1$ we have

$$V_l(s) = \tau \log \sum_a \exp\left(\frac{K_l(s, a)}{\tau}\right) - \tau \log \mu_l(s) - \tau$$

for each $s \in \mathcal{S}_l$, where $\mu_l(s)$ is the dual variable associated with the inequality constraint and corresponds to the stationary state distribution at $s$. Substituting in for $V$ we can write the K-values as

$$K_l(s, a) = \mathbb{E}_\phi r_l(s, a) + \frac{\sigma_l^2(s, a)}{2\tau} + \sum_{s'} P_l(s' \mid s, a) \tau \log \frac{1}{\mu_l(s)} \sum_{a'} \exp\left(\frac{K_{l+1}(s', a')}{\tau}\right) - \tau,$$

Thus we have derived a variant of K-learning, with additional terms corresponding to the stationary state distribution. These additional terms becomes entropy in the objective function, and since the sum of these entropy terms can be bounded above by $L \log S$, it only contributes a small additional term to the regret bound. Practically speaking this version of K-learning requires knowing the stationary state distribution $\mu(s)$, so it is more challenging to implement than the original K-learning. This analysis is not to suggest this algorithm as a practical alternative, but instead to demonstrate the connection between K-learning and VAPOR.

# G   Gaussian Sub-Gaussian Stochastic Optimism

The following technical lemma, which may be of independent interest, establishes a relation of stochastic optimism (Definition 3) between a sub-Gaussian and a Gaussian whose variance is inflated by a small universal constant.

**Lemma 19.** *Let $Y$ be $\sigma$-sub-Gaussian. Let $X \sim \mathcal{N}(0, \kappa^2\sigma^2)$ with $\kappa \geq 3.6$. Then $X \geq_{SO} Y$.*

*Proof.* Let $u : \mathbb{R} \to \mathbb{R}$ be a convex (increasing) function. We aim to show that $\mathbb{E}[u(X)] \geq \mathbb{E}[u(Y)]$. Let $s_0 \in \mathbb{R}$ be a subgradient of $u$ at 0 (which exists since $u$ is convex). Then by convexity of $u$, we have for all $x \in \mathbb{R}$, $u(x) \geq u(0) + s_0 x$. Define the function $f : \mathbb{R} \to \mathbb{R}$ as $f(x) := u(x) - u(0) - s_0 x$. We have that $f(x) \geq 0$ for all $x \in \mathbb{R}$ and $f(0) = 0$. Since $\mathbb{E}X = \mathbb{E}Y = 0$, it suffices to show that $\mathbb{E}[f(X)] \geq \mathbb{E}[f(Y)]$. For any $\epsilon \geq 0$, define the $\epsilon$-sublevel set of the convex function $f$ as $K_\epsilon := \{x \in \mathbb{R} : f(x) \leq \epsilon\}$. Since $K_\epsilon$ is convex and by property of $f$, there exists $a(\epsilon) \in \mathbb{R}_- \cup \{-\infty\}$ and $b(\epsilon) \in \mathbb{R}_+ \cup \{+\infty\}$ such that $K_\epsilon = [a(\epsilon), b(\epsilon)]$. Then, for any random variable $Z$,

$$\mathbb{E}f(Z) = \int_0^\infty \mathbb{P}(f(Z) \geq \epsilon) \, d\epsilon$$
$$= \int_0^\infty (\mathbb{P}(Z \geq b(\epsilon)) + \mathbb{P}(Z \leq a(\epsilon))) \, d\epsilon.$$

By the $\sigma$-sub-Gaussian property of $Y$,

$$\mathbb{E}f(Y) = \int_0^\infty (\mathbb{P}(Y \geq b(\epsilon)) + \mathbb{P}(Y \leq a(\epsilon))) \, d\epsilon$$
$$= \int_0^\infty \left( \exp\left( \frac{-b(\epsilon)^2}{2\sigma^2} \right) + \exp\left( \frac{-a(\epsilon)^2}{2\sigma^2} \right) \right) d\epsilon$$
$$= \int_0^\infty (G(b(\epsilon)) + G(a(\epsilon))) \, d\epsilon,$$

where we define

$$G(x) := \exp\left( -x^2/(2\sigma^2) \right).$$

Let $d := 2.69$ and $c := \frac{\kappa}{d} \geq 1.338$. Let $Z \sim \mathcal{N}(0, c^2\sigma^2)$ and $F(x) := \mathbb{P}(Z \geq x)$. Note that

$$F(x) = \frac{1}{2}\left( 1 - \mathrm{erf}\left( \frac{x}{\sigma c\sqrt{2}} \right) \right),$$

where $\mathrm{erf}$ denotes the error function, *i.e.*, $\mathrm{erf}(y) = 2\pi^{-1/2} \int_0^y \exp(-t^2) \, dt$. From Lemma 20, by choice of constants $c, d$, it holds that $G(x) \leq dF(x)$. Therefore,

$$\mathbb{E}\left[ f(Y) \right] \leq d \int_0^\infty (F(b(\epsilon)) + F(a(\epsilon))) \, d\epsilon$$
$$= d \int_0^\infty (\mathbb{P}(Z \geq b(\epsilon)) + \mathbb{P}(Z \geq a(\epsilon))) \, d\epsilon$$
$$= d\mathbb{E}\left[ f(Z) \right]$$
$$\leq \mathbb{E}\left[ f(dZ) \right]$$
$$= \mathbb{E}\left[ f(X) \right],$$

where the last inequality applies Lemma 21 to the convex function $f$ that satisfies $f(0) = 0$, and the last equality is because $\kappa = cd$ which means that $X$ and $dZ$ follow the same distribution. $\square$

**Lemma 20.** *Denote by $c_0 > 0$ the (unique) solution of the equation*

$$\frac{1}{2}\left( 1 - \mathrm{erf}\left( \frac{1}{c_0\sqrt{2}} \right) \right) = \frac{1}{c_0\sqrt{2\pi}} \exp\left( -1/(2c_0^2) \right).$$

*and let $d_0 > 0$ be defined as*

$$d_0 := \frac{2\exp(-1/2)}{1 - \mathrm{erf}(\frac{1}{c_0\sqrt{2}})}.$$

*Note that $c_0 \approx 1.33016$ and $d_0 \approx 2.68271$.*

*Then for any $c \geq c_0, d \geq d_0$, the function $\phi : \mathbb{R} \to \mathbb{R}$ defined as*

$$\phi(x) := \frac{d}{2}\left(1 - \operatorname{erf}\left(\frac{x}{c\sqrt{2}}\right)\right) - \exp(-x^2/2)$$

*is non-negative everywhere.*

*Proof.* Define

$$\phi_0(x) := \frac{d_0}{2}\left(1 - \operatorname{erf}\left(\frac{x}{c_0\sqrt{2}}\right)\right) - \exp(-x^2/2).$$

It suffices to prove that $\phi_0(x) \geq 0$ for all $x \in \mathbb{R}$. By the choices of $c_0, d_0$, it holds that

$$\phi_0(1) = \frac{d_0}{2}\left(1 - \operatorname{erf}\left(\frac{1}{c_0\sqrt{2}}\right)\right) - \exp(-1/2) = 0,$$

$$\phi_0'(1) = \frac{-d_0}{c_0\sqrt{2\pi}}\exp\left(-1/(2c_0^2)\right) + \exp(-1/2) = 0.$$

Moreover, analyzing the variations of $\phi$ yields that there exists $x_0 > 1$ such that $\phi_0'(x) \geq 0$ for $x \in [1, x_0]$, and $\phi_0'(x) \leq 0$ otherwise. Therefore $\phi_0$ is non-decreasing on $[1, x_0]$ and non-increasing elsewhere. Since $\lim_{x \to +\infty} \phi_0(x) = 0$ and $\phi_0(1) = 0$, we get that $\phi_0(x) \geq 0$ for all $x \in \mathbb{R}$. $\qquad \square$

**Lemma 21.** *For any convex function $g : \mathbb{R} \to \mathbb{R}$ such that $g(0) = 0$, for any $s \geq 1$ and $x \in \mathbb{R}$, it holds that $sg(x) \leq g(sx)$.*

*Proof.* By convexity of $g$, for any $x' \in \mathbb{R}$, we have $g(s^{-1}x') \leq s^{-1}g(x') + (1 - s^{-1})g(0) = s^{-1}g(x')$, so by setting $x' = sx$ it holds that $sg(x) = sg(s^{-1}x') \leq g(x') = g(sx)$. $\qquad \square$

## H  VAPOR-lite Analysis

Denoting the uncertainty signal of the `VAPOR` optimization problem by $\widehat{\sigma}$, we define the `VAPOR-lite`$(c)$ optimization problem for any multiplicative scalar $c > 0$ as follows

$$
\max_{\lambda \in \Lambda(\mathbb{E}_\phi P)} \quad \mathcal{U}_{\widehat{\phi}}(\lambda, c)
$$

$$
:= \sum_{l,s,a} \lambda_l(s,a) \left( \mathbb{E}_\phi r_l(s,a) + c\widehat{\sigma}_l(s,a) - \sum_{a'} c\widehat{\sigma}_l(s,a') \frac{\lambda_l(s,a')}{\sum_b \lambda_l(s,b)} \log\left( \frac{\lambda_l(s,a')}{\sum_b \lambda_l(s,b)} \right) \right)
$$

$$
= \sum_{l,s,a} \lambda_l(s,a) \left( \mathbb{E}_\phi r_l(s,a) + c\widehat{\sigma}_l(s,a) + \mathcal{H}_{c\widehat{\sigma}_l(s,\cdot)}\left( \frac{\lambda_l(s,\cdot)}{\sum_b \lambda_l(s,b)} \right) \right).
$$

$$(13)$$

Note that this corresponds to the same objective as (5) except that it is optimized over the occupancy measures instead of the policies.

**Lemma 22.** *The* `VAPOR-lite`$(c)$ *optimization problem is concave in* $\lambda$.

*Proof.* The objective is concave due to the fact that the normalized entropy function $h$ of $u \in \mathbb{R}^n_{>0}$ given by $h(u) = \sum_{i=1}^n u_i \log(\mathbf{1}^\top u/u_i)$ is concave (see [6, Example 3.19]) since the perspective operation preserves convexity. $\qquad\square$

**Lemma 23.** *For any* $\eta > 0$ *and* $\lambda \in \Lambda(\mathbb{E}_\phi P)$, *it holds that*

$$
\mathcal{V}_{\widehat{\phi}}(\lambda) \leq \mathcal{U}_{\widehat{\phi}}(\lambda, c_\eta) + \sqrt{2}\widehat{\sigma}_{\max} LS\eta,
$$

*where* $c_\eta := \sqrt{2}(1 + \log 1/\eta)$ *and* $\widehat{\sigma}_{\max} := \max_{l,s,a} \widehat{\sigma}_l(s,a)$.

*Proof.* Any occupancy measure $\lambda \in \Lambda(\mathbb{E}_\phi P)$ can be decomposed as $\lambda_l(s,a) = \mu_l^\pi(s)\pi_l(s,a)$, where $\mu^\pi$ denotes the stationary state distribution under $\pi$, *i.e.*, $\mu_l^\pi(s) := \sum_{a \in \mathcal{A}} \lambda_l(s,a)$, and $\pi$ is the policy given by $\pi_l(s,a) := \lambda_l(s,a)/\mu_l^\pi(s)$ so long as $\mu_l^\pi(s) > 0$, otherwise $\pi_l(s,\cdot)$ can be any distribution, *e.g.*, uniform. It holds that

$$
\begin{aligned}
\lambda_l(s,a)\sqrt{-\log \lambda_l(s,a)} &\overset{(i)}{\leq} \lambda_l(s,a) - \lambda_l(s,a)\log\lambda_l(s,a) \\
&= \lambda_l(s,a) - \mu_l^\pi(s)\pi_l(s,a)\log\pi_l(s,a) - \pi_l(s,a)\mu_l^\pi(s)\log\mu_l^\pi(s) \\
&\overset{(ii)}{\leq} \lambda_l(s,a) - \mu_l^\pi(s)\pi_l(s,a)\log\pi_l(s,a) + \pi_l(s,a)\mu_l^\pi(s)\log 1/\eta + \pi_l(s,a)\eta \\
&= \lambda_l(s,a)\left(1 + \log 1/\eta\right) - \mu_l^\pi(s)\pi_l(s,a)\log\pi_l(s,a) + \pi_l(s,a)\eta,
\end{aligned}
$$

where (i) uses that $\sqrt{-\log x} \leq 1 - \log(x)$ for any $0 < x \leq 1$, and (ii) uses that $-x\log x \leq x\log 1/\eta + \eta$ for any $0 \leq x \leq 1$ and $\eta > 0$. Hence, we have

$$
\begin{aligned}
\mathcal{V}_{\widehat{\phi}}(\lambda) &= \sum_{l,s,a} \lambda_l(s,a)\mathbb{E}_\phi r_l(s,a) + \sum_{l,s,a} \widehat{\sigma}_l(s,a)\lambda_l^\pi(s,a)\sqrt{-2\log\lambda_l^\pi(s,a)} \\
&\leq \sum_{l,s,a} \lambda_l(s,a)\left( \mathbb{E}_\phi r_l(s,a) + \sqrt{2}(1 + \log 1/\eta)\widehat{\sigma}_l(s,a) \right) \\
&\quad + \sum_{l,s} \mu_l^\pi(s)\left( -\sum_a \sqrt{2}\widehat{\sigma}_l(s,a)\pi_l(s,a)\log\pi_l(s,a) \right) + \sum_{l,s,a} \sqrt{2}\eta\widehat{\sigma}_l(s,a)\pi_l(s,a) \\
&\leq \mathcal{U}_{\widehat{\phi}}(\lambda, c_\eta) + \sqrt{2}\widehat{\sigma}_{\max} LS\eta.
\end{aligned}
$$

$\qquad\square$

**Lemma 24.** *Consider the algorithm* `VAPOR-lite` *that at each episode* $t = 1, \ldots, N$ *solves the optimization problem* `VAPOR-lite`$(c = \sqrt{2}(1 + \log SLt))$. *Then under Assumptions 1 and 2, it holds that*

$$
\mathcal{BR}_\phi(\texttt{VAPOR-lite}, T) \leq \widetilde{O}(L\sqrt{SAT}).
$$

*Proof.* Denote by $\pi^t$ the policy executed by the algorithm at episode $t$, and by $\mathbb{E}^t\mathbb{P}_{\Gamma^\star}$ the posterior probability of state-action optimality in the expected model $\mathbb{E}^t P$. It holds that

$$\mathbb{E}_{s\sim\rho}\mathbb{E}^t V_1^\star(s) \overset{(i)}{\leq} \mathcal{V}_{\widehat{\phi}^t}(\mathbb{E}^t\mathbb{P}_{\Gamma^\star})$$

$$\overset{(ii)}{\leq} \mathcal{U}_{\widehat{\phi}^t}\left(\mathbb{E}^t\mathbb{P}_{\Gamma^\star}, 1/(SLt)\right) + \sqrt{2}\widehat{\sigma}_{\max}/t$$

$$\overset{(iii)}{\leq} \mathcal{U}_{\widehat{\phi}^t}\left(\mathbb{E}^t\lambda^{\pi^t}, 1/(SLt)\right) + \sqrt{2}\widehat{\sigma}_{\max}/t,$$

where (i) combines Lemmas 4 and 6, (ii) applies Lemma 23 to $\mathbb{E}^t\mathbb{P}_{\Gamma^\star} \in \Lambda(\mathbb{E}^t P)$ with the choice $\eta = 1/(SLt)$, and (iii) comes from the fact that $\mathbb{E}^t\lambda^{\pi^t} \in \arg\max_{\lambda\in\Lambda(\mathbb{E}^t P)}\mathcal{U}_{\widehat{\phi}^t}(\lambda, 1/(SLt))$. Hence, retracing the steps of the regret derivation of the proof of Lemma 12, we can bound the Bayes regret of `VAPOR-lite` as

$$\mathcal{BR}_\phi(\texttt{VAPOR-lite}, T)$$

$$\leq \mathbb{E}\sum_{t=1}^{N}\left[\mathcal{U}_{\widehat{\phi}^t}(\mathbb{E}^t\lambda^{\pi^t}, 1/(SLt)) - \sum_{l=1}^{L}\sum_{s,a}\mathbb{E}^t\lambda_l^{\pi^t}(s,a)\mathbb{E}^t r_l(s,a)\right] + \sqrt{2}\widehat{\sigma}_{\max}\sum_{t=1}^{N}\frac{1}{t}$$

$$\leq \underbrace{\mathbb{E}\sum_{t=1}^{N}\sum_{l=1}^{L}\sum_{s,a}\mathbb{E}^t\lambda_l^{\pi^t}(s,a)\sqrt{2}(1+\log SLt)\widehat{\sigma}_l^t(s,a)}_{:=Z_1}$$

$$+ \underbrace{\mathbb{E}\sum_{t=1}^{N}\sum_{l=1}^{L}\sum_{s,a}\mathbb{E}^t\lambda_l^{\pi^t}(s,a)\sqrt{2}(1+\log SLt)\sum_{a'}\left(-\widehat{\sigma}_l^t(s,a')\pi_l^t(s,a')\log\pi_l^t(s,a')\right)}_{:=Z_2}$$

$$+ \sqrt{2}\widehat{\sigma}_{\max}\sum_{t=1}^{N}\frac{1}{t}.$$

To bound $Z_1$ we apply the Cauchy-Schwarz inequality and the pigeonhole principle, as in the proof of Lemma 12, which gives that $Z_1 = \widetilde{O}(\widehat{\sigma}_{\max}\sqrt{SAT})$. To bound $Z_2$, we introduce the second moment of the information [33] of a discrete random variable $X$ supported on finite set $\mathcal{X}$ as

$$\mathcal{H}^{(2)}(X) := \sum_{x\in\mathcal{X}}\mathbb{P}(X=x)\left(-\log\mathbb{P}(X=x)\right)^2,$$

and note that $\mathcal{H}^{(2)}(X) \leq \log^2(|\mathcal{X}|)$ for $|\mathcal{X}| \geq 3$ [33, Proposition 8]. We bound

$$Z_2 \overset{(i)}{\leq} \mathbb{E}\sum_{t=1}^{N}\sum_{l=1}^{L}\sum_{s,a}\mathbb{E}^t\lambda_l^{\pi^t}(s,a)\sqrt{2}(1+\log SLt)\widehat{\sigma}_l^t(s,a)\left(-\log\mathbb{E}^t\lambda_l^{\pi^t}(s,a)\right)$$

$$\overset{(ii)}{\leq} \mathbb{E}\sqrt{\sum_{t=1}^{N}\sum_{l=1}^{L}\mathcal{H}^{(2)}(\mathbb{E}^t\lambda_l^{\pi^t})}\sqrt{\sum_{t=1}^{N}\sum_{l=1}^{L}\sum_{s,a}2\mathbb{E}^t\lambda_l^{\pi^t}(s,a)(1+\log SLt)^2\widehat{\sigma}_l^t(s,a)^2}$$

$$\overset{(iii)}{\leq} \widetilde{O}(\widehat{\sigma}_{\max}\sqrt{SAT}),$$

where (i) uses that $\mathbb{E}^t\lambda_l^{\pi^t}(s,a) \leq \pi_l^t(s,a)$, (ii) applies the Cauchy-Schwarz inequality and (iii) uses the pigeonhole principle (as in the $Z_1$ bound) combined with the aforementioned logarithmic upper bound on $\mathcal{H}^{(2)}$. Finally, we bound the last term by using that $\sum_{t=1}^{N}\frac{1}{t} \leq 1 + \log N$. Putting everything together and using that $\widehat{\sigma}_{\max} = O(L)$ yields the desired bound $\mathcal{BR}_\phi(\texttt{VAPOR-lite}, T) \leq \widetilde{O}(L\sqrt{SAT})$. $\qquad\square$

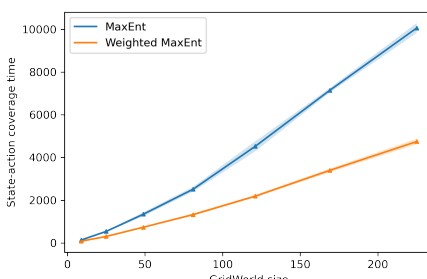

Figure 7: Optimizing for a weighted maximum entropy objective results in faster coverage time and thus more aggressive exploration in a reward-free GridWorld.

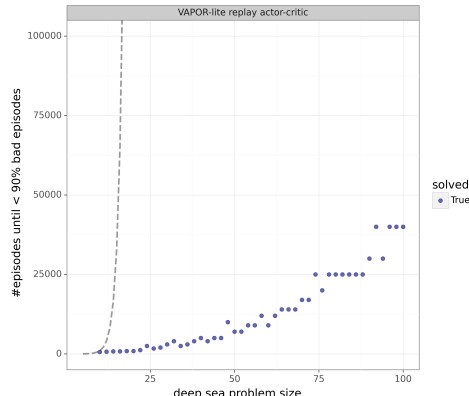

Figure 8: Learning time of `VAPOR-lite` on DeepSea (one-hot pixel representation into neural net). Dashed line represents $2^L$ where $L$ denotes the depth.

# I Experimental Details

## I.1 Reward-Free GridWorld

We empirically complement our discussion in Section 6 on the connection of `VAPOR` to maximum entropy exploration. We consider a simple 4-room GridWorld domain with no rewards, known deterministic transitions, four cardinal actions and varying state space size. We measure the ability of the agent to cover the state-action space as quickly as possible, *i.e.*, visit each state-action at least once. We consider the state-of-the-art algorithm of [79] for the original maximum entropy exploration objective $\max_{\lambda \in \Lambda(P)} \mathcal{H}(\lambda)$, and compare it to the same algorithm that optimizes for weighted entropy $\max_{\lambda \in \Lambda(P)} \mathcal{H}_\sigma(\lambda)$, where we define the uncertainty as $\sigma(s,a) = \mathbb{1}\{n(s,a) = 0\}$, with $n(s,a)$ the visitation count to state-action $(s,a)$. We see in Figure 7 that optimizing for this weighted entropy objective results in faster coverage time. This illustrates the exploration benefits of the state-action weighted entropy regularization of `VAPOR` (*e.g.*, weighted by the uncertainty as done in `VAPOR-lite`).

## I.2 DeepSea (tabular)

We provide details on the experiment of Figure 3. For the transition function we use a prior Dirichlet$(1/\sqrt{S})$ and for rewards a standard normal $\mathcal{N}(0,1)$, as done by [53]. Similar to [62, 47, 40], we accelerate learning by imagining that each experienced transition $(s, a, s', r)$ is repeated 100 times. Effectively, this strategy forces the MDP posterior to shrink faster without favoring any algorithm, making them all converge in fewer episodes.

`VAPOR` **implementation.** The `VAPOR` optimization problem is an exponential cone program that can be solved efficiently using modern optimization methods. We point out that we do not need to solve the two-player zero-sum game between $\lambda$ and $\tau$ since the minimization over $\tau$ conveniently admits a closed-form solution, see Equation (3). In our experiments, we use CVXPY [11], specifically the ECOS solver [12] (with 1e-8 absolute tolerance). In terms of runtime, this took a few seconds on the largest DeepSeas, which was sufficient for our purposes. There is a natural trade-off between a less accurate optimization solution (i.e., better computational complexity) and a more accurate policy (i.e., better sample complexity), which can be balanced with the choice of CVXPY solver, its desired accuracy, its maximum number of iterations, *etc.* The implementation of the `VAPOR` optimization problem in CVXPY is straightforward using the library's atomic functions. Just note that although the expression $x\sqrt{-\log x}$ that appears in the $\mathcal{V}_\phi$ function (3) is not an atomic function in CVXPY, a simple trick is to use the equivalence

$$\begin{aligned} \max_x \quad & x\sqrt{-\log x} & \Longleftrightarrow \qquad \max_{x,y} \quad & y \\ \text{s.t.} \quad & x \geq 0 & \text{s.t.} \quad & x \geq 0, \quad y \geq 0, \quad -x\log x \geq y^2/x \end{aligned}$$

where $-x\log x$ (cvxpy.entr$(x)$) and $y^2/x$ (cvxpy.quad_over_lin$(y,x)$) are atomic functions.

| Common Hyperparameter | Value | | | |
|---|---|---|---|---|
| Discount factor | 0.995 | | | |
| Replay buffer size | $1e5$ | | | |
| Replay fraction | 0.9 | | | |
| Replay prioritization exponent | 1.0 | | | |
| Adam step size | $1e-4$ | | | |
| $\lambda$ of V-Trace($\lambda$) | 0.9 | | | |
| Algorithm-specific Hyperparameter (Figure 10) | None | Fixed, scalar | Tuned, scalar | Tuned, state-action |
| Entropy regularization $\beta$ | / | 0.01 | / | / |
| Uncertainty scale $\sigma_{\text{scale}}$ | 0.01 | 0.01 | 0.01 | 0.005 |
| $\tau_{\min}$ | / | / | 0.005 | / |
| $\tau_{\max}$ | / | / | 10 | / |
| $\tau_{\text{init}}$ | / | / | 0.02 | / |
| $\tau$ step size | / | / | $1e-4$ | / |

Table 2: Hyperparameters used in the Atari experiments.

### I.3 DeepSea (neural network)

We consider the DeepSea domain where instead of using a tabular state representation, we feed a one-hot representation of the agent location into a neural network, using bsuite [58]. As discussed in prior works [62, 53, 49], agents that do not adequately perform deep exploration are far from being able to solve depths of up to 100 within $10^5$ episodes. This includes vanilla actor-critic or Soft Q-learning (which can only solve depths up to around 14, showcasing a learning time that suffers from an exponential dependence on depth), but also agents with some exploration mechanisms such as optimistic actor-critic or Bootstrapped DQN (depths up to around 50). In contrast, Figure 8 shows that `VAPOR-lite` is able to solve DeepSea instances out to size 100 within $4 \times 10^4$ episodes, without a clear performance degradation. We chose the exact same algorithmic configuration and hyperparameter choices as our experiments on Atari (Appendix I.4), except $\sigma_{\text{scale}} = 3.0$ and Replay fraction $= 0.995$. This experiment shows that `VAPOR-lite` is capable of *deep exploration*, as suggested by the regret analysis in Appendix H.

### I.4 Atari

**RL agent.** Our setup involves actors generating experience and sending them back to a learner, which mixes online data and offline data from a replay buffer to to update the network weights [30]. Our underlying agent is an actor-critic algorithm with use of replay data, thus we call it 'Replay Actor-Critic (RAC)'. We also made use of V-trace clipped importance sampling to the off-policy trajectories [15]. Replay was prioritized by TD-error and when sampling the replay prioritization exponent was 1.0 [72]. We note that, as commonly done on Atari, we consider the discounted RL setting (rather than the finite-horizon setting used to derive our theoretical results). We refer to Table 2 for hyperparameter details.

**Uncertainty measure in `VAPOR-lite`.** For the uncertainty measure $\widehat{\sigma}$, we use an ensemble of reward predictors. Specifically, we set it to the standard deviation of $H = 10$ randomly initialized reward prediction heads $\widehat{r}^{(i)}$ with randomized prior functions [56], *i.e.*,

$$\widehat{\sigma}(s, a) = \min\left( \sigma_{\text{scale}} \, \text{std}_{1 \leq i \leq H}(\{\widehat{r}^{(i)}(s, a)\}), 1 \right),$$

with $\sigma_{\text{scale}} > 0$ a scaling hyperparameter. We also added a small amount of noise to the rewards in the replay buffer for the targets for the reward prediction ensemble (specifically, 0.1 times an independent standard normal distribution) [14], to prevent the ensemble from potentially collapsing due to the use of replay data.

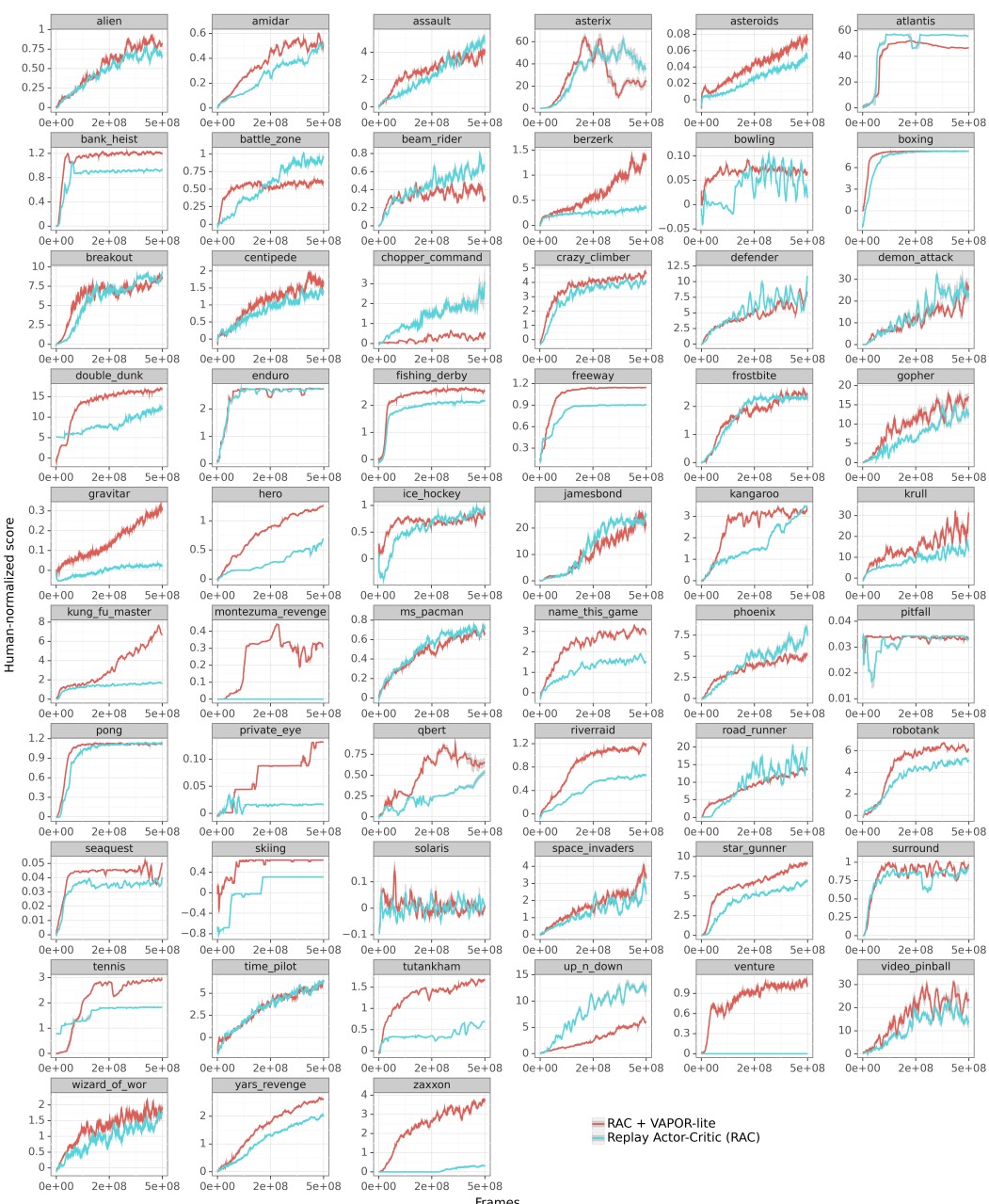

Figure 9: Per-game human normalized score (averaged over 5 seeds).

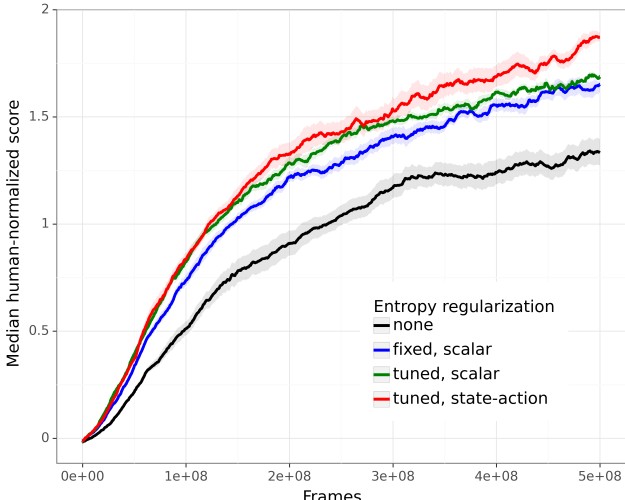

Figure 10: Median human normalized score across 57 Atari games (with standard errors across 5 seeds) of an *optimistic* replay actor-critic agent, for different choices of *entropy regularization*. `VAPOR-lite`'s 'tuned, state-action' performs best.

**Baselines.** In Figure 4, we compare `VAPOR-lite`, whose policy loss may be condensely expressed as

$$\max_{\pi \in \Pi} \quad (\mu^\pi \pi)^\top (r + \hat{\sigma}) + (\mu^\pi)^\top (\mathcal{H}_{\hat{\sigma}}(\pi)), \tag{14}$$

to a standard actor-critic objective

$$\max_{\pi \in \Pi} \quad (\mu^\pi \pi)^\top r, \tag{15}$$

as well as to an actor-critic with *fixed scalar* entropy regularization

$$\max_{\pi \in \Pi} \quad (\mu^\pi \pi)^\top r + (\mu^\pi)^\top (\beta \mathcal{H}(\pi)), \tag{16}$$

where $\beta > 0$ is a hyperparameter (*e.g.*, $\beta = 0.01$ in [42], which is also used in Figure 4 as it performed best). We later compare to additional baselines (Figure 10).

**On entropy regularization for policy improvement/evaluation.** We refer to [84] for in-depth discussion on the difference of using entropy for policy improvement and/or for policy evaluation in entropy-regularized RL. Theory prescribes both [87] and some algorithms such as Soft Actor-Critic [26] implement both, yet as remarked by [84], omitting entropy for policy evaluation (*i.e.*, not having entropy as an intrinsic reward) tends to result in more stable and efficient learning. We also observed the same and thus only use entropy for policy improvement.

**Related works.** We mention here a couple of related works to our proposed algorithm. [22] argue that standard policy entropy regularization can be harmful when the RL problem contains actions that are rarely useful, and propose a method that uses mutual-information regularization to optimize a prior action distribution. [27] discuss the limitation of policy entropy regularization being 'sample-unaware', and propose to regularize with respect to the entropy of a weighted sum of the policy action distribution and the sample action distribution from the replay buffer.

**Per-game results.** Figure 9 reports the performance of the Replay Actor-Critic (RAC) baseline compared to RAC augmented with VAPOR-lite, on each Atari game. We see that VAPOR-lite yields improvements in many games, especially in hard-exploration games such as Montezuma's Revenge or Venture.

**Ablation on entropy regularization.**   `VAPOR-lite` advocates for two ingredients: *optimism* and *weighted entropy regularization*. While the first has already been investigated in deep RL for various choices of uncertainty bonus [4], the second is novel so we focus our attention on it here. For a fair comparison, we augment all agents with the optimistic reward component, to isolate the effect of the adaptive state-action entropy regularization of `VAPOR-lite`. We consider the following policy losses for various choices of entropy regularization

$$\text{none:} \qquad \max_{\pi \in \Pi} \; (\mu^\pi \pi)^\top (r + \widehat{\sigma}) \,,$$

$$\text{fixed, scalar:} \qquad \max_{\pi \in \Pi} \; (\mu^\pi \pi)^\top (r + \widehat{\sigma}) + (\mu^\pi)^\top (\beta \mathcal{H}(\pi)) \,,$$

$$\text{tuned, scalar [49]:} \qquad \max_{\pi \in \Pi} \min_{\tau > 0} \; (\mu^\pi \pi)^\top \left(r + \widehat{\sigma}^2/(2\tau)\right) + (\mu^\pi)^\top (\tau \mathcal{H}(\pi)) \,,$$

$$\text{tuned, state-action (VAPOR-lite):} \qquad \max_{\pi \in \Pi} \; (\mu^\pi \pi)^\top (r + \widehat{\sigma}) + (\mu^\pi)^\top (\mathcal{H}_{\widehat{\sigma}}(\pi)) \,.$$

In particular, we have further compared to an optimistic actor-critic objective with *tuned, scalar* entropy regularization, specifically the epistemic-risk-seeking objective of [49] which corresponds to solving a principled saddle-point problem between a policy player and a scalar temperature player $\tau$ parameterized by a neural network. We refer to Table 2 for hyperparameter details. We observe in Figure 10 the isolated benefits of `VAPOR-lite`'s *tuned, state-action* entropy regularization.

**Future directions.**   Below we list some relevant directions for future empirical investigation:

- Although our simple $\widehat{\sigma}$ uncertainty measure using an ensemble of reward predictors worked well, more sophisticated domain-specific uncertainty signals could be used [63, 64, 7].
- A first 'looseness' of `VAPOR-lite` with respect to `VAPOR` is that it sets the temperatures $\tau$ to the uncertainties $\widehat{\sigma}$ (which yields the same $\widetilde{O}$ regret bound). An interesting extension for `VAPOR-lite` could be to optimize directly the temperatures by minimizing `VAPOR`'s saddle-point problem with a state-action dependent temperature player $\tau$ parameterized by an independent neural network.
- A second 'looseness' of `VAPOR-lite` with respect to `VAPOR` is that it bypasses the challenging optimization of the entropy of the state visitation distribution, by only (weight-)regularizing with the entropy of the policy. Various attempts have turned to density models or non-parametric entropy estimation [29, 36, 43], and it could be relevant to incorporate such techniques to get closer to the `VAPOR` objective.
- Our agent is relatively simple compared to modern state-of-the-art Atari agents since it is missing components like model-based rollouts, distributional heads, auxiliary tasks, *etc.* An interesting extension would be to incorporate the techniques discussed in this paper into the most effective policy-based agents.

