# OpenReview forum: "Probabilistic Inference in Reinforcement Learning Done Right"
_NeurIPS.cc/2023/Conference — NeurIPS 2023 poster_

### Official Review · Reviewer_MCkd · 2023-07-03

**Soundness:** 3 good
**Presentation:** 1 poor
**Contribution:** 3 good
**Rating:** 7
**Confidence:** 3

**Summary:**

The authors propose a model-based RL algorithm for MDPs with unknown rewards and transition dynamics, which approximates the posterior probability of an action being optimal in a given state to ensure efficient exploration. The approach is mostly contrasted with model-free RL as inference, but there is also an experiment comparing with a Thompson sampling-based baseline.

EDIT AFTER DISCUSSION
I revised my score from 3 to 7.

**Strengths:**

The presentation is clear and the proposed method is easy to understand.

**Weaknesses:**

I think the abstract is rather misleading and the authors are missing a large portion of existing literature on the topic. RL as inference, as presented by Levine [30], is a model-free algorithm. Any shortcomings of this approach notwithstanding, it is simple and computationally efficient, as model-free methods generally are. The abstract of this submission promises that the authors fix some problems with RL as inference, when in fact they just use a model to compute optimal actions instead. Sure, in some sense one could argue that this is better, but not exactly novel. The RL as inference framework is thus only tangentially relevant to the proposed method and the authors should compare with model-based approaches instead. There is some discussion of the relationship to PSRL in Section 6,  but way too little and too late.

I don't do model-based RL myself, so I'm not really in a position to evaluate how the proposed method relates to existing approaches in that space. I also don't really know the right papers to use as baselines, but here are a few articles I found with a quick search in case that's helpful.
Asmuth, J. and Littman, M. Learning is planning: Near Bayes-optimal reinforcement learning via Monte-Carlo tree search. In UAI, 2011.
Asmuth, J., Li, L., Littman, M. L., Nouri, A., and Wingate, D. A Bayesian sampling approach to exploration in reinforcement learning. In UAI 2009, Proceedings of the Twenty-Fifth Conference on Uncertainty in Artificial Intelligence, 2009, pp. 19–26. AUAI Press, 2009.
Guez, A., Silver, D., and Dayan, P. Efficient bayes-adaptive reinforcement learning using sample-based search. In NIPS, 2012.
Strens, M. J. A. A Bayesian framework for reinforcement learning. In Proceedings of the Seventeenth International Conference on Machine Learning (ICML 2000), pp. 943–950, 2000.

**Questions:**

Where does this paper stand with respect to existing literature on Bayesian approaches to model-based RL? Only once this question is answered exhaustively would it be possible to review this paper. I don't see how that's going to be possible within this cycle, though.

**Limitations:**

Unclear, as this paper is not properly positioned with respect to existing work.

---

> ### Author Rebuttal · Authors · 2023-08-09
>
> We thank the reviewer for their feedback. In this rebuttal we hope to convince you that our work is adequately positioned with respect to existing works on Bayesian model-based RL, and that its framing in the light of ‘RL as inference’ is relevant.
>
> __Where does this paper stand with respect to existing literature on Bayesian approaches to model-based RL?__ Thanks for the pointers to existing model-based literature! Bayesian model-based RL is indeed a rich line of research, inspired by Thompson (1933) and dating back from the 2000s (e.g., 2nd and 4th reference of the reviewer). These early works have laid the algorithmic foundations for the PSRL algorithm analyzed by Osband et al. (2013, 2017), which remains state-of-the-art both theoretically and empirically, along with the more recent K-learning. For completeness, we also point out that there are information-theoretic approaches (e.g., information-directed sampling [Russo and Van Roy, 2014], [Hao and Lattimore, 2022]) that provide elegant insights but are challenging to implement in practice (because of the difficulty of estimating mutual information). Finally, another line of research has adopted a tree-search approach to Bayesian RL (e.g., 1rst and 3rd reference of the reviewer), yet these approaches do not provide learning guarantees and are beyond the scope of this paper. All in all, we believe that PSRL, RLSVI and K-learning are the most relevant related works on no-regret Bayesian model-based RL and this is why we focus our theoretical (Section 6) and empirical (Section 8) comparison on them.
>
> [Russo and Van Roy, 2014] Learning to Optimize via Information-Directed Sampling.
>
> [Hao and Lattimore, 2022] Regret Bounds for Information-Directed Reinforcement Learning.
>
> __RL as inference, as presented by Levine [30], is a model-free algorithm…__ “RL as inference” in Levine, 2018 is not an algorithm but a framework. Since it is not an RL algorithm it is neither model-free nor model-based, although both model-free and model-based algorithms can in principle be derived from it. Our paper derives a _new_ framework that overcomes the (quite serious) shortcomings of the previous one. From our framework we can also instantiate both model-free and model-based algorithms, since we can use standard RL tools to solve the variational optimization problem we propose. We derive one particular model-based algorithm VAPOR, but our framework is not bound to be model-based. For instance, VAPOR-lite (Section 8 and Appendix E.3), which optimizes in the space of policies instead of occupancy measures, is a model-free, policy-gradient-based algorithm that approximates our probabilistic inference framework.
>
> __…they just use a model to compute optimal actions instead…__ We wish to clarify that one of our key insights is to sample (according to its posterior probability) the optimal action at any state _conditioned on the state being optimal_. We refer to this as state-action optimality (Definition 1) and prove that sampling from it yields efficient exploration and provides a principled view of RL as inference. We point out the subtle importance of such conditioning, without which simply following ‘action optimality’ is well known to lead to inefficient exploration (see our illustrative example in Section 3.1).

---

> > ### Comment · Reviewer_MCkd · 2023-08-17
> > **Clarifying questions**
> >
> > Having read the rebuttal and the other reviews, I think there's something interesting here, but the presentation of the paper makes it very difficult to judge what's novel, what problems it solves, and what are the limitations of the proposed approach. The paper is full of lengthy derivations and vague terms like "genuine statistical inference", "rigorous Bayesian treatment", or "false posterior", but there's little analysis of the objectives produced and how they compare to existing methods, and effectively no discussion of shortcomings and limitations. Before going further, some clarifying questions:
> > 1. It seems that the key step is to optimize in the space of occupancy measures. Is this approach novel? Or is anything about the optimization algorithm used novel? If not, what's the closest existing algorithm that operates in the space of occupancy measures?
> > 2. How much does the entropy term matter? It seems to be what's gained by the lengthy derivations. Did you run ablations without this term, just using the unconditional expected reward with an exploration bonus?
> > 3. Why does VAPOR always go right in Table 1? It has both entropy and exploration bonuses, so how can it produce a deterministic policy?
> > 4. In VAPOR-lite, once you throw away the entropy term, are you just adding a heuristic exploration bonus to the entropy-regularized actor-critic, or does it do anything more?

---

> > > ### Author Response · Authors · 2023-08-17
> > > **Response to Reviewer MCkd (1/2)**
> > >
> > > Thanks for your clarifying questions! $\newcommand{\PG}{\mathbb{P}_{\Gamma^\star}}$
> > >
> > > Our contributions can be centered around the new object that we uncover as key for inference and control, $\PG$:
> > > - We start by formalizing what it means for a ‘state-action pair to be optimal’. This event is at the core of the ‘RL as inference’ framework but had never been properly analyzed, leading to serious exploration shortcomings. Crucially, we reveal that the posterior probability of this event, $\PG$, is an occupancy measure.
> > > - We prove that $\PG$ suffices for principled exploration (i.e., extracting a policy from it has a guaranteed regret bound).
> > > - Since computing $\PG$ is intractable, we propose a variational optimization problem (VAPOR) that tractably approximates it.
> > > - We solve this optimization problem in two different ways: exactly (giving a tabular model-based algorithm with regret guarantees) and approximately (giving a scalable model-free policy-gradient algorithm).
> > > - We show that both TS and K-learning can also be directly linked to $\PG$, thus shedding a new light on these algorithms and tightly connecting them to our variational approach, and unifying these approaches within our framework.
> > >
> > > We will also make sure to expand on the limitations of our work in the revised version, in particular the challenge in solving the VAPOR optimization problem, the open problem of relaxing Assumption 1 for the regret analysis, and if there is a tighter way to approximate VAPOR with policy gradients.

---

> > > > ### Author Response · Authors · 2023-08-17
> > > > **Response to Reviewer MCkd (2/2)**
> > > >
> > > > 1. To the best of our knowledge, we are the first to operate in the space of occupancy measures for the purpose of Bayesian exploration (and ‘RL as inference’). This stems from our novel observation that the core quantity that formalizes RL as inference from a Bayesian viewpoint, $\mathbb{P}_{\Gamma^\star}$, is an occupancy measure, which gives rise to our optimization in this space.\
> > > > Operating in the space of occupancy measures has been done for various reasons in the RL literature, for instance for policy search [Peters et al., 2010; Suttle et al., 2022] or (frequentist) regret minimization in adversarial MDPs [e.g., Zimin and Neu, 2013; Rosenberg and Mansour, 2019; Jin et al., 2020]. \
> > > > The variational _optimization problem_ we propose is novel. It can be solved using various standard _optimization algorithms/techniques_. For instance, our tabular VAPOR learning algorithm uses convex optimization techniques (e.g., ECOS solver from CVXPY), while our approximation VAPOR-lite uses policy gradients. As a comparison, [Peters et al., 2010] have a different optimization problem in the space of occupancy measures, and they propose to solve it with BFGS or using sample-based policy iteration.
> > > >
> > > > 2. The entropy term matters both theoretically and empirically. In the tabular case, just being optimistic (i.e., exploration reward bonus) amounts to the UCBVI [Azar et al., 2017] algorithm (and variants), which have been shown to empirically perform worse than Bayesian algorithms (see e.g., Figure 2 of [O’Donoghue, 2021] on DeepSea). In deep RL, our Figure 7 in Appendix ablates the entropy regularization for an optimistic actor-critic agent on Atari. It reveals that VAPOR-lite’s ‘tuned, state-action’ entropy regularization performs better than standard ‘fixed, scalar’ regularization, which itself improves over ‘none’.
> > > >
> > > > 3. Great question! For the MDP in Fig 1 we can break down the behavior of the agent into two cases (the first state $s_1$, and any other state $s_{>1}$):\
> > > > \- $s_{>1}$: VAPOR will always go right for any state $s_{>1}$. This is because, conditioned on optimality only ‘move right’ has probability of being optimal under the posterior. \
> > > > \- $s_1$: VAPOR will go right with some probability $\geq 0.6$. This lower bound is determined by solving a convex optimization problem, and as $L \epsilon  \ll 1$ it tends to $1$. In the text we assumed that $L \epsilon \ll 1$, in which case the probability of moving right at $s_1$ is approximately $1$. Note that probability $\geq 0.6$ is sufficient for an expected regret bound tighter than that of TS, which always goes right from the first state with probability $= 0.5$. \
> > > > Recall that VAPOR uses a *state-action weighted* entropy, where the weighting is also computed from the beliefs. It turns out that the weighting on the entropy terms goes to zero for actions at $s_{>1}$ along the chain, allowing a deterministic policy.
> > > >
> > > > 4. VAPOR-lite maintains a weighted policy entropy regularization (it just removes the weighted entropy of the policy’s stationary state distribution). As such, VAPOR-lite is (i) adding an exploration bonus and (ii) adding a state-action weighted policy entropy regularization, the latter of which is novel. We refer to Appendix E.3 for details (e.g., Eq.16) but we will make sure to provide more details of VAPOR-lite in the main text!
> > > >
> > > > [Peters et al., 2010] Relative Entropy Policy Search\
> > > > [Suttle et al., 2022] Occupancy Information Ratio: Infinite-Horizon, Information-Directed, Parameterized Policy Search\
> > > > [Zimin and Neu, 2013] Online Learning in Episodic MDPs by Relative Entropy Policy Search\
> > > > [Rosenberg and Mansour, 2019] Online Convex Optimization in Adversarial MDPs\
> > > > [Jin et al., 2020] Learning Adversarial MDPs with Bandit Feedback and Unknown Transition\
> > > > [Azar et al., 2017] Minimax Regret Bounds for Reinforcement Learning\
> > > > [O’Donoghue, 2021] Variational Bayesian Reinforcement Learning with Regret Bounds

---

> > > > > ### Comment · Reviewer_MCkd · 2023-08-17
> > > > >
> > > > > Re 3, what you say about $s_{>1}$ is true, but how does VAPOR know that? Do you set $\sigma$ to zero for all states other than $s_L$? In general, I can't find a clear description in the paper for how $\sigma$ is selected.

---

> > > > > > ### Author Response · Authors · 2023-08-17
> > > > > > **Response to Reviewer MCkd**
> > > > > >
> > > > > > In that example, as is the usual case in Bayesian RL, the prior $\phi$ is known to all the algorithms. See the caption for Figure 1 or the text lines 137-144 for descriptions on what is known. So indeed VAPOR (and the Bayes-optimal algorithm, Thompson Sampling, etc.) is given the fact that the variance at states other than $s_L$ is zero, and that the reward at state $s_L$ is $\pm 1$ with probability $(0.5, 0.5)$ —
> > > > > > note that this is epistemic uncertainty and not aleatoric uncertainty. Assuming a less informative prior (i.e., $P$ and $r$ unknown) would lead to more complicated $\pi_l(s_l, \rightarrow)$ values (and initially $<1$) for any learning algorithm, but to similar conclusions: sampling according to Approximation 1 or the marginal $\mathbb{P}{\scriptsize \phi}(\pi^\star)$ suffers exponential regret, sampling according to the conditional $\mathbb{P}{\scriptsize \phi}(\pi^\star | \Gamma^\star)$, TS or VAPOR yields sublinear regret.

---

> ### Comment · Reviewer_MCkd · 2023-08-17
> **Updated evaluation**
>
> Following the discussion, here's my updated evaluation. I think there's an interesting contribution in this paper that could be published, but there are serious issues with presentation that make me hesitate to recommend acceptance. I don't agree with the framing of the paper, I think there's a substantial amount of discussion of existing literature that's missing, and, most importantly, the limitations of the proposed method are not discussed clearly enough.
>
> I think the key contribution is a tractable optimization objective, VAPOR, that approximates the optimal policy, correctly taking into account the epistemic uncertainty. The key insight is to optimize in the space of occupancy measures, targeting the occupancy measure corresponding to the optimal policy. I hesitate to call it Bayesian inference, since there aren't really any observations or posteriors in the usual sense. The "variational approximation" of VAPOR is also hardy justified. The solution of VAPOR is shown to be an upper bound on the expected reward of the optimal policy, but that of itself doesn't guarantee anything. The bound on KL between VAPOR policy and the optimal policy is self-referential. Ultimately, the justification of the algorithm is derived from the regret analysis, which is fine, but that's specifically not about the inference interpretation. I suppose it is debatable whether this whole procedure is in some sense Bayesian inference, but claims in published papers should not be debatable.
>
> Separately from the framing issue, the applicability of VAPOR is quite limited (I discuss the proposed extensions below). Specifically, there are two significant limitations:
> 1. VAPOR is only applicable in the tabular setting, and its optimization space grows with the product of states, actions, and time steps.
> 2. VAPOR can only handle epistemic uncertainty over reward distribution, and not the transition dynamics.
>
> I think those limitations are acceptable, but they need to be stated very clearly throughout the paper. As is, they are absent from the abstract and not prominently stated in the paper.
>
> The extension to uncertain transition dynamics relies on very restrictive assumptions. It's bad enough to require Dirichlet priors, but I think assuming independence is even worse. I don't mind a result like this being included, but again it should be very clear what the limitations are, along with some examples of priors which do and don't satisfy those assumptions. It's even less clear to me what happens to those assumptions later in the process, as beliefs are updated.
>
> VAPOR-lite, applicable in a non-tabular setting, is described very briefly and only in the appendix. If the authors want to claim that VAPOR is applicable outside of the tabular setting, VAPOR-lite needs to be given a prominent sport in the main body of the paper, with adequate analysis.
>
> As is, the discussion of existing literature is lacking, but I think the authors' responses cover the shortcomings. They just need to be included in the paper in some coherent form.
>
> Finally, this is all assuming that the theoretical analysis of the convergence bounds is correct. I'm not competent to check it, so I'm relying on the other reviewers for that.
>
> Overall, I think there's a good contribution here, but the authors oversell the paper quite a bit. Either walking the claims back or providing additional evidence for the questionable claims would work, but I think the latter would necessitate another round of reviewing.

---

> > ### Author Response · Authors · 2023-08-18
> > **Response to Reviewer MCkd**
> >
> > We appreciate the reviewer’s involved discussion. We wish to clarify some claims made by the reviewer.
> >
> > - “*There aren't really any observations or posteriors in the usual sense*”. Our object of interest is $\mathbb{P}_{\Gamma^\star}$, the **posterior** probability of state-action optimality conditioned on **observed data**. Computing this exactly requires Bayesian inference.
> > - The variational approximation and regret analysis are “*debatable / hardly justified*”. **RL is both inference and control**: it is important to analyze the control behavior resulting from our inference framework, and we believe regret is a relevant measure of performance. Our approach is ‘variational’ because we replace exact inference of an intractable posterior $\mathbb{P}_{\Gamma^\star}$ with a (convex) optimization problem.
> > - “*VAPOR can only handle epistemic uncertainty over reward distribution, and not the transition dynamics*”. Lemma 7 establishes an equivalence relationship which implies that VAPOR can handle epistemic uncertainty in the transition dynamics.
> > - “*The bound on KL between VAPOR policy and the optimal policy is self-referential*”. The bound relates the KL divergence of the policies to the absolute difference between the value functions. Thus it draws a connection between optimism (familiar to RL practitioners) and quality of approximation as measured by KL divergence (familiar to variational Bayes researchers).
> > - “*The solution of VAPOR is shown to be an upper bound on the expected reward of the optimal policy, but that of itself doesn't guarantee anything [...] The justification of the algorithm is derived from the regret analysis*”. The upper bound property and the concentration property is what guarantees the regret bound. The upper bound is ‘optimism’, an idea that is very standard in reinforcement learning.
> > - “*VAPOR is only applicable in the tabular setting, and its optimization space grows with the product of states, actions, and time steps.*” This is basically the case for all RL algorithms (without suitable parametrization) too. The point is to properly analyze our framework and obtain a guaranteed regret bound (on which we focus our paper), then transfer to more complicated setups with suitable parametrization of the policy or occupancy measure (which we briefly touch upon with VAPOR-lite, which is promising for more thorough future investigation).
> > - It is “*bad/worse to assume independent Dirichlet transition priors*" $\alpha$. This is the canonical prior for transitions in Bayesian RL, see “Bayesian Reinforcement Learning: A Survey” (Ghavamzadeh et al., 2016). This is because visiting a state-action pair simply increments by 1 the appropriate entry of the vector $\alpha$ — note the ease of the update because the Dirichlet is the **conjugate** prior of the categorical distribution (which models transition probabilities of discrete-state-action-MDPs). We hope this clarifies “*what happens to those assumptions later in the process, as beliefs are updated*”. We also note that the state-of-the-art Bayesian regret analyses (including PSRL, K-learning, RLSVI) assume it — it is an open question in the community how to obtain a $L \sqrt{S A T}$ Bayes regret bound without this assumption.

---

> > > ### Comment · Reviewer_MCkd · 2023-08-22
> > > **Have it your way**
> > >
> > > After some deliberation, I came to the conclusion that this is not worth arguing about. There's a good contribution here, the results are sound, and I would be doing a disservice to the community by making the authors jump through additional hoops and use up another set of reviewers' time in the next conference cycle. There are a lot of papers out there that overclaim their results and frame them in objectionable ways, and that's bad, but I'm not going to die on that hill. If there were a journal-like mechanism to review a revised manuscript, I would insist on some editorial changes, but there isn't, we're out of time and have to move on. I'll raise my score to recommend acceptance. I thank the authors for patiently answering my questions, and I hope they use this discussion to improve their paper.
> > >
> > > As a disclaimer, this evaluation is predicated on the proofs being correct, which I'm not able to check by myself. As of writing this, I believe that any concerns about correctness were adequately addressed in the rebuttal, but I reserve the right to lower my score if this changes.

---

### Official Review · Reviewer_QdEN · 2023-07-04

**Soundness:** 1 poor
**Presentation:** 2 fair
**Contribution:** 3 good
**Rating:** 3
**Confidence:** 4

**Summary:**

The paper views reinforcement learning as a Bayesian variational inference problem, and proposes VAPOR, an algorithm which computes an approximately optimal occupancy measure and its corresponding policy. Bayesian regret of VAPOR is analyzed to have a sub-linear bound. In numerical experiments, VAPOR shows good performance in simple GridWord and DeepSea environments. For more complex Atari environments, a further approximated VAPOR-lite is used and is compared with its entropy-regularized counterpart.

**Strengths:**

- While the idea of "RL as inference" has been considered, this paper proposes a new approach which aims to directly approximate the optimal occupancy measure in the Bayesian setting. This approach has the potential to better approximate Bayesian optimal reinforcement learning policy which might handle hard exploration situations.

- The motivation of the algorithm is explained well. The main idea of VAPOR is based on inequality (5) in Lemma 4. This inequality provides an upper bound of the optimal Bayesian performance, and the objective of VAPOR is basically finding the policy achieving the upper bound.  Some analysis of the approximation is provided in Lemma 5.

- The performance of VAPOR is analyzed in terms of Bayesian regret in Theorem 1 with known transitions, and then the analysis is extended in Theorem 2 under the independent Dirichlet assumption on the transition dynamics. Bayesian regret is shown to be sub-linear in both cases.

- VAPOR shows good performance compared with Thompson sampling based methods in simple GridWord and DeepSea environments.

- For more complex Atari environments, a further approximated VAPOR-lite is used and is compared with its entropy-regularized counterpart.


**Weaknesses:**

- Although Lemma 5 provides a bound on the KL-divergence between the VAPOR solution and the optimal occupancy measure, the bound is in terms of the gap of the VAPOR approximation of the optimal Bayesian performance in (5). So this bound is basically using a property of VAPOR itself to bound another property of VAPOR and none of them can be evaluated. Therefore, beside numerical experiments, we have no idea how loose the upper bound (5) can.

- Since the considered dynamics are time-inhomogeneous, the space of occupancy measure VAPOR is solving has dimension growing in the time horizon. This means that VAPOR might not be able to handle problems with long horizon, and this contradicts the statement at the end of Section 5 that VAPOR produces stationary policies, while it doesn't due to the time-inhomogeneous nature of the occupancy measure.

- In VAPOR, $\sigma$ seems to be a hyper-parameter, but it's not clearly defined. In particular, $\sigma$ seems to be a constant in most lemma and theorem statements, but $\sigma$ looks like a multi-dimensional vector with entry $\sigma_l(s, a)$ in (2) and (4). This confusion makes the description of the algorithm inconsistent and it might lead to some errors.

- The issue of $\sigma$ mentioned above might lead to a major error in the regret analysis and the VAPOR algorithm. In the proof of Lemma 12, equation (14) implies $c_l(s, a) = \sigma_l(s, a) \sqrt{n^t_l(s, a) \vee 1}$. According to (14), $c_{\max} = \max_{l, s, a} \sigma_l(s, a) \sqrt{n^t_l(s, a) \vee 1}$. Therefore, unless $\sigma_l(s, a)$ decrease at least at the rate of $1/\sqrt{n^t_l(s, a) \vee 1}$, the value of $c_{\max}$ will not be a constant.  But from the statement of Theorem 1, $\sigma$ seems to be treated as constant which is inconsistent with the proof of Lemma 12. It seems like, the uncertainty measure $\sigma$ needs to follow a decay procedure. Gradually decreased uncertainty is generally required for a policy to have sub-linear regret because non-decreasing randomness would lead to linear regret due to the constant noise in policy.

- Lemma 7 is very counter-intuitive. If it's true in general, uncertainty in the transition dynamics can be easily handled by replacing it with its mean. This would establish a kind of certainty equivalence for MDP which seems too strong to be true. I try to follow the proof of Lemma 7 and found several issues. Those issues may be correctable, but even if Lemma 7 is correct, it is probably too strong due to Assumption 1. Although Assumption 1 has been used in the literature, this assumption is restrictive and is not applicable in most applications. For example, any MDP with an unknown but fixed (over times/steps) transition function violates the assumption because the transition functions are not independent over times/steps.

- VAPOR-lite sounds promising with its application in complex environments like Atari games, but the paper only provides very limited information for VAPOR-lite and there is no details available for its implementation. The numerical results are also very limited, with no results on individual games and no comparison with popular existing algorithms.



**Questions:**

- As described in the weakness, is $\sigma_l(s, a)$ a constant or it follows a decay process? This seems critical to the algorithm and its analysis. It would be great if the authors can clarify the role of $\sigma_l(s, a)$ and correct the proofs one way or another.

- There are several issues in the lemmas that lead to the proof of Lemma 7.
  - In the proof of Lemma 13, summations are missing from Bellman operators. It seems like the authors may want to use some short-hand notations for those summations, what are the definitions for those notations? It's not possible to follow the proof without proper descriptions and definitions.
  - In the proof of Lemma 14, the second sentence claims to be conditioned on $\phi$, but $\hat{\mathcal B}_l$ is defined using the distribution of $\hat \phi$ instead of $\phi$. Is there some typos for $\hat \phi$ and $\phi$? Either way, the proof requires more steps transiting from $\phi$ to $\hat\phi$ to be true.
  - In the proof of Lemma 18, why can one brings the expectation $\mathbb E_\phi$ inside the Q-function? This seems to be a consequence of independent Dirichlet, but one needs to show that it leads to some kind of independence for the Q-function.


**Limitations:**

Beside possible errors in the analysis, the assumption makes the analysis in the paper limited and there is no discussion on this aspect.

---

> ### Author Rebuttal · Authors · 2023-08-10
>
> We thank the reviewer for their thorough review. When we encounter a reviewer that has clearly taken the time to read the paper in detail, as you have, and given the paper a low score, that is a signal to us that the paper is unclear in the highlighted areas — so thank you for bringing these issues to our attention! In this rebuttal we hope to clarify one-by-one your inquiries on the technical soundness, which we will incorporate in the revised version.
>
> __Q1: Is $σ_l(s,a)$ a constant or does it follow a decay process?__
> Thanks for flagging that $σ$ was overwritten as a vector following a decay process (Lemma 3) and as a constant (Theorem 1). In short, the correspondence is $σ^{\text{vector}}_l(s,a) = \frac{σ^{\text{constant}}}{\sqrt{n_l(s,a)}}$. First, the constant $σ^{\text{constant}}$ controls how noisy the rewards are (larger noise corresponds to a larger constant), we will replace it with the notation $c_0$ to avoid confusion. Second, we assume that the posterior uncertainty decays with data (of size $n$) at a particular $\frac{c_0}{\sqrt{n}}$ rate, which is a quite natural rate (also considered in the PSRL, RLSVI, K-learning analyses) that arises commonly in practice —  it holds, for instance, for Gaussian or bounded reward noise. As such, the condition of Lemma 12 should now read: ‘We _assume_ that the uncertainty $σ^t_l(s,a)$ decays at least as fast as $\frac{c_0}{\sqrt{n^t_l(s,a)}}$’.
>
> __Q2: Clarifications on the proof of Lemma 7.__
> - __Proof of Lemma 13.__ We will clarify the dot product notation $^\top$ in line 737 that gives the summation, i.e., for any step $l \in [L]$ the Bellman operator $B_l: \mathbb{R}^{S_{l+1} \times A} \rightarrow \mathbb{R}^{S_{l} \times A}$ is defined for any $Q_{l+1} \in \mathbb{R}^{S_{l+1} \times A}$ and $(s,a) \in \mathcal{S}_l \times \mathcal{A}$ as
>
> $(B_l Q_{l+1})(s,a) := r_l(s,a) + P_l(\cdot \mid s,a) ^\top \max_{a'} Q_{l+1}(\cdot, a') := r_l(s,a) + \sum_{s’ \in \mathcal{S_{l+1}}} P_l(s’ \mid s,a) \max_{a'} Q_{l+1}(s’, a')$.
>
> - __Proof of Lemma 14.__  Thanks for flagging that it should indeed be $\hat\phi$ instead of $\phi$ in the conditioning of the second sentence. We will expand the proof with more detail, with steps broken out clearly (due to the rebuttal character limit we do not include it here but we can add it as an Official Comment if the reviewer wishes). We also recall that the proof is not our contribution as it follows exactly the steps of the proof of Lemma 3 of RLSVI (Osband et al., 2019).
>
> - __Proof of Lemma 18.__ We understand your question as: why can we use $\mathbb{E}(P Q) = \mathbb{E}(P) \mathbb{E}(Q)$? This is due to the time-inhomogeneity of the MDP, which implies that $P$ at time $l$ and $Q$ at time $l+1$ are independent, since the future return from a fixed state-action pair cannot be influenced by the dynamics that gets the agent to that state-action.
>
> __KL-divergence bound of Lemma 5.__ The result shows that the weighted KL-divergence between $\mathbb{P}_{\Gamma}^{\star}$ and its variational approximation is bounded by how much the VAPOR objective is optimistic. Thus it draws a connection between optimism (familiar to RL practitioners) and quality of approximation as measured by KL divergence (familiar to variational Bayes researchers). For this we included Lemma 5, which we do not feel is a major piece of the paper, it is more about building intuition.
>
> __'Stationary' terminology.__ Following some prior works we used the term ‘stationary’ to describe a strategy that does not depend on the episode number K (unlike many frequentist algorithms that have a log(K) dependence in the algorithm). In other words, stationary algorithms are entirely a function of the beliefs, not of the episode number. We will clarify this definition in the paper.
>
> __Lemma 7 is counter-intuitive.__ Yes indeed, we believe that Lemma 7 is actually an important contribution of the paper! Intuitively speaking, it doesn’t really matter to the algorithm where the uncertainty comes from (between r and P), just that there is uncertainty at some far away state-action that must be reduced by visiting it. Note that the amount of uncertainty moved from the transitions into the rewards is large, O(L) for finite horizon L, so it’s not a ‘free lunch’ so to speak. We also emphasize that the result holds only for Dirichlet posteriors over P, so quite a lot of structure is required on the problem for it to hold. However, Dirichlet posteriors over P are very natural for transition functions so we believe it is useful in practice. In fact, similar results have been used in the literature for TS, RLSVI, and K-learning. Our contribution is to make it general (i.e., it holds for any algorithm) and extend to the full sub-Gaussian case rather than just the Bernoulli reward case.
>
> __Assumption of time-inhomogeneous dynamics.__ This assumption is not inherent to the VAPOR algorithm, but rather to existing $L\sqrt{SAT}$ Bayesian regret analyses. It is straightforward to instantiate the VAPOR algorithm under time-homogeneous dynamics (as we do in the GridWorlds of Fig. 2 and 6). The assumption is solely used to present clean theoretical results (which require the property that the transition function and the value function at the next state are conditionally independent). In fact, the analyses of K-learning, PSRL, RLSVI also require this assumption. Note that the example you gave (MDP with unknown but fixed transition function P) can be converted into a time-inhomogeneous MDP by unrolling (essentially copying the states L times), picking up an additional factor of sqrt(L) in the regret bound. In practice, this does not seem to matter much (most of these algorithms perform well in either case), but we take it for ease of analysis.
>
> __VAPOR-lite limited information.__ We will include much more details on the implementation given in Appendix E.3 (e.g., detailed pseudo-code and hyper-parameters) and add the learning curves for each individual Atari game.

---

> > ### Comment · Reviewer_QdEN · 2023-08-19
> >
> > I would like to thank the authors for the detailed responses. From the response, Theorem 1 indeed requires an additional assumption to be correct. The likely non-verifiable assumption not only makes the results much weaker, there are still many unspecified details missing for this possible assumption. If the assumption considers the vector-version $σ_l(s,a)$ under the prior distribution as in Lemma 3, they are still time independent, and therefore cannot decay with $t$. If the assumption will involve some time-dependent $σ^t_l(s,a)$ which might come from the posterior, the paper needs to add more details on the posterior distribution and some analysis on its evolution under the proposed policy. I feel like these issues require some major revision and another round of reviews.

---

> > > ### Author Response · Authors · 2023-08-19
> > > **Response to Reviewer QdEN**
> > >
> > > We thank the reviewer for engaging. We believe that the reviewer has misunderstood our use of standard and familiar tools in Bayesian inference and hope to correct that misunderstanding below.
> > >
> > > We do not require any additional assumptions for Theorem 1. The only fact that we rely on is that in Bayesian inference the posteriors concentrate as we gather more data. This is entirely standard, e.g., [1] and references therein. The only question is how fast do the posteriors concentrate: Under our already stated assumptions the variance of the posteriors concentrate like 1/(#data samples).
> > >
> > > Concretely, as the agent navigates the environment it visits state-actions at *different schedules*, so the posteriors concentrate (uncertainty decays) at the same 1/(#data samples) *rate*, but at *different schedules*. As is usual in RL, we have assumed sub-Gaussian additive reward noise. That implies the following:
> > >
> > > - If each state-action reward posterior starts with some $\kappa$ uncertainty (the prior) and the reward noise has variance $c_0^2$ and for simplicity take $\kappa \gg c_0$ (if not it can only help the posteriors concentrate faster),
> > > - Then at time $t$ the uncertainty has decayed to $c_0^2 / n^t(s,a)$, where $n^t(s,a)$ is the number of times the agent has visited state-action (s,a) before episode $t$ (ie, the number of data samples of the reward from state-action (s,a)).
> > >
> > > This rate is all we need to prove the regret bound. It is ‘time-dependent’, but only in the sense that as time progresses the agent is gathering more data and the posteriors (or confidence sets) are concentrating, this is entirely standard in RL, both frequentist and Bayesian approaches (otherwise how would the agent ever learn?).
> > >
> > > The sub-Gaussian assumption is standard in the literature and arises commonly in practice. For instance, in both Atari and DeepSea (the experiments we ran) the assumption holds because the rewards are bounded.
> > >
> > > In slightly more detail:
> > > - Theorem 1 reads: For known $P$ and $c_0$-sub-Gaussian additive reward noise, it holds that $\mathcal{BR}(\text{VAPOR}, T) \leq c_0 \sqrt{S A T}$.
> > > - The *definition* of $c_0$-sub-Gaussian additive reward noise means that: at any time $t$, step $l$, state-action pair $(s,a)$, $\mathbb{E}^t [\exp(x (r_l(s,a) - \mathbb{E}^t r_l(s,a)))] \leq \exp \left(x^2 c_0^2 / 2 (n_l^t(s,a) \vee 1) \right), \quad \forall x \in \mathbb{R}.$
> > > - This means that at any time $t$, we can *instantiate* the uncertainty $\sigma_l^t(s,a)$ in the VAPOR optimization problem as: $\sigma_l^t(s,a) = \frac{c_0}{\sqrt{(n_l^t(s,a) \vee 1)}}$.
> > >
> > > [1] Conjugate Bayesian analysis of the Gaussian distribution, Murphy, 2007.

---

> > > > ### Comment · Reviewer_QdEN · 2023-08-21
> > > >
> > > > Appreciate the authors' further clarification. As we discussed, Theorem 1 clearly requires concentration of the posterior. There is no discussion nor analysis on the concentration of posterior in the current version of the paper. One possible way to handle the posterior concentration could be making it as an assumption of the algorithm, but it then required much further discussion on why this assumption makes sense. Another possibility is that the posterior concentration is a consequence of the assumptions on prior and noise distributions, which is kind of suggested by the authors' response. This concentration could be true and might be established as in other works, but to do it the paper needs to provide analysis on the posterior evolution and prove that the posterior distribution indeed satisfies the desired concentration property. Either way, the paper requires extra justifications and does not look ready in the current form.

---

> > > > > ### Author Response · Authors · 2023-08-21
> > > > >
> > > > > In the paper we clearly state that we assume additive sub-Gaussian reward noise. For simplicity let's just say the reward noise is Gaussian. It is a standard result that variance of the posterior of a Gaussian concentrates like $1/n$ (for a worst case, improper prior - any other prior would have *faster* concentration). We are simply using that fact for the posterior of the mean reward at each state and action. Sub-Gaussian simply replaces the exact concentration of $1/n$ with an upper bound of $1/n$. There is no extra analysis we can do other than to re-prove ancient results about Gaussians that you can read from this wikipedia page: https://en.wikipedia.org/wiki/Conjugate_prior#When_likelihood_function_is_a_continuous_distribution

---

> > > > > > ### Comment · Reviewer_MCkd · 2023-08-21
> > > > > > **Not my thread, but adding my two cents**
> > > > > >
> > > > > > I think a lot of confusion could be avoided if the authors clearly stated their assumptions regarding the priors and likelihoods used to model the environment. The paper silently assumes that we can exactly compute the posterior, which is generally not tractable. The authors avoid this problem by using independent conjugate priors, where the beliefs can be updated in closed form. This is fine, but something that needs to be clearly stated. A short paragraph in the preliminaries describing how the inference part is handled would go a long way towards making the paper more accessible.
> > > > > >
> > > > > > Now, to the issue at hand. I agree this concentration property holds for standard conjugate priors. What's not clear to me is whether this extends to the general case, where the likelihood is not Gaussian, the prior is not conjugate, and the priors on different variables (indexed by t,s,a) are not independent. I suspect that it does, but it's not obvious to me.

---

> > > > > > > ### Author Response · Authors · 2023-08-22
> > > > > > >
> > > > > > > We state all the assumptions we require clearly in Theorem 1 and Assumption 1. These are:
> > > > > > > - (A) A time-inhomogeneous MDP.
> > > > > > > - (B) The mean rewards are bounded in [0, 1] almost surely with independent priors.
> > > > > > > - (C) The reward noise is additive σ-sub-Gaussian.
> > > > > > > - (D) The prior over transition functions is independent Dirichlet.
> > > > > > >
> > > > > > > We are happy to add a paragraph explaining these assumptions in more detail, but we are not "silently assuming that we can exactly compute the posterior" or anything like that, nor do we require closed-forms of the posteriors. Note combining assumptions (B) and (C) implies that the posterior is also sub-Gaussian (see App D.2 of [Russo and Van Roy, 2016, An Information-Theoretic Analysis of Thompson Sampling]). The sub-Gaussian property means we can *upper bound* _intractable_ posteriors with a _tractable Gaussian_ distribution. We do not require conjugate priors (other than Dirichlet) or Gaussian likelihoods, only these assumptions.
> > > > > > >
> > > > > > > This is the exact same set of assumptions as the published, state-of-the-art related works of K-learning [O’Donoghue, 2021, Asm 1], PSRL [Osband et al., 2017, Lem 3] and RLSVI [Osband et al., 2019, Asm 2 and 3].
> > > > > > >
> > > > > > > Relaxing (partially) these assumptions without suffering a worse regret bound is indeed an important open question, which we believe is outside the scope of this paper but which we will clearly highlight in the paper as per reviewers’ suggestion.  Finally, we emphasize that these assumptions are technicalities for the regret analysis, not fundamental ones for the algorithmic idea. A practical (approximate) implementation of VAPOR would likely be model-free and compute heuristically an uncertainty signal $\sigma(s,a)$, such as with an ensemble of reward predictors as done in VAPOR-lite.

---

> > > > > > > > ### Comment · Reviewer_MCkd · 2023-08-22
> > > > > > > > **I'm satisfied**
> > > > > > > >
> > > > > > > > OK, thank you for providing a clarification. I'm satisfied on this front, unless Reviewer QdEN objects further.

---

> > > > > > > > > ### Comment · Reviewer_QdEN · 2023-08-22
> > > > > > > > >
> > > > > > > > > Not sure if they are typos or just very bad notations, but $\phi$ is referred to as the prior in some places (like line 200) but called the posterior in some other places (like line 164). This inconsistency makes the paper almost not possible to understand, and it also creates issues for many claims, including those in the responses.
> > > > > > > > >
> > > > > > > > > If we view $\phi$ in Assumption 1 as the prior as assumptions in the literature, there are still many issues. First, prior distribution and observation noise are different things, but the way the assumption is written messed up with the two. Second, despite the first issue, the assumption is still not exactly the same as in prior papers. For example, the PSRL paper does not assume time-inhomogeneous and independent prior across time. Lastly, despite all the mess-ups in notations/typos, there is no analysis on the posterior concentration. Concentration could well be true, but there is no proof, not even a statement, on the posterior concentration in the paper.
> > > > > > > > >
> > > > > > > > > In RL problems, because the posterior distribution depends on the algorithm which collects the data, there could be some subtle but critical issue appearing the the posterior analysis given certain properties of the algorithm. The claims in the paper may be true, but they can only be verified after the notations/typos being fixed, the non-consistent statements being rewritten, and the proof details being added.

---

> > > > > > > > > > ### Comment · Reviewer_QdEN · 2023-08-22
> > > > > > > > > >
> > > > > > > > > > Just want to make a comment on the necessity of a careful analysis for posterior in RL. Some properties for posterior do not always hold for all RL algorithms. For example, Example 2 in Osband, Ian, and Benjamin Van Roy. "Posterior sampling for reinforcement learning without episodes." arXiv preprint arXiv:1608.02731 (2016) provides a counterexample of a posterior sampling property under general policies.

---

### Official Review · Reviewer_UaER · 2023-07-06

**Soundness:** 4 excellent
**Presentation:** 3 good
**Contribution:** 3 good
**Rating:** 6
**Confidence:** 2

**Summary:**

The paper provides a Bayesian treatement for the reinforcement learning as probabilistic inference framework for the discrete state/action space. As extension to the standard formulation, posterior probabilities of state-action optimalities are regarded and formally defined. For tractability, an approximation of the probabilities via variational inference is derived. Further, the case of unknown dynamics is considered and links to other approaches discussed. The authors evaluate their method on a grid world problem, deep sea, and (in an reduced form) to atari.

**Strengths:**

The problem which is considered in the paper is interesting for the use in state-of-the-art reinforcement learning. While not being obvious at first sight why a Bayesian treatment is advantageous, the simple decision problem makes that very clear.

By providing a formalization of the Bayesian formulation and the variational approximation and details about the algorithm the contribution of the paper is very good.

The paper is well-written and easy to follow.

While just briefly looking at the appendix, all theorems seem to be proven and assumptions listed.

**Weaknesses:**

There is no code provided in the supplementary material or paper. In these days, I would expect this for an accepted paper at NeurIPS.

Future work and limitations are not sufficiently discussed in my opinion.

For atari evaluation, only a reduced version of the algorithm is considered (without weighted entropy)

Information on how to solve the VAPOR objective are only given in the appendix.

The paper seems very crowded, for example the table besides lemma 6 is a bit confusing. It is good, however, that it has a lot of content/contribution.

Minor:
The quantity $\lambda$ seems not to be well-defined for me as it is used a function but not defined this way. It took me a while to understand what it really is. This should be written in a more consistent way.

line 110: Add "pair" after state-action.

**Questions:**

How strong is the assumption for VAPOR that $r_l$ is a sub-Gaussian (in Lemma 3)?

Is it possible to apply the method to continuous state/action spaces?

What are the runtimes for the experiments?

**Limitations:**

The formulation is limited to discrete/action spaces and does not seem to be scalable.

No runtimes are provided. It would have been very interesting to see how much time it takes to solve the two-player zero-sum geme in every iteration.

---

> ### Author Rebuttal · Authors · 2023-08-10
>
> We thank the reviewer for their valuable comments and suggestions (such as expanding on the future work and limitations), which we will incorporate in the revised version.
>
> __No code.__ We will include more details on the implementation of both VAPOR and VAPOR-lite, such as detailed pseudo-code (including the code snippet of the VAPOR optimization problem that we solve using CVXPY) and hyper-parameter list.
>
> __$\lambda$ notation.__ We will clarify on lines 71-74 that the notation $\lambda_l \in \mathbb{R}+^{S \times A}$ means that $\lambda_l$ is a function $S \times A \rightarrow \mathbb{R}_{+}$ for every $l \in [L]$.
>
> __Q1: How strong is the assumption for VAPOR that $r_l$ is a sub-Gaussian (in Lemma 3)?__ We consider the sub-Gaussian mean reward assumption in the main paper since it is very standard in the literature (e.g., K-learning [O’Donoghue, 2021], PSRL [Osband and Van Roy, 2017], RLSVI [Osband et al., 2019]) and it allows us to present clean results and regret bounds. This assumption is not required _computationally_ for VAPOR (only for the analysis): in Appendix C.3 we extend Lemma 3 to the general case, and we refer to Appendix C.5 for the resulting VAPOR optimization problem.
>
> __Q2: Is it possible to apply the method to continuous state/action spaces?__ Our new framework of ‘RL as inference’ may be instantiated and approximated in various ways, targeting theoretical guarantees or scalability, which dictates which standard RL tools to use to solve the variational optimization problem we propose. In particular, we derive a tabular, model-based algorithm VAPOR with theoretical guarantees (Algorithm 1). Meanwhile, by optimizing in the space of policies instead of occupancy measures, VAPOR-lite is a model-free, policy-gradient-based algorithm that approximates our probabilistic inference framework, so it can readily leverage existing policy-gradient techniques suitable for continuous state/action spaces.
>
> __"Only a reduced version of the algorithm is considered (without weighted entropy)".__
> We will make sure to add more details on VAPOR-lite in the revised version. We will clarify that VAPOR-lite does consider weighted entropic regularization (but in the space of policies instead of occupancy measures, which is a weaker but more scalable form of regularization). As such, VAPOR-lite retains the core algorithmic novelty of VAPOR of weighting optimism and entropy regularization on a per state-action basis.
>
> __Q3: What are the runtimes for the experiments?__
> Although VAPOR requires solving a convex optimization problem at each iteration, which makes it slower than dynamic programming approaches like Thompson Sampling, the problem is an exponential cone program that can be solved efficiently using modern optimization methods. We point out that we do not need to solve the two-player zero-sum game since the minimization over $\tau$ conveniently admits a closed-form solution (see Equation 4). In our experiments, with the CVXPY solver ECOS (and 1e-8 absolute tolerance), solving the VAPOR optimization problem took a few seconds on the largest DeepSeas, which was sufficient for our purposes. There is a natural trade-off between a less accurate optimization solution (i.e., better computational complexity) and a more accurate policy (i.e., better sample complexity), which can be balanced with the choice of CVXPY solver, its desired accuracy, its maximum number of iterations, etc. We will include runtime details and discussion in the revised version. As for VAPOR-lite, its computational cost is essentially the same as the replay actor-critic baseline (it only needs to compute the uncertainty signal σ with an ensemble of reward predictors).

---

> > ### Comment · Reviewer_UaER · 2023-08-20
> >
> > Thank you for answering my questions. After reading the other reviews and discussions, I have decreased my confidence as the other reviewers probably have a deeper understanding of the paper. I still think the paper is well written and the technical contributions seem good to me, but there might be some issues as mentioned by reviewers QdEN and MCkd. I especially agree with them that limitations are not sufficiently discussed in the paper.

---

### Official Review · Reviewer_NW3r · 2023-07-06

**Soundness:** 4 excellent
**Presentation:** 3 good
**Contribution:** 3 good
**Rating:** 6
**Confidence:** 4

**Summary:**

This paper undertakes a rigorous Bayesian treatment of the posterior probability of state-action optimality. It proposes a variational approach to approximate the state-action optimality. The proposed method involves a tractable convex optimization problem and is provably efficient. This paper also conducts experiments showing that the proposed method compares favorably with previous methods.

**Strengths:**

This paper, for the first time, undertakes a rigorous Bayesian treatment of the posterior probability of state-action optimality. It proposes a novel method having deep connections to previous work. Related work has been adequately cited.

This paper seems to be technically sound, though I did not check the proof in Appendix. All claims are well supported by theoretical analysis and experimental results. It is also clearly written and well organized.

I believe the result presented in this paper is significant. What I find most interesting is that the proposed method finds a balance between optimism and entropy regularization, and the resulting reward bonus is the product of surprise and uncertainty.

**Weaknesses:**

This paper formally treats the posterior probability of state-action optimality and demonstrates its connection to Thompson sampling. Similar analysis has been given in [O’Donoghue et al.](https://openreview.net/pdf?id=S1xitgHtvS). This should be mentioned in the paper.

The proposed method is novel. However, its connection to previous work, especially K-learning, should be discussed in more depth. Specifically, K-learning could be understood as a variational approximation of the policy induced by the state-action optimality ([O’Donoghue et al.](https://openreview.net/pdf?id=S1xitgHtvS)). VAPOR proposed in this paper additionally optimizes the stationary state distribution, which, to my understanding, complicates the optimization by introducing the need for forward message passing. It is not discussed in this paper whether and why this is beneficial.

In Figure 2, this paper compares the stationary state distribution induced by VAPOR and TS. I believe it should additionally provide a comparison with K-learning as K-learning also approximates the policy induced by the state-action optimality.

This paper provides a performance comparison in environments like DeepSea and Atari. As Bayesian approaches, it should also compare the Bayesian regret by simulating in MDPs randomly generated from prior.

**Questions:**

Is there a theoretical benefit for optimizing both the policy and stationary state distribution?
Is there an intuitive explanation for the balance between optimism and entropy regularization?

**Limitations:**

This paper has not discussed the limitations of the proposed method and the theoretical analysis.

---

> ### Author Rebuttal · Authors · 2023-08-10
>
> We thank the reviewer for their valuable comments and for highlighting connections to previous work, on which we will expand in the revised version. We answer their queries below:
>
> __Discussion of O’Donoghue et al.__ Thanks for highlighting that O’Donoghue et al., 2020 argue that TS solves at each episode an inference problem that samples from “the _joint_ [posterior] probability over all the binary [_action_] optimality variables”. While this is stated without proof nor rigorous definition of this joint quantity (which appears quite convoluted at first glance), our paper reveals that it can be condensely expressed: it represents the policy induced by the posterior probability of _state-action_ optimality (Def 1) (which is an occupancy measure). Thanks to this new definition, we can write a formal link between TS and RL as inference (Lemma 8).
>
> __Connection to K-learning.__ As you say, O’Donoghue et al., 2020 show that K-learning can be derived as an approximate inference procedure, although this hinges on the _assumption_ that inference is over the moment generating function of $Q^\star$ (Equation (7) of O’Donoghue et al., 2020). On the other hand, VAPOR’s derivation through our inference perspective is free from any assumption on the distribution of the optimality variables. Interestingly, we can show that VAPOR with equal temperature variables $\tau$ approximately recovers K-learning, thus shedding a new light on K-learning as a variational approximation of our probabilistic inference framework with an additional constraint of equal $\tau$ variables.
>
> __Suggestions of additional experimental insights.__ Thank you for the two suggestions (i.e., plot the occupancy measure of K-Learning in Figure 2, and compare the Bayesian regret of the approaches by simulating in MDPs randomly generated from a prior), which we will incorporate in the revised version.
>
> __Q1: Is there a theoretical benefit for optimizing both the policy and stationary state distribution?__ We show that the core quantity that formalizes RL as inference from a Bayesian viewpoint — the posterior probability of state-action optimality — is an occupancy measure (Lemma 1). Our variational approximation VAPOR naturally optimizes in this space. Intuitively, having a forward message passing enables the agent to capture prior episode information (i.e., to condition on prior steps in the episode being optimal), which is key to taking consistent actions (see our example in Figure/Table 1). Directly optimizing over occupancy measures is difficult in practice, thus we also propose VAPOR-lite which instead optimizes over the policies, akin to standard policy gradient methods (in this case, the optimization problem is no longer concave but good maxima can be found).
>
> __Q2: Is there an intuitive explanation for the balance between optimism and entropy regularization?__ This is a great question, in particular because it naturally falls out of our approach. One interpretation of having both exploration mechanisms work in tandem may be that optimism is providing a guidance on _which_ areas of the state space to visit next (i.e., uncertain states with high intrinsic reward). Entropy regularization, meanwhile, could be seen as providing guidance on _how_ to reach such desired states, where stochastic trajectories are preferred (which adds some local coverage/exploration).

---

> ### Comment · Reviewer_NW3r · 2023-08-12
> **Acknowledgement**
>
> I have read through all reviews and rebuttals and decided to keep my original score for acceptance.

---

### Official Review · Reviewer_3w7N · 2023-07-06

**Soundness:** 4 excellent
**Presentation:** 4 excellent
**Contribution:** 4 excellent
**Rating:** 10
**Confidence:** 4

**Summary:**

The paper identifies "the posterior probability of state-action optimality" denoted $\mathbb{P}_{\Gamma^*}$ as a key object for inference and control and provides a variational optimization approach to estimate it. Clear presentation, insightful analysis, and experiments on GridWorld, DeepSea, and Atari are provided.

**Strengths:**

Outstanding paper in all respects. A fresh and principled approach to RL as inference. High quality execution both on the theory and experimental side.

**Weaknesses:**

None

**Questions:**

None

**Limitations:**

There is no dedicated section for limitations, but limitations are sufficiently discussed in the conclusion.

---

> ### Author Rebuttal · Authors · 2023-08-09
>
> Thank you for the complimentary review! We are delighted that you think our work is valuable. We welcome any further comments.

---

### Decision · Program_Chairs · 2023-09-21

**Decision:**

Accept (poster)

**Comment:**

I am thrilled to convey my recommendation for the acceptance of your paper to be presented at NeurIPS. Your work won over both the reviewers and myself and I am confident that it will make a valuable contribution to the NeurIPS conference.

This is a highly intriguing paper with a lot of discussion. It is, effectively, an attempt to entirely reframe the RL as inference framework which works in the tabular setting, probably, and may be useful in more general settings after much future work. In the absence of strongly negative initial reviews which suggest something important about the clarity of the paper, and some remaining negative sentiment, I could have seen this being spotlight material.

Your responsiveness to the feedback provided during the review process reflects your commitment to advancing the field and strengthening your contribution. Your willingness to engage with the reviewers' insights demonstrates your dedication to delivering a high-quality presentation that will resonate with the conference attendees.

NeurIPS is a prestigious platform that attracts the brightest minds in the field, and your paper's acceptance adds to the conference's reputation for excellence. I am genuinely excited to see your work presented and discussed among peers who share your passion for pushing the boundaries of knowledge.

Congratulations once again on this well-deserved achievement.